# Discrete Tilt Matching

Yuyuan Chen [* 1]  Shiyi Wang [* 1]  Peter Potaptchik [1 2]  Jaeyeon Kim [1]  Michael S. Albergo [1 3 4]

## Abstract

Masked diffusion large language models (dLLMs) are a promising alternative to autoregressive generation. While reinforcement learning (RL) methods have recently been adapted to dLLM fine-tuning, their objectives typically depend on sequence-level marginal likelihoods, which are intractable for masked diffusion models. To address this, we derive Discrete Tilt Matching (DTM), a likelihood-free method that recasts dLLM fine-tuning as state-level matching of local unmasking posteriors under reward tilting. DTM takes the form of a weighted cross-entropy objective with explicit minimizer, and admits control variates that improve training stability. On a synthetic maze-planning task, we analyze how DTM's annealing schedule and control variates affect training stability and prevent mode collapse. At scale, fine-tuning LLaDA-8B-Instruct with DTM yields strong gains on Sudoku and Countdown while remaining competitive on MATH500 and GSM8K. Our code is available here.

## 1. Introduction

Fine-tuning large language models (LLMs) is critical for aligning models with downstream objectives and user preferences. For autoregressive LLMs, a mature set of reinforcement learning (RL) and preference-optimization methods has emerged (Schulman et al., 2017; Ouyang et al., 2022; Rafailov et al., 2024; Shao et al., 2024). Meanwhile, masked diffusion large language models (dLLMs) (Nie et al., 2025; Xie et al., 2025; Gong et al., 2025; Bie et al., 2025; Fan et al., 2026) have become a promising alternative to autoregressive models, enabling parallel and flexible any-order inference, potentially offering new avenues for modeling unconventional modalities such as structured inputs (Chang

et al., 2022; Sahoo et al., 2024; Shi et al., 2024; Campbell et al., 2024; Gong et al., 2025).

In light of this, a large body of recent studies has adopted RL post-training to dLLMs and reported empirical gains (Zhao et al., 2025; Wang et al., 2025a; Tang et al., 2025; Zhu et al., 2025; Yang et al., 2025; Rojas et al., 2025). However, these methods come with a fundamental, structural complication: the objectives depend on the model's *marginal likelihood* for a complete sequence, which is intractable for dLLMs. This is because a single terminal sequence can be obtained through various decoding orders; the likelihood thus entails aggregating over an exponentially large set of unmasking trajectories. This makes exact likelihood evaluation impractical and necessitates biased likelihood surrogates or high-variance estimators.

This suggests that post-training for dLLMs should be formulated in terms of tractable *state-level* quantities—namely, local unmasking posteriors on partially masked states—rather than *sequence-level* objectives based on marginal likelihood. In this work, we introduce *Discrete Tilt Matching (DTM)*, a likelihood-free fine-tuning method for dLLMs that recasts post-training as *state-level posterior matching under reward tilting*, rather than as LLM-style policy optimization (Shao et al., 2024). Specifically, given a pretrained dLLM with target distribution $\rho_1$ and a reward function $r(\cdot)$, we define the tilted target

$$\rho_{1,A}(x) \propto \rho_1(x)e^{Ar(x)},$$

where $A > 0$ is a scaling factor that controls the strength of the tilt. Rather than targeting $\rho_{1,A}$ in a single step, we follow recent work in continuous diffusion (Potaptchik et al., 2025; Guo et al., 2025) and progressively anneal a tilt parameter $a \in [0, A]$, performing incremental updates from $\rho_{1,a}$ to $\rho_{1,a+h}$.

We extend this incremental tilting perspective to discrete-space continuous-time Markov chains and derive DTM as a cross-entropy fine-tuning objective for dLLMs. The minimizer of the objective is *explicitly characterized* by the Markov chain (with rate matrix) corresponding to the reward-tilted target, enabling updates from $a$ to $a+h$ *without requiring sequence-level likelihood evaluation*. Our formulation also admits control variates that preserve the minimizer while reducing gradient variance, improving training

---

[*]Equal contribution, alphabetical order. [1]Harvard University [2]University of Oxford [3]IAIFI [4]Kempner Institute. Correspondence to: Yuyuan Chen <yuyuanchen@math.harvard.edu>, Shiyi Wang <fwang@math.harvard.edu>.

*Proceedings of the 43rd International Conference on Machine Learning*, Seoul, South Korea. PMLR 306, 2026. Copyright 2026 by the author(s).

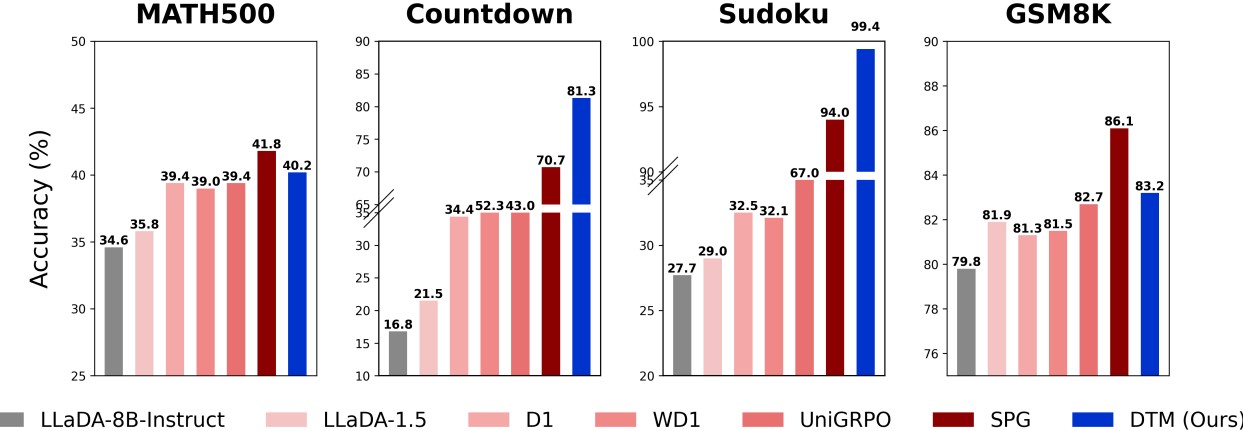

*Figure 1.* Accuracy of DTM and baseline methods on benchmarks. For each method, we pick the best accuracy between generation length 256 and 512, all denoising 2 tokens per step.

stability.

Empirically, we show that DTM remains competitive with, and often outperforms, prior RL-based fine-tuning efforts (Zhao et al., 2025; Tang et al., 2025; Wang et al., 2025a; Rojas et al., 2025; Yang et al., 2025) on LLaDA-8B-Instruct (Nie et al., 2025) across Sudoku, Countdown, MATH500, and GSM8K.

We highlight our **main contributions**:

- **Derivation of the DTM Algorithm:** We introduce Discrete Tilt Matching, a likelihood-free fine-tuning method for dLLMs that recasts post-training as state-level matching of unmasking posteriors. We derive a cross-entropy objective whose explicit minimizer is the reward-tilted posterior and which admits variance reduction via control variates.

- **Fine-Tuning LLaDA-8B-Instruct:** We validate DTM at scale by applying it to LLaDA-8B-Instruct, where it achieves particularly strong results on structured planning tasks such as Sudoku and Countdown while remaining competitive on mathematical reasoning benchmarks.

**Organization.** In Section 2, we provide an overview of the continuous-time Markov chain formulation of MDMs, as well as how to phrase reward fine-tuning in this context. From there, in Section 3, we describe Discrete Tilt Matching (DTM), a likelihood-free fine-tuning method for dLLMs. In Section 4, we provide empirical results on using DTM to fine-tune LLaDA-8B-Instruct (Nie et al., 2025).

## 2. Background

We review masked diffusion models (MDMs) as continuous-time Markov chains (CTMCs) on discrete sequences, and formalize fine-tuning as reward-tilting in this setting.

**Notation.** Let $\mathcal{V}$ be a vocabulary set and let $\mathbf{m}$ denote the special mask token. We denote $\mathcal{X} = (\mathcal{V} \cup \{\mathbf{m}\})^L$ for the space of length-$L$ sequences. For $x \in \mathcal{X}$, $x^i$ indicates the token at position $i \in [L]$. We use $\rho_t$ for the marginal distribution of a time-indexed stochastic process $X_t$.

### 2.1. Continuous-time Markov Chains: Dynamical Transport on Discrete State Space

A continuous-time markov chain (CTMC) $\{X_t\}_{t \in [0,1]}$ on $\mathcal{X}$ is a stochastic process with an associated family of rate matrices $\{R_t(x, y)\}_{t \in [0,1]}$ capturing the instantaneous transition likelihood from a state $x$ to a state $y$, with

$$\mathbb{P}[X_{t+\Delta t} = y \mid X_t = x] = \mathbf{1}_{x=y} + \Delta t \cdot R_t(x, y) + o(\Delta t) \tag{1}$$

for small $\Delta t > 0$. For $x \neq y$, $R_t(x, y) \geq 0$, and $R_t(x, x) = \sum_{y \neq x} -R_t(x, y)$. Intuitively, this captures the law of total probability.

The marginal distribution $\rho_t$ of $X_t$ evolves by the Kolmogorov forward equation

$$\partial_t \rho_t(x) = \sum_{y \in \mathcal{X}} \rho_t(y) R_t(y, x). \tag{2}$$

With generative modeling, given a prior distribution $\rho_0$ and target distribution $\rho_1$, the goal is to learn the rate matrices $\{R_t\}_{t \in [0,1]}$ such that (2) is satisfied with the correct marginals $\rho_{t=0} = \rho_0$ and $\rho_{t=1} = \rho_1$.

## 2.2. Masked Diffusion Models

Let $\rho_0$ be a point mass at the fully masked length-$L$ sequence $(\mathbf{m}, \cdots, \mathbf{m})$. MDMs (Chang et al., 2022; Sahoo et al., 2024; Shi et al., 2024; Campbell et al., 2024) instantiate a CTMC that transports this simple prior $\rho_0$ to a data distribution $\rho_1$ on clean sequences of length $L$. The evolution from $\rho_0$ to $\rho_1$ can thus be interpreted as gradual unmasking.

**Discrete stochastic interpolant.** To construct this transport between $\rho_0$ and $\rho_1$, one can use discrete stochastic interpolant, (Gat et al., 2024; Shaul et al., 2025; Kim et al., 2025a) which characterizes the intermediate distributions $\{\rho_t\}_{t \in (0,1)}$ in law. Let $\alpha : [0,1] \to [0,1]$ be a smooth, monotone schedule with $\alpha(0) = 0$ and $\alpha(1) = 1$. By sampling a clean $x_1 \sim \rho_1$, and for each $i \in [L]$ independently drawing an unmasking time $T^i$ with probability $\mathbb{P}[T^i < t] = \alpha(t)$ (or equivalently with density $T^i \sim \dot{\alpha}(t)dt$), we define the partially masked state $x_t$ by

$$x_t^i := \begin{cases} x_1^i, & t \geq T^i \\ \mathbf{m}, & t < T^i \end{cases}, \qquad (3)$$

Let $\rho_t := \text{Law}(x_t)$.

**From the interpolant to the CTMC rate matrices.** For the masking interpolant (3), the only transitions over an infinitesimal interval $[t, t+dt]$ are *single-coordinate reveals*: from a state $x$ we may unmask one masked position $i$ and move to $x[x^i \leftarrow v]$ for some $v \in \mathcal{V}$. Moreover, since each reveal time $T^i$ satisfies $\mathbb{P}[T^i < t] = \alpha(t)$, the instantaneous likelihood that a still-masked coordinate reveals in $[t, t+dt]$ is the hazard rate $\lambda(t)dt$ where

$$\lambda(t) = \frac{\dot{\alpha}(t)}{1 - \alpha(t)}.$$

Then the marginal law $\{\rho_t\}_{t \in [0,1]}$ induced by the interpolant (3) can be realized by a CTMC whose rate matrices are

$$R_t(x, x[x^i \leftarrow v]) = \lambda(t) \cdot \mathbb{P}[x_1^i = v \mid x_t = x], \quad (4)$$

i.e. $\lambda(t)$ is the instantaneous unmask event rate at the still-masked position $i$, and multiplying by conditional probability allocates that event to the specific token $v$.

**Unmasking posterior.** From (4), the key object needed for generation is the *unmasking posterior*

$$\pi^*(v \mid x_t, i) := \mathbb{P}[x_1^i = v \mid x_t],$$

i.e., the distribution of the clean token at a masked position $i$ given the current partially-masked sequence $x_t$. A neural network $\pi_\theta$ [1] approximates $\pi^*(v \mid x_t, i)$ and is trained by

---

[1]Slight abuse of notation: although we write $\pi_\theta(\cdot \mid x_t, i)$ as a conditional distribution over $\mathcal{V}$, in implementation the neural network $\pi_\theta$ takes in the partially masked sequence $x_t$ and outputs an $L$-by-$|\mathcal{V}|$ matrix of logits; $\pi_\theta(\cdot \mid x_t, i)$ corresponds to the $i$-th row of this matrix after softmax.

---

minimizing the ELBO-based loss

$$\mathcal{L}_{\text{MDM}}(\theta) = -\int_0^1 \lambda(t)\mathbb{E}\Big[\sum_{i \in [L]: x_t^i = \mathbf{m}} \log \pi_\theta(x_1^i \mid x_t, i)\Big]dt$$
$$(5)$$

where the expectation is taken over $x_1 \sim \rho_1$ and $x_t \sim \rho_t(\cdot \mid x_1)$. The unique minimizer is the ground-truth unmasking posterior $\pi^*$. More precisely, if $\rho_1^\theta$ is the terminal distribution induced by the parametric model, then

$$\text{KL}(\rho_1 \| \rho_1^\theta) \leq \mathcal{L}_{\text{MDM}}(\theta) - \min_{\theta'} \mathcal{L}_{\text{MDM}}(\theta'). \qquad (6)$$

Appendix C of Kim et al. (2025a) provides a more comprehensive overview of the framework of MDMs through the lens of discrete stochastic interpolants.

**MDM inference and the intractability challenge.** At inference time, given a learned unmasking posterior $\pi_\theta(\cdot \mid x_t, i)$, a simple sampling procedure using (1) discretizes the given CTMC into $N$ steps, starting from the fully-masked sequence, and iteratively selects $\lfloor L/N \rfloor$ masked positions and reveals them according to the prediction $\pi_\theta(\cdot \mid x_t, i)$ until all tokens are unmasked.

This inference procedure makes the terminal marginal likelihood $\rho_\theta(x_1)$ computationally intractable, as a single terminal sequence $x_1$ can be generated via exponentially many unmasking trajectories, i.e., orders of revealing tokens, and $\rho_\theta(x_1)$ marginalizes over all such trajectories. As a result, likelihood-based reinforcement learning objectives such as GRPO, which require computing $\log \rho_\theta(x_1)$, are not directly tractable.

Therefore, rather than adopting a likelihood-based RL viewpoint, we recast fine-tuning as *reward tilting* of the terminal distribution. As we will see, this perspective, combined with the structure of MDMs, yields *tractable objectives* that depend only on local unmasking posteriors, providing the framework for what we call *Discrete Tilt Matching*.

## 2.3. Fine-tuning as Reward-tilting

Let $r : \mathcal{V}^L \to \mathbb{R}$ be a reward function and $A > 0$ be a constant. Given a pre-trained MDM with unmasking posterior $\pi_0(\cdot \mid x_t, i)$ that samples from the data distribution $\rho_1$, we can frame the goal of fine-tuning as training a tilted unmasking posterior $\pi_A(\cdot \mid x_t, i)$ whose induced terminal marginal $\rho_{1,A}$ is given by

$$\rho_{1,A}(x) \propto \rho_1(x)e^{Ar(x)}. \qquad (7)$$

**Annealing from $a = 0$ to $a = A$.** Directly re-weighting samples under $\rho_1$ to be samples under $\rho_1(x)e^{Ar(x)}$ on which to perform supervised fine-tuning is usually an untenable strategy, as the weights have high variance and cause mode collapse (Guo et al., 2025). An alternative strategy is to

anneal toward the terminal target tilted distribution with a parameter $a \in [0, A]$, i.e. to set an intermediate target

$$\rho_{1,a}(x) \propto \rho_1(x)e^{ar(x)}. \tag{8}$$

Recent works (Potaptchik et al., 2025; Guo et al., 2025) have found it useful to phrase fine-tuning in this light, not only because it allows one to consider optimization procedures via a sequence of local subproblems whose minimizers remain close, but also to enable new objective functions that arise from dynamical relations between the different $\rho_{t,a}$.

To sample from the tilted distribution (8), a natural approach is to keep the same masked-diffusion CTMC structure as in (4), and simply replace the base unmasking posterior $\mathbb{P}[x_1^i = v \mid x_t]$ with one conditional on a new interpolant $x_{t,a}$. In light of this, we define the *tilted interpolant* by first drawing $x_1 \sim \rho_{1,a}$ and then applying the same schedule:

$$x_{t,a}^i := \begin{cases} x_1^i, & t \geq T^i, \\ \mathbf{m}, & t < T^i, \end{cases} \quad T^i \sim \dot{\alpha}(t)\,dt, \ \ x_1 \sim \rho_{1,a} \tag{9}$$

The induced marginals $\{\rho_{t,a}\}_{t \in [0,1]}$ can then be realized by the CTMC with rate matrices

$$R_{t,a}(x, x[x^i \leftarrow v]) = \lambda(t) \cdot \mathbb{P}_a[x_1^i = v \mid x_t]$$

where $\mathbb{P}_a$ denotes the tilted distribution induced by (9).

**Problem formulation.** Using this notation, we cast fine-tuning as an incremental problem: given at *tilt level* $a$, we aim to fine-tune it to match the next tilted distribution at level $a + h$, where $h > 0$ is a small annealing step size.

Fix $0 \leq a < A$. Suppose we have an unmasking posterior $\pi_a(\cdot \mid x_t, i)$ whose induced terminal marginal is $\rho_{1,a}$. Our objective is to train a further *tilted unmasking posterior*

$$\pi_\theta(\cdot \mid x_t, i) \approx \pi_{a+h}(v \mid x_t, i) = \mathbb{P}_{a+h}[x_1^i = v \mid x_t]$$

whose induced terminal marginal satisfies

$$\rho_{1,a+h}(x) \propto \rho_{1,a}(x)e^{hr(x)} \propto \rho_1(x)e^{(a+h)r(x)}.$$

## 3. Discrete Tilt Matching

Section 2.3 reduces fine-tuning under reward-tilting to the *incremental tilting problem*: given $\pi_a$ sampling $\rho_{1,a}$, train $\pi_\theta \approx \pi_{a+h}$ to sample from $\rho_{1,a+h}$. In this section, we derive Discrete Tilt Matching (DTM), a tractable objective for carrying out this update without requiring likelihood evaluation.

**The ideal objective.** If we had access to tilted samples $x_1 \sim \rho_{1,a+h}$ and could evaluate the ground-truth posterior $\pi_{a+h}$, then the most direct approach would be the standard

MDM objective with respect to the tilted unmasking posterior, i.e. minimize the cross-entropy on masked positions

$$\min_\theta \mathbb{E}_{a+h}\Big[ \sum_{i:\, x_t^i = \mathbf{m}} \mathrm{CE}\big(\pi_{a+h}(\cdot \mid x_t, i) \| \pi_\theta(\cdot \mid x_t, i)\big)\Big], \tag{10}$$

where $\mathrm{CE}(\cdot \| \cdot)$ denotes the cross-entropy. The above equation, however, is intractable, as we cannot sample from $x_1 \sim \rho_{1,a+h}$, i.e., $\pi_{a+h}(\cdot \mid x_t, i)$ is inaccessible.

### 3.1. Esscher Transform: Tilting as Conditional Reweighting

The idea then is to express the unknown tilt $a + h$ in terms of quantities under the base tilt $a$ to which we do have access. The path measure $\mathbb{P}_{a+h}$ of the tilted MDM is defined via $x_1 \sim \rho_{1,a+h}$ and then sampling $x_t$ via the same masking construction in (9). Crucially, by construction of the interpolant in (9), conditional on $x_1$, the masking process—and hence the conditional law of $x_t$ given $x_1$—is independent of the tilt level. The tilt changes only the marginal law of $x_1$ from $\rho_{1,a}$ to $\rho_{1,a+h}$. Since $\rho_{1,a+h}(x) \propto \rho_{1,a}(x)e^{hr(x)}$, it follows that the corresponding path measures are related by the Radon–Nikodym derivative

$$\frac{d\mathbb{P}_{a+h}}{d\mathbb{P}_a} = \frac{e^{hr(x_1)}}{\mathbb{E}_a\big[e^{hr(x_1)}\big]}. \tag{11}$$

Consequently, for any integrable functional $\varphi$ of the path,

$$\mathbb{E}_{a+h}[\varphi] = \frac{\mathbb{E}_a\big[e^{hr(x_1)}\varphi\big]}{\mathbb{E}_a\big[e^{hr(x_1)}\big]}, \tag{12}$$

which gives a way to rewrite the previously intractable expectation $\mathbb{E}_{a+h}$ in (10) as tractable importance-weighted expectations $\mathbb{E}_a$. Similarly, we have the conditional change-of-measure identity: for any $t \in [0, 1]$,

$$\mathbb{E}_{a+h}[G \mid x_t] = \frac{\mathbb{E}_a[Ge^{hr(x_1)} \mid x_t]}{\mathbb{E}_a[e^{hr(x_1)} \mid x_t]}. \tag{13}$$

Applying (13) with $G = \mathbf{1}\{x_1^i = v\}$, the left hand side of (13) becomes precisely the target unmasking posterior

$$\mathbb{E}_{a+h}[\mathbf{1}\{x_1^i = v\} \mid x_t] = \pi_{a+h}(v \mid x_t, i)$$

Estimating this ratio directly can be high-variance. However, clearing denominators yields the following equivalent characterization of the tilted posterior:

**Proposition 3.1.** *Fix $t \in [0, 1]$, partially masked $x \in \mathcal{X}$, and position $i$ with $x^i = \mathbf{m}$. Then $\pi_{a+h}(\cdot \mid x, i)$ is the unique probability simplex on $\mathcal{V}$ such that*

$$\mathbb{E}_a[e^{hr(x_1)}\pi_{a+h}(v \mid x, i) \mid x_t] = \mathbb{E}_a[e^{hr(x_1)}\mathbf{1}\{x_1^i = v\} \mid x_t] \tag{14}$$

*for every $v \in \mathcal{V}$.*

This is the key step toward making the ideal objective (10) tractable: it characterizes the unknown $\pi_{a+h}$ through known conditional weighted moments under $\mathbb{P}_a$.

**Remark.** The conditional reweighting identities (11)–(13) are a special case of the classical *Esscher transform* (i.e. exponential tilting) and, more generally, of a conditional tilting principle for arbitrary random targets and losses. For completeness, Appendix A presents a general tilted regression framework that recovers our setting by a particular choice of variables and Bregman divergence.

## 3.2. Moving beyond the Ideal World: DTM as a Variance-reduced Projection

Using (12) and (14), we now turn (10) into a tractable objective using only samples from $\mathbb{P}_a$ and evaluations of $\pi_a$.

**Fixing the expectation.** By (12), any $\mathbb{E}_{a+h}[\cdot]$ appearing in the ideal objective can be rewritten as an importance-weighted $\mathbb{E}_a[\cdot]$. Concretely, we can replace samples $x_1 \sim \rho_{1,a+h}$ in (10) by rollouts from $\mathbb{P}_a$ and weight each $x_1 \sim \rho_{1,a}$ by $e^{hr(x_1)}$.

**Fixing the target without evaluating $\pi_{a+h}$.** The remaining issue is the target distribution inside the cross-entropy. Proposition 3.1 shows that the weighted target moments we need are those of the one-hot indicator $\mathbf{1}\{x_1^i = v\}$, which is typically high variance. This motivates introducing a *random target distribution* $T_c(\cdot \mid x_t, i, x_1)$ (parameterized by a control parameter $c$) that is conditionally unbiased in the same weighted sense:

$$\mathbb{E}_a[e^{hr(x_1)}T_c(v \mid x_t, i, x_1) \mid x_t] = \mathbb{E}_a[e^{hr(x_1)}\mathbf{1}\{x_1^i = v\} \mid x_t]$$
(15)

for all $v \in \mathcal{V}$. Intuitively, $T_c$ replaces the raw one-hot target with a lower-variance random simplex while preserving the same weighted conditional mean as in (14). One concrete choice of $T_c$ is given below in (17).

**The $c$-DTM objective.** Given (15), we arrive at the (variance-controlled) Discrete Tilt Matching objective:

**Proposition 3.2** ($c$-DTM objective)**.** *Assume* (15) *holds for all* $(t, x, i)$*. Then the unique minimizer of*

$$\mathcal{L}_{a \to a+h}^{c\text{-}DTM}(\theta) = \int_0^1 \lambda(t)\,\mathbb{E}_a\Big[\sum_{i \in [L],\ x_t^i = \mathbf{m}} e^{hr(x_1)}$$
(16)
$$\cdot\,\mathrm{CE}\big(T_c(\cdot \mid x_t, i, x_1)\|\pi_\theta(\cdot \mid x_t, i)\big)\Big]\,dt.$$

*is the ground-truth tilted unmasking posterior* $\pi_{a+h}$*, i.e.*

$$\pi_{\theta^*}(\cdot \mid x_t, i) = \pi_{a+h}(\cdot \mid x_t, i).$$

*Proof sketch.* Fix $(t, x, i)$ and consider minimizing the conditional objective $\mathbb{E}_a[e^{hr(x_1)}\mathrm{CE}(T_c\|\pi) \mid x_t]$ over $\pi$ on the

simplex over $\mathcal{V}$. By convexity of cross-entropy, the minimizer is the normalized conditional mean of $e^{hr(x_1)}T_c(\cdot)$ given $x_t$. Using (15) and Proposition 3.1, this normalized mean is precisely $\pi_{a+h}(\cdot \mid x_t, i)$. $\qquad\square$

**A concrete unbiased target with control variate.** One concrete target that satisfies (15) is

$$T_c(v \mid x_t, i, x_1) := \frac{c\pi_a(v \mid x_t, i) + (e^{hr(x_1)} - c)\mathbf{1}\{v = x_1^i\}}{e^{hr(x_1)}},$$
(17)

where in most generality $c$ can be any $\sigma(x_t)$-measurable function. It recovers the simple one-hot target $\mathbf{1}\{v = x_1^i\}$ when $c = 0$. By construction, we check that

$$\mathbb{E}_a[e^{hr(x_1)}T_c(v \mid x_t, i, x_1) \mid x_t]$$
$$= \mathbb{E}_a[c\,\pi_a(v \mid x_t, i) + (e^{hr(x_1)} - c)\mathbf{1}\{x_1^i = v\} \mid x_t]$$
$$= \mathbb{E}_a[e^{hr(x_1)}\mathbf{1}\{x_1^i = v\} \mid x_t]$$

since by definition $\pi_a(v \mid x_t, i) = \mathbb{E}_a[\mathbf{1}\{x_1^i = v\} \mid x_t]$. This establishes (15), and concludes our derivation of the $c$-DTM loss function as a principled and tractable objective with guaranteed unique minimizer and requiring no likelihood computations.

In practice, with sufficiently small $h$, it would be convenient to choose $c = 1$:

**Proposition 3.3.** *For sufficiently small $h > 0$,*

$$\mathrm{Var}[\nabla\mathcal{L}_{a \to a+h}^{0\text{-}DTM}(\theta)] \geq \mathrm{Var}[\nabla\mathcal{L}_{a \to a+h}^{1\text{-}DTM}(\theta)].$$
(18)

We present a proof of (18) in Appendix B.

## 3.3. DTM Loss Bounds KL on Terminal Law

Via the reweighting by the unmasking schedule, the $c$-DTM objective (16) inherits the standard MDM principle that controlling local posteriors controls the terminal marginal. Analogous to (6), we have

**Proposition 3.4.** *Let $\rho_\theta$ be the terminal marginal distribution on $\mathcal{V}^L$ induced by the parametric model's unmasking posterior $\pi_\theta$. Then*

$$\mathrm{KL}(\rho_{1,a+h}\|\rho_\theta) \leq \frac{1}{Z}\Big(\mathcal{L}_{a \to a+h}^{c\text{-}DTM}(\theta) - \mathcal{L}_{a \to a+h}^{c\text{-}DTM}(\theta^*)\Big),$$
(19)

*where $Z = \mathbb{E}_a[e^{hr(x_1)}]$.*

A proof is given in Appendix B.

# 4. Experiments

In this section, we evaluate DTM on both controlled synthetic settings and large-scale dLLM fine-tuning. In 4.1, we first describe two practical interventions: aligning DTM

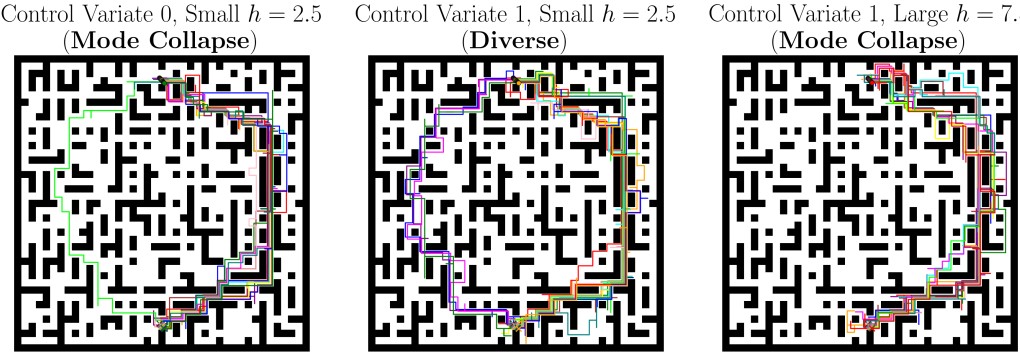

*Figure 2.* Comparison of performance on maze planning task for DTM with and without the control variate.

with semi-autoregressive (SAR) decoding, a commonly used dLLM inference strategy, and improving sample efficiency via a replay buffer. In 4.2, we then implement DTM on a synthetic maze-planning task to study the effects of the control variate and annealing step size on stability, reward, and diversity. Finally, in 4.3, we fine-tune LLaDA-8B-Instruct with DTM on math reasoning and planning benchmarks, and complement the main results with ablations and a wallclock comparison against prior RL-based methods.

### 4.1. Practical Interventions

**DTM is adaptive to semi-autoregressive decoding.** Many state-of-the-art dLLMs are deployed with semi-autoregressive (SAR) decoding, generating blocks autoregressively while allowing any-order updates within each block (Han et al., 2023; Arriola et al., 2025; Nie et al., 2025; Kim et al., 2025b). This makes it desirable to align the $x_t$'s masking pattern encountered during training to match those arising from SAR decoding, rather than from random masking over the full sequence.

Our key observation is that DTM's guarantee naturally extends to this setup. Since the $c$-DTM loss is defined pointwise in $(t, x_t, i)$ pair, its minimizer depends on the decoding procedure *only* through the distribution of the conditioning states $x_t$ and the set of masked indices $i$. Therefore, to align DTM with SAR decoding, it suffices to replace the fully random-masking interpolant (9) with an SAR-compatible one and restrict the per-time summation to the active block.

The only further modification is to replace the hazard $\lambda(t)$ with the one induced by the SAR interpolant, which preserves the same KL-control guarantee through the corresponding reweighting of local cross-entropies. We present the SAR interpolant, its hazard, and the resulting SAR-aligned $c$-DTM objective in Appendix C.

**Replay buffer.** A practical advantage that DTM offers is that it directly optimizes intermediate training states $x_t$, rather than necessarily requiring a fresh, online rollout for

every iteration. Consequently, a single rollout and reward pair $(x_1, r(x_1))$ can be recycled many times, as many distinct training states $x_t$'s can be produced per $x_1$ via the stochastic interpolant (9).

We therefore maintain a replay buffer of terminal samples and their rewards. To form each minibatch, we first sample $x_1$ from the buffer and then construct fresh intermediate states $x_t$ via the interpolant, which requires *no additional neural network calls*. The buffer is refreshed periodically with newly generated rollouts. This design substantially amortizes the cost of expensive online rollouts.

### 4.2. Maze Planning

We first study DTM on a synthetic maze-planning task, which provides a controlled and visualizable setting to examine how the control variate $c$ and annealing step size $h$ affect reward, validity, and solution diversity. We find that both materially affect training: using the control variate together with a smaller annealing step improves stability and preserves diverse high-reward solutions, whereas removing the control variate or taking overly aggressive tilting steps leads to visible mode collapse.

**Task.** At a high level, an MDM is given the start and goal coordinates in a fixed maze (see Fig. 7) and is tasked with generating a valid path connecting them without hitting walls. Concretely, a prompt has the format $(\text{s}, \text{g}, \text{SEP}, \mathbf{m}, \dots, \mathbf{m})$, where SEP, s, and g denote the separation token, start and goal, respectively. The MDM then generates a sequence $(\text{s}, \text{g}, \text{SEP}, z_1, z_2, \dots, z_n, \text{PAD}, \dots, \text{PAD})$, where PAD is a padding token and $(z_1, \dots, z_n)$ corresponds to a path in the maze grid.

**Setup.** We start from a pretrained MDM that already solves the task well on a fixed $41 \times 41$ maze. We then further fine-tune this model using DTM with a task-specific reward. Concretely, we employ a *stay-away-from-center* reward, which assigns to each valid path the minimum Manhattan distance, over all cells along the path, to the center of the

*Table 1.* Model performance on four reasoning benchmarks. The best results are bolded and the second best are underlined.

| MODEL / SEQ LEN | MATH500 (0-SHOT) | | COUNTDOWN (0-SHOT) | | SUDOKU (3-SHOT) | | GSM8K (0-SHOT) | |
|---|---|---|---|---|---|---|---|---|
| | 256 | 512 | 256 | 512 | 256 | 512 | 256 | 512 |
| LLaDA-8B-INST. | 32.4 | 34.6 | 16.8 | 16.8 | 27.7 | 26.2 | 77.2 | 79.8 |
| LLaDA-1.5 | 32.2 | 35.8 | 21.1 | 21.5 | 26.9 | 29.0 | 80.5 | 81.9 |
| D1 | 36.0 | 39.4 | 30.9 | 34.4 | 32.5 | 29.3 | 80.6 | 81.3 |
| WD1 | 37.4 | 39.0 | 52.3 | 50.8 | 32.1 | 22.5 | 81.5 | 80.3 |
| UNIGRPO | 37.4 | 39.4 | 43.0 | 57.0 | 67.0 | 62.9 | 82.5 | 82.7 |
| SPG | **40.0** | **41.8** | 70.7 | 70.3 | 94.0 | 93.1 | **86.1** | **84.5** |
| DTM (OURS) | 36.0 | 40.2 | **81.3** | **78.9** | **99.2** | **99.4** | 81.6 | 83.2 |

grid. A path is considered valid if it connects s and g without hitting walls or making jumps; invalid paths receive a negative penalty. To isolate the stability effects of our design choices, we ablate (i) the control variate $c \in \{0, 1\}$ in the $c$-DTM target, and (ii) the annealing step size $h \in \{2.5, 7.5\}$, holding all else fixed and scaling the number of steps per $h$-phase so each setting uses the same compute. We defer more details to App. D.1.

**Result.** Figure 2 and Figure 5 show that the annealing step size and control variate can materially affect the training quality: using the control variate ($c = 1$) with a smaller step size yields higher reward and validity while maintaining substantially better coverage, whereas either removing the control variate ($c = 0$) or taking overly aggressive annealing steps (large $h$) leads to a visually apparent mode collapse (rollouts mostly concentrate on a few near-identical paths). This suggests that it is crucial in practice to balance the aggressiveness of the tilting step size against optimization stability: $h$ should be large enough to make meaningful progress, but small enough—and paired with the control variate—to avoid collapse and preserve solution diversity.

### 4.3. Fine-tuning LLaDA-8B-Instruct

We now present results that demonstrate the performance of DTM *at scale* by fine-tuning LLaDA-8B-Instruct (Nie et al., 2025), which has been a common testbed for post-training algorithms for dLLMs.

**Tasks and Baselines.** Following prior protocols, we focus on math reasoning and planning benchmarks; GSM8K (Cobbe et al., 2021), MATH500 (Lightman et al., 2024), Countdown (Pan et al., 2025), and Sudoku (Arel, 2025). For a fair comparison, we adapt the prompt formatting and training-test splitting from SPG (Wang et al., 2025a). All training and evaluation are conducted in the zero-shot setting, except for Sudoku where we use 3-shot prompts for both training and evaluation due to the base model's difficulty with Sudoku. We benchmark against recent RL methods for fine-tuning dLLMs, including LLaDA-1.5 (Zhu et al., 2025), d1 (Zhao et al., 2025), wd1 (Tang et al., 2025), Uni-

GRPO (Yang et al., 2025), and SPG (Wang et al., 2025a).

**Implementation details.** Across all tasks, we fine-tune the model with Low-Rank Adaptation (LoRA) (Hu et al., 2022) under rank $r = 128$ and scaling factor $\alpha = 64$, following all of the baselines. During the training rollout, we adopt SAR sampling with generation length 256 and block size 32 and train with the SAR-aligned DTM objective (34). We decode 2 tokens per step for a total of $T = 128$ steps. These are all standard design choices aligned with all our baselines.

For all four tasks, we use confidence-based within-block updates (selecting the to-be-revealed positions by the model's confidence in the sampled token). We use sampling temperature 1.0 for GSM8K, MATH500, and Countdown to encourage diverse rollouts, and 0.0 for Sudoku as it has a single correct completion per prompt. Finally, for all tasks we use the $c$-DTM objective with control variate $c = 1$ and set the annealing step size to $h = 2.5$ for Sudoku, $h = 6$ for Countdown, and $h = 1$ for GSM8K and MATH500.

At evaluation, aligned with all baselines, we by default employ a confidence-based and semi-autoregressive strategy with block size 32 at temperature $\tau = 0.0$.

**Results.** The complete results on the performance of DTM are summarized in Table 1, in comparison with the base model and other baselines. DTM delivers strong gains over the base LLaDA-8B-Instruct on planning benchmarks, and is particularly effective on Sudoku: with the same evaluation protocol and SAR decoding as prior work, DTM reaches **99.2%** accuracy at length 256 (and **99.4%** at length 512), substantially improving over the base model (27.7/26.2) and surpassing the strongest prior baselines (e.g., SPG at 94.0/93.1). On Countdown, DTM also achieves the best performance at both length 256 (**81.6%**) and length 512 (**76.6%**).

### 4.4. Additional Qualitative Analysis and Ablation

**Local posterior matching and reasoning.** On MATH500 and GSM8K, DTM improves upon the base model but is less competitive than the top-performing baselines. We hy-

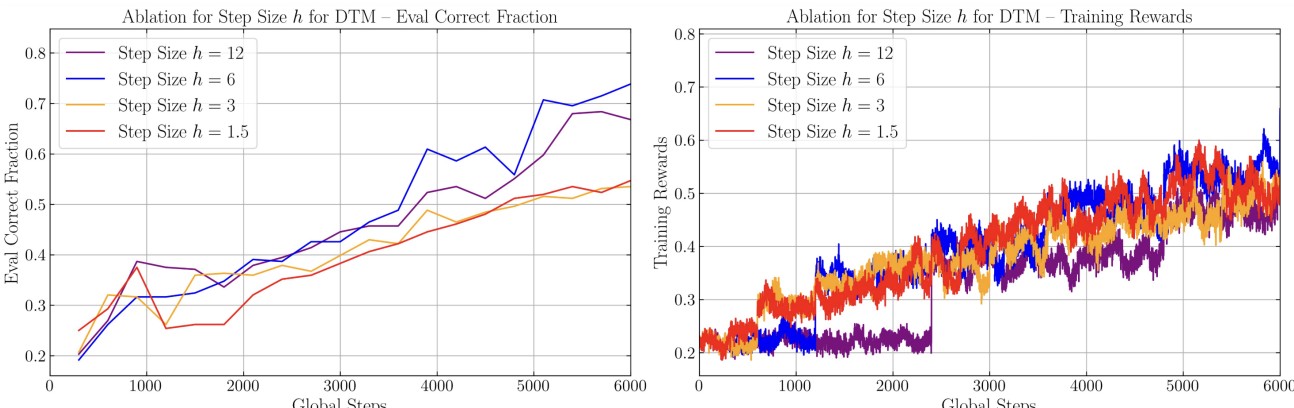

*Figure 3.* Ablation on annealing step size $h$ on Countdown. Left figure shows the correct fraction on the evaluation set of the model checkpoints. Right figure shows the training reward trajectory. A moderate step size $h = 6$ achieves the best result.

pothesize that this reflects a difference between improving *state-level local predictions* and achieving *globally coherent long-horizon reasoning*: DTM trains directly on local unmasking posteriors at intermediate denoising states, which can improve local reasoning consistency, but harder mathematical reasoning problems place stronger demands on maintaining coherent reasoning over longer and more diverse decoding trajectories. In such settings, better state-level posterior prediction may translate *less directly* into final-answer accuracy. Still, DTM improves substantially with additional decoding budget on MATH500 and GSM8K, rising from 36.0 and 81.6 at generation length 256 to 40.2 and 83.2 at length 512, which is consistent with the view that stronger local predictions can be converted into better long-horizon reasoning when given more inference steps.

To further support the claim that DTM's direct state-level training improves reasoning consistency, we provide additional qualitative evidence from Countdown, where reasoning is more structured and easier to inspect. As shown in Fig. 6, DTM produces a more stable effective generation length than sequence-level RL baselines, suggesting less reward hacking and more stable intermediate reasoning.

**Wallclock Comparison.** On the tasks where DTM outperforms the strongest RL baselines, we show that DTM achieves higher rewards more efficiently. Under the same training setup on 8 H100 GPUs, we compared DTM to SPG on the Sudoku dataset. As shown in Figure 4, DTM converges more quickly and to a higher reward.

**Ablation on $h$.** We ablate on the annealing step size and perform training on Countdown with $h \in \{1.5, 3, 6, 12\}$, while matching the overall compute budget and keeping the remaining hyperparameters fixed. As shown in Fig. 3, the best performance is attained with $h = 6$, while DTM remains competitive across other choices of $h$, which indicates that the method is reasonably *robust* to the choice of $h$. We did

not observe instability caused by the importance weighting in these runs. Empirically, the main effect of $h$ appears to be the expected tradeoff: when $h$ is too small, each phase makes only a limited update to the policy, and errors due to insufficient minimization in each phase accumulate; when $h$ is too large, each phase tilts toward a target that is farther away and optimization becomes less effective, with a greater risk of mode collapse (also consistent with (Guo et al., 2025) and our Fig. 2 ablation). A moderate value ($h = 6$) provides the best balance between these two effects, achieving high convergence without mode collapse: Countdown questions naturally have two modes: ones solvable with addition/-subtraction and ones requiring multiplication/division; the second mode is tiny with base LLaDA achieving only 1.95% accuracy, yet our $h = 6$ fine-tuned model is able to *preserve the mode* and improve it to 2.78% accuracy, whereas this mode collapsed (0%) in the SPG fine-tuned model.

## 5. Related Work

**Reinforcement Learning for Fine-tuning Masked Diffusion LLMs.** A rapidly growing line of work adapts online RL and preference optimization to dLLMs, but faces a central obstacle: policy-optimization objectives require (explicitly or implicitly) evaluating likelihood ratios or sequence-level likelihood surrogates. Early approaches such as Zhao et al. (2025) introduce diffusion-compatible variants of group-based policy optimization by leveraging tractable *local* unmasking quantities, and subsequent works explore alternative estimators and objectives to reduce the bias and/or variance induced by the inherent likelihood intractability, including weighted policy-optimization objectives (Tang et al., 2025), masking-structure-based likelihood/trajectory estimators (Wang et al., 2025b), variance-reduced ELBO-based preference optimization (Zhu et al., 2025), lower-variance ELBO approximations for group objectives (Rojas et al., 2025), and bound-sandwiching strategies to mitigate ELBO-

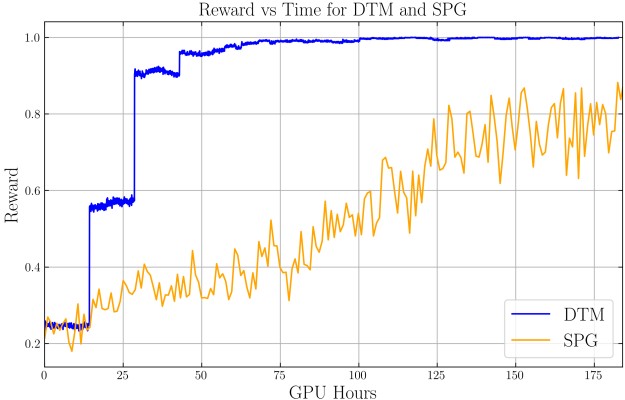

*Figure 4.* Wallclock comparison for DTM and SPG on Sudoku, both trained on 8 H100 GPUs, with reward normalized to $[0, 1]$. DTM attains a higher reward and is more efficient. Since DTM reward is evaluated for frozen model $\pi_a$ within each $a \mapsto a + h$ phase, the reward is roughly constant per phase, and jumps at the phase boundary when the model $\pi_a$ is updated to $\pi_\theta \approx \pi_{a+h}$ as in Algorithm 1. The SPG reward is evaluated on the training batch, thus showing a continual increase.

induced bias (Wang et al., 2025a). Relatedly, Luo et al. (2026) proposes a direct group-preference optimization objective that avoids a policy-gradient likelihood-ratio form, enabling efficient deterministic samplers in some diffusion settings.

As we noted in the introduction, our approach is orthogonal: rather than improving or approximating marginal-likelihood surrogates, DTM derives a *local* projection objective with theoretical guarantee of the ground-truth minimizer for the incremental update $\pi_a \mapsto \pi_{a+h}$ that bypasses likelihood evaluation altogether, while still supporting practical inference procedures (including semi-autoregressive variants).

**Tilt Matching for Continuous Diffusion** Our work builds most directly on *Tilt Matching* (TM), which studies exponential reward tilting for continuous-space diffusion/flow models by viewing fine-tuning as an $a$-*indexed evolution* of intermediate targets $\rho_{1,a} \propto \rho_1 e^{ar}$ and learning update rules that move from $a$ to $a+h$ rather than attempting a one-shot tilt (Potaptchik et al., 2025). TM derives conditional (Esscher) relations that express tilted dynamics using expectations under the current model, motivating incremental annealing in $a$, practical objectives, and variance-reduction strategies.

While DTM borrows this core *incremental tilting* philosophy, our contribution is not a routine translation to a new setting: the discrete CTMC / masked-diffusion domain changes the object being learned and forces a different set of technical and practical resolutions. In particular, discrete MDMs are naturally specified by a CTMC generator whose learnable component is a local *unmasking posterior*; connecting

this to the interpolant-induced marginal path yields a concrete rate factorization and lets us derive a *cross-entropy* fine-tuning objective that is aligned with base MDMs pretraining and the discrete space geometry while still providing a principled control of the induced terminal distribution (via a KL bound tied to the loss). Moreover, because state-of-the-art dLLMs commonly rely on semi-autoregressive decoding at inference, we show how to modify the interpolant and time-weighting so that training matches the *same* family of partially-masked states encountered at test time, improving train-test alignment without changing the population optimum. Finally, we validate these discrete-specific design choices at scale by fine-tuning LLaDA-8B-Instruct and demonstrating that DTM remains effective in the large-model regime where practical stability and decoding alignment matter most.

## 6. Conclusion

In this paper, we introduce Discrete Tilt Matching (DTM). DTM views reward fine-tuning as progressive distribution tilting and leverages the continuous-time Markov chain structure of masked diffusion language models. DTM bypasses intractable likelihood issues of masked diffusions, but also is provably minimized at the desired tilted unmasking posterior.

**Broader viewpoint.** The success of RL-style post-training for autoregressive language models has led to an implicit premise that similar techniques should extend seamlessly to diffusion language models. Broadly speaking, our work questions this; the likelihood-based RL-style post-training may not be the ideal framework in the setup of MDMs. Viewed this way, DTM represents a step that *opens the door* to alternative post-training paradigms that are better aligned with the structure of diffusion-based language models.

## Impact Statement

This paper presents work whose goal is to advance the field of Machine Learning. There are many potential societal consequences of our work, none which we feel must be specifically highlighted here.

## Acknowledgement

YC and SW contributed equally to this work and agree that the order of their names may be exchanged as useful to highlight their contributions in individual professional pursuits. PP is supported by the EPSRC CDT in Modern Statistics and Statistical Machine Learning [EP/S023151/1], a Google PhD Fellowship, and an NSERC Postgraduate Scholarship (PGS D). MSA is supported by a Junior Fellowship at the Harvard Society of Fellows as well as the National Science

Foundation under Cooperative Agreement PHY-2019786 (The NSF AI Institute for Artificial Intelligence and Fundamental Interactions[2] ). This work has been made possible in part by a gift from the Chan Zuckerberg Initiative Foundation to establish the Kempner Institute for the Study of Natural and Artificial Intelligence.

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

---

**Algorithm 1** DTM Training

---

**Require:** Unmasking posterior $\pi_0(\cdot \mid x, i)$ of base model, annealing step size $h$, terminal reward multiplier $A$, reward function $r : \mathcal{V}^L \to \mathbb{R}$, control variate $c$, learning rate $\eta$, batch size $s$, replay buffer size $N$, buffer refresh interval $K$
1: Initialize $a \leftarrow 0$ and $\pi_\theta \leftarrow \pi_0$
2: **while** $a < A$ **do**
3:     Initialize step $\leftarrow 0$
4:     **while** not converged **do**
5:         **if** step $\mathrm{mod}\, K = 0$ **then**
6:             $\mathcal{D} \leftarrow$ BUILDBUFFER$(\pi_a, r, N)$
7:         **end if**
8:         Draw minibatch $\mathcal{B} \sim$ SAMPLE$(\mathcal{D}, s)$
9:         For each clean sample $x_1$ in $\mathcal{B}$, independently draw a random time $t \in [0, 1]$ and create partially masked sequence $x_t$ according to (3)
10:        Compute loss $\mathcal{L}$ in (16) by empirical mean of the minibatch
11:        Gradient descent on model $\theta \leftarrow \theta - \eta \nabla_\theta \mathcal{L}$
12:        step $\leftarrow$ step $+1$
13:     **end while**
14:     Update $\pi_{a+h} \leftarrow \pi_\theta$
15:     Update $a \leftarrow a + h$
16: **end while**
17: **return** $\pi_\theta$

---

**Algorithm 2** BUILDBUFFER: Roll out SAR sampling

---

**Require:** Unmasking posterior $\pi_a$, reward $r$, buffer size $N$, total denoising steps $T$, block size $B$
1: $\mathcal{D} \leftarrow \emptyset$
2: **for** $n = 1, \ldots, N$ **do**
3:     $X \leftarrow \mathbf{m}^L$ {fully masked length-$L$ sequence (optionally prepend a prompt)}
4:     $M \leftarrow L/B$ {# blocks; assume $B|M$ for ease}
5:     $S \leftarrow T/M$ {steps per block; assume $M|T$}
6:     **for** $b = 0, \ldots, M - 1$ **do**
7:         $\mathcal{J} \leftarrow \{bB + 1, \ldots, (b + 1)B\}$ {current block indices}
8:         **for** $s = 1, \ldots, S$ **do**
9:             $U \leftarrow \{i \in \mathcal{J} : X^i = \mathbf{m}\}$ {masked positions in current block}
10:          $\mathcal{I} \leftarrow$ sample $B/S$ indices uniformly from $U$ {random order within block}
11:          $P \leftarrow \pi_a(\cdot \mid X)$ {*one* NN call, $L$-by-$|\mathcal{V}|$ matrix}
12:          **for** each $i \in \mathcal{I}$ **do**
13:             sample $v \sim P[i, \cdot]$; set $X^i \leftarrow v$
14:          **end for**
15:         **end for**
16:     **end for**
17:     $\mathcal{D} \leftarrow \mathcal{D} \cup \{(X, r(X))\}$
18: **end for**
19: **return** $\mathcal{D}$

---

# A. General Esscher Transform

This appendix presents a general change-of-measure identity generalizing the derivation in the main text. At a high level, our discrete Esscher reweighting (Section 3.1) is an instance of the classical *Esscher transform* (exponential tilting), together with a standard fact that minimizing a weighted Bregman divergence reduces to a conditional expectation under the tilted measure.

## A.1. Classical Esscher transform (exponential tilting)

Let $Z$ be a real-valued random variable on a probability space $(\Omega, \mathcal{F}, \mathbb{P})$ such that $\mathbb{E}[e^{hZ}] < \infty$ for a given $h \in \mathbb{R}$. The (classical) Esscher transform with parameter $h$ defines a new probability measure $\mathbb{P}^{(h)}$ via the Radon–Nikodym derivative

$$\frac{d\mathbb{P}^{(h)}}{d\mathbb{P}} = \frac{e^{hZ}}{\mathbb{E}[e^{hZ}]}.$$

Equivalently, expectations under $\mathbb{P}^{(h)}$ are given by $\mathbb{E}^{(h)}[f] = \frac{\mathbb{E}[e^{hZ}f]}{\mathbb{E}[e^{hZ}]}$ for any integrable $f$. This transform goes back to Esscher (1932).

## A.2. A conditional Esscher transform

We now state the conditional identity we repeatedly exploit.

Let $(\Omega, \mathcal{F}, \mathbb{P})$ be a probability space. Let $X : \Omega \to \mathcal{X}$ be a random variable taking values in a space $\mathcal{X}$, and let $U : \Omega \to \mathcal{U}$ and $T : \Omega \to \mathcal{T}$ be arbitrary random variables (possibly dependent on $X$) taking values in $\mathcal{U}$ and $\mathcal{Y}$, respectively. Let $w : \mathcal{U} \to \mathbb{R}$ be a measurable function such that $\mathbb{E}[w(U)] < \infty$.

**Definition A.1** (Exponential tilt / Esscher-tilted measure). Define the tilted measure $\mathbb{P}'$ by

$$\frac{d\mathbb{P}'}{d\mathbb{P}} = \frac{w}{\mathbb{E}[w]}. \tag{20}$$

We write $\mathbb{E}$ (resp. $\mathbb{E}'$) for expectation taken under $\mathbb{P}$ (resp. $\mathbb{P}'$).

**Lemma A.2** (Conditional reweighting). *For any integrable $f : \mathcal{X} \times \mathcal{U} \times \mathcal{T} \to \mathbb{R}$,*

$$\mathbb{E}'[f(X, U, T) \mid X] = \frac{\mathbb{E}[w \, f(X, U, T) \mid X]}{\mathbb{E}[w \mid X]}.$$

*Proof.* This is the standard conditional change-of-measure identity:

$$\mathbb{E}'[f \mid X] = \frac{\mathbb{E}\left[f \frac{d\mathbb{P}'}{d\mathbb{P}} \mid X\right]}{\mathbb{E}\left[\frac{d\mathbb{P}'}{d\mathbb{P}} \mid X\right]} = \frac{\mathbb{E}[wf \mid X]}{\mathbb{E}[w \mid X]},$$

where we substituted (20) and canceled the (deterministic) constant $\mathbb{E}[w]$. $\square$

## A.3. Tilted Bregman regression

Let $\Phi : \mathbb{R}^d \to \mathbb{R}$ be strictly convex and twice continuously differentiable. Its associated Bregman divergence is

$$D_\Phi(y\|z) := \Phi(y) - \Phi(z) - \langle \nabla\Phi(z), y - z \rangle.$$

Consider functions $f : \mathcal{X} \to \mathbb{R}^d$ and the weighted objective

$$\mathcal{L}(f) := \mathbb{E}[w \, D_\Phi(T\|f(X))]. \tag{21}$$

**Proposition A.3** (Tilted regression lemma). *The unique minimizer $f^\star$ of (21) over measurable $f$ is*

$$f^\star(x) = \mathbb{E}'[T \mid X = x] = \frac{\mathbb{E}[wT \mid X = x]}{\mathbb{E}[w \mid X = x]}. \tag{22}$$

*Proof.* Fix $x$ and define $F_x(z) := \mathbb{E}[w\, D_\Phi(T\|z) \mid X = x]$. Using $\nabla_z D_\Phi(y\|z) = \nabla^2 \Phi(z)\,(z - y)$, we obtain

$$\nabla F_x(z) = \mathbb{E}[w\, \nabla^2 \Phi(z)(z - T) \mid X = x] = \nabla^2 \Phi(z)\Big(z\, \mathbb{E}[w \mid X = x] - \mathbb{E}[wT \mid X = x]\Big).$$

Since $\nabla^2 \Phi(z)$ is positive definite and $F_x$ is strictly convex in $z$, the unique minimizer satisfies $z\, \mathbb{E}[w \mid X = x] = \mathbb{E}[wT \mid X = x]$, i.e. $z = f^\star(x)$ as in (22). The equality to $\mathbb{E}'[T \mid X = x]$ follows from Lemma A.2. $\qquad\square$

### A.4. A control variate that preserves the minimizer

The lemma above remains valid if we replace $T$ by any random target $T_c$ satisfying $\mathbb{E}[wT_c \mid X] = \mathbb{E}[wT \mid X]$. A simple construction yields a family of such targets.

Let $\mu(X) := \mathbb{E}[T \mid X]$. For any measurable $c : \mathcal{X} \to \mathbb{R}$, define

$$T_c := T + \frac{c(X)}{w}\big(\mu(X) - T\big).$$

In particular, $T_c$ recovers $T$ with $c \equiv 0$. Then, conditioning on $X$,

$$\mathbb{E}[wT_c \mid X] = \mathbb{E}[wT \mid X] + c(X)\big(\mu(X) - \mathbb{E}[T \mid X]\big) = \mathbb{E}[wT \mid X].$$

Therefore replacing $T$ with $T_c$ in (21) yields the same minimizer:

**Corollary A.4** (Control variate preserves the tilted minimizer). *Let*

$$\mathcal{L}_c(f) := \mathbb{E}[w\, D_\Phi(T_c\|f(X))]$$

*Then* $\arg\min_f \mathcal{L}_c(f) = \arg\min_f \mathcal{L}(f)$ *and the unique minimizer is still* $f^\star(x) = \mathbb{E}_{a+h}[T \mid X = x]$.

### A.5. Recovering the identities used in the main text

Our discrete Esscher identities (Section 3) are obtained by instantiating the above with:

- $X = x_t$ (the partially masked state), $U = x_1$ (the terminal clean sequence), and $w = e^{hr(U)}$;

- $T$ chosen as a one-hot indicator $\mathbf{1}\{x_1^i = v\}$ or the target with control variate (17), and $\Phi$ chosen to be the negative Shannon entropy so that $D_\Phi$ is the KL divergence, which up to an additive constant corresponds to the per-token training loss.

In particular, Lemma A.2 specializes exactly to the conditional reweighting formula $\mathbb{E}_{a+h}[\cdot \mid x_t] = \mathbb{E}_a[w(\cdot) \mid x_t]/\mathbb{E}_a[w \mid x_t]$ used to derive (13) in the main text.

## B. Proofs

**Proposition 3.2** (*c*-DTM objective). *Assume* (15) *holds for all* $(t, x, i)$. *Then the unique minimizer of*

$$\mathcal{L}_{a \to a+h}^{c\text{-}DTM}(\theta) = \int_0^1 \frac{\dot\alpha(t)}{1 - \alpha(t)}\, \mathbb{E}_a\Big[ \sum_{i \in [L],\, x_t^i = \mathbf{m}} e^{hr(x_1)} \cdot \mathrm{CE}\big(T_c(\cdot \mid x_t, i, x_1)\|\pi_\theta(\cdot \mid x_t, i)\big) \Big]\, dt. \tag{16}$$

*is the ground-truth tilted unmasking posterior* $\pi_{a+h}$*, i.e.*

$$\pi_{\theta^*}(\cdot \mid x_t, i) = \pi_{a+h}(\cdot \mid x_t, i).$$

*Proof.* We work in the nonparametric setting (equivalently assume sufficient network capacity) where we may choose an arbitrary measurable family $\pi(\cdot \mid x_t, i) \in \Delta(\mathcal{V})$ for each $(t, x_t, i)$ with $x_t^i = \mathbf{m}$.

**Step 1: Reduce the random target to a deterministic conditional objective.** Fix any $(t, x_t, i)$ with $x_t^i = \mathbf{m}$. For any candidate conditional distribution $\pi(\cdot \mid x_t, i)$, define the conditional loss given $x_t$:

$$\mathcal{L}_{t,i}(\pi; x_t) := \mathbb{E}_a\left[e^{hr(x_1)} \, \mathrm{CE}\big(T_c(\cdot \mid x_t, i, x_1) \,\|\, \pi(\cdot \mid x_t, i)\big) \mid x_t\right]. \tag{23}$$

Using the definition $\mathrm{CE}(p\|q) = -\sum_{v \in V} p(v) \log q(v)$ and pulling $\log \pi(\cdot \mid x_t, i)$ outside the conditional expectation,

$$\mathcal{L}_{t,i}(\pi; x_t) = -\sum_{v \in \mathcal{V}} \mathbb{E}_a\left[e^{hr(x_1)} \, T_c(v \mid x_t, i, x_1) \mid x_t\right] \log \pi(v \mid x_t, i).$$

Plugging in the defining unbiasedness property (15), we get

$$\mathcal{L}_{t,i}(\pi; x_t) = -\sum_{v \in \mathcal{V}} \mathbb{E}_a\left[e^{hr(x_1)} \, \mathbf{1}\{x_1^i = v\} \mid x_t\right] \log \pi(v \mid x_t, i)$$

$$= \mathbb{E}_a[e^{hr(x_1)} \mid x_t] \, \mathrm{CE}\big(\pi_{a+h}(\cdot \mid x_t, i) \,\|\, \pi(\cdot \mid x_t, i)\big). \tag{24}$$

where the last line uses the conditional Esscher identity (13), i.e.

$$\pi_{a+h}(v \mid x_t, i) = \mathbb{E}_{a+h}[\mathbf{1}\{x_1^i = v\} \mid x_t] = \frac{\mathbb{E}_a[e^{hr(x_1)}\mathbf{1}\{x_1^i = v\} \mid x_t]}{\mathbb{E}_a[e^{hr(x_1)} \mid x_t]}.$$

**Step 2: Pointwise minimization and uniqueness.** Fix $(t, x_t, i)$. Because the factor $\mathbb{E}_a[e^{hr(x_1)} \mid x_t]$ in (24) is positive and $\mathrm{CE}(p\|q)$ is strictly convex in $q$ over the simplex whenever $p$ is not degenerate, the unique minimizer of (23) over $\pi(\cdot \mid x_t, i) \in \Delta(\mathcal{V})$ is

$$\pi^\star(\cdot \mid x_t, i) = \pi_{a+h}(\cdot \mid x_t, i).$$

**Step 3: Lift pointwise optimality to the global $c$-DTM objective.** The full objective (16) is an integral over $t$ of the pointwise loss $\mathcal{L}_{t,i}(\pi; x_t)$ multiplied by the nonnegative weight $\dot\alpha(t)/(1 - \alpha(t))$. Since all prefactors are nonnegative and the dependence on $\pi$ separates across $(t, x_t, i)$, the unique global minimizer is obtained by choosing the unique pointwise minimizer for every $(t, x_t, i)$ with $x_t^i = \mathbf{m}$, namely $\pi^\star(\cdot \mid x_t, i) = \pi_{a+h}(\cdot \mid x_t, i)$. $\qquad\square$

**Proposition 3.3.** *For sufficiently small $h > 0$,*

$$\mathrm{Var}[\nabla\mathcal{L}_{a\to a+h}^{\textit{0-DTM}}(\theta)] \geq \mathrm{Var}[\nabla\mathcal{L}_{a\to a+h}^{\textit{1-DTM}}(\theta)]. \tag{18}$$

*Proof.* Fix a time $t \in [0, 1]$, a partially masked state $x_t$ and a masked position $i$ with $x_t^i = \mathbf{m}$. Write $w := e^{hr(x_1)}$ and define the score-vector

$$\phi(v) := \nabla_\theta \log \pi_\theta(v \mid x_t, i) \in \mathbb{R}^d, \qquad \mu(x_t, i) := \sum_{v \in \mathcal{V}} \pi_a(v \mid x_t, i) \, \phi(v) = \mathbb{E}_a[\phi(x_1^i) \mid x_t, i].$$

Consider the per-sample (single $(t, i)$) contribution to the weighted DTM objective

$$\ell_c(\theta) := w \cdot \mathrm{CE}\big(T_c(\cdot \mid x_t, i, x_1) \,\|\, \pi_\theta(\cdot \mid x_t, i)\big) = -w \sum_{v \in \mathcal{V}} T_c(v \mid x_t, i, x_1) \log \pi_\theta(v \mid x_t, i).$$

Since $\nabla_\theta \mathrm{CE}(q\|p_\theta) = -\sum_v q(v) \nabla_\theta \log p_\theta(v)$, we have

$$\nabla_\theta \ell_c(\theta) = -w \sum_{v \in \mathcal{V}} T_c(v \mid x_t, i, x_1) \, \phi(v).$$

For our concrete $T_c$ in (17),

$$T_c(\cdot \mid x_t, i, x_1) = c\,\pi_a(\cdot \mid x_t, i) + \frac{w - c}{w} \mathbf{1}\{\cdot = x_1^i\},$$

hence

$$\nabla_\theta \ell_c(\theta) = -w\Big(c \sum_v \pi_a(v \mid x_t, i)\phi(v) + \frac{w-c}{w}\phi(x_1^i)\Big)$$

$$= -\Big(c\,w\,\mu(x_t, i) + (w-c)\,\phi(x_1^i)\Big).$$

Therefore (ignoring the overall minus sign, which does not affect variance), the single-sample gradient estimators for $c = 0$ and $c = 1$ are

$$G_0 := w\,\phi(x_1^i), \qquad G_1 := w\,\mu(x_t, i) + (w-1)\,\phi(x_1^i).$$

We compare $\mathrm{Var}(G_0)$ and $\mathrm{Var}(G_1)$ under the joint sampling of $(x_1, x_t)$ from $\mathbb{P}_a$. For small $h > 0$, using the expansion $w = e^{hr(x_1)} = 1 + hr(x_1) + O(h^2)$, we obtain

$$G_0 = \phi(x_1^i) + hr(x_1)\phi(x_1^i) + O(h^2), \qquad G_1 = \mu(x_t, i) + hr(x_1)\big(\mu(x_t, i) + \phi(x_1^i)\big) + O(h^2),$$

where the $O(h^2)$ terms are in $L^2$ provided $r$ and $\phi$ have finite second moments.

At $h = 0$, we have $G_0|_{h=0} = \phi(x_1^i)$ and $G_1|_{h=0} = \mu(x_t, i) = \mathbb{E}_a[\phi(x_1^i) \mid x_t, i]$. By the law of total variance,

$$\mathrm{Var}(\phi(x_1^i)) = \mathrm{Var}\big(\mathbb{E}_a[\phi(x_1^i) \mid x_t, i]\big) + \mathbb{E}\big[\mathrm{Var}(\phi(x_1^i) \mid x_t, i)\big] > \mathrm{Var}(\mu(x_t, i)),$$

so $\mathrm{Var}(G_0) > \mathrm{Var}(G_1)$ holds at $h = 0$. Since $G_0$ and $G_1$ depend smoothly on $h$ through $w = e^{hr(x_1)}$ and both admit $L^2$ Taylor expansions around $h = 0$, their variances are continuous in $h$ near 0. Therefore, the inequality $\mathrm{Var}(G_0) > \mathrm{Var}(G_1)$ persists for all sufficiently small $h > 0$. Averaging over the random choice of $(t, i)$ and integrating against the schedule weight (as in the full DTM loss) preserves the inequality, completing the proof.

This argument mirrors the control-variate variance comparison in Tilt Matching (Potaptchik et al., 2025) (Proposition 7 and its proof), adapted here to the discrete DTM cross-entropy gradient estimator. □

In order to prove Proposition 3.4, we first introduce the following lemma that expresses the KL divergence between two CTMC path measures.

**Lemma B.1** (CTMC path–KL for masked diffusion rates). *Let $P$ and $\widehat{P}$ be path measures on a continuous-time Markov chain $\{X_t\}_{t\in[0,1]}$ with the same initial distribution and (possibly time-inhomogeneous) rate matrices $R_t$ and $\widehat{R}_t$. Assume $\widehat{R}_t$ is absolutely continuous with respect to $R_t$, i.e. $R_t(x, y) > 0 \Rightarrow \widehat{R}_t(x, y) > 0$. Then the path-space KL admits the standard expansion*

$$\mathrm{KL}(P\|\widehat{P}) = \mathbb{E}_P\Big[\int_0^1 \big(R_t(X_t, X_t) - \widehat{R}_t(X_t, X_t)\big)\,dt + \sum_{t:\,X_t \neq X_{t^-}} \log\frac{R_t(X_{t^-}, X_t)}{\widehat{R}_t(X_{t^-}, X_t)}\Big]. \tag{25}$$

*Moreover, suppose $R_t$ and $\widehat{R}_t$ are* masked diffusion *rates of the form*

$$R_t(x, x[x^i \leftarrow v]) = \lambda(t)\,\mathbf{1}\{x^i = \mathbf{m}\}\,\pi(v \mid x, i), \qquad \widehat{R}_t(x, x[x^i \leftarrow v]) = \lambda(t)\,\mathbf{1}\{x^i = \mathbf{m}\}\,\widehat{\pi}(v \mid x, i),$$

*for $\lambda(t) = \frac{\dot{\alpha}(t)}{1-\alpha(t)}$ and some posteriors $\pi(\cdot \mid x, i), \widehat{\pi}(\cdot \mid x, i)$ on $\mathcal{V}$, where $x[x^i \leftarrow v]$ denotes the partially masked sequence $x$ with $i$-th token replaced by $v$. Then*

$$\mathrm{KL}(P\|\widehat{P}) = \int_0^1 \lambda(t)\,\mathbb{E}_P\Big[\sum_{i:\,X_t^i = \mathbf{m}} \mathrm{KL}\big(\pi(\cdot \mid X_t, i)\,\|\,\widehat{\pi}(\cdot \mid X_t, i)\big)\Big]dt. \tag{26}$$

*Proof.* Equation (25) is the standard Radon–Nikodym / Girsanov formula for CTMCs. See Kim et al. (2025a) Proposition 5, Shaul et al. (2025) Appendix D. See also Campbell et al. (2024) Appendix C.1 for a more comprehensive treatment.

We now specialize (25) to the masked diffusion rates. For any state $x$, the total exit rate under $R_t$ is

$$\sum_{y \neq x} R_t(x, y) = \sum_{i:\,x^i = \mathbf{m}} \sum_{v \in \mathcal{V}} \lambda(t)\pi(v \mid x, i) = \lambda(t) \cdot \#\{i : x^i = \mathbf{m}\}.$$

The same holds for $\widehat{R}_t$ because $\sum_v \widehat{\pi}(v \mid x, i) = 1$:

$$\sum_{y \neq x} \widehat{R}_t(x, y) = \lambda(t) \cdot \#\{i : x^i = \mathbf{m}\}.$$

Hence the diagonal terms coincide:

$$R_t(x, x) = -\sum_{y \neq x} R_t(x, y) = -\lambda(t)\#\{i : x^i = \mathbf{m}\} = \widehat{R}_t(x, x),$$

so the integral term in (25) vanishes identically.

Next, any jump $X_{t-} \to X_t$ must be of the form $X_t = X_{t-}[X_{t-}^i \leftarrow v]$ for a unique masked coordinate $i$. For such a jump,

$$\log \frac{R_t(X_{t-}, X_t)}{\widehat{R}_t(X_{t-}, X_t)} = \log \frac{\lambda(t)\pi(v \mid X_{t-}, i)}{\lambda(t)\widehat{\pi}(v \mid X_{t-}, i)} = \log \frac{\pi(v \mid X_{t-}, i)}{\widehat{\pi}(v \mid X_{t-}, i)}.$$

Plugging this into (25), conditioning on $X_t$ (equivalently $X_{t-}$ a.s. for CTMCs), and using that under $P$ the revealed token at a masked position has law $\pi(\cdot \mid X_t, i)$, the expected jump contribution at time $t$ becomes $\lambda(t) \sum_{i: X_t^i = \mathbf{m}} \mathrm{KL}(\pi(\cdot \mid X_t, i)\|\widehat{\pi}(\cdot \mid X_t, i))$. Integrating over $t$ yields (26). $\qquad\square$

**Proposition 3.4.** *Let $\rho_\theta$ be the terminal marginal distribution on $\mathcal{V}^L$ induced by the parametric model's unmasking posterior $\pi_\theta$. Then*

$$\mathrm{KL}(\rho_{1,a+h}\|\rho_\theta) \leq \frac{1}{Z}\Big(\mathcal{L}_{a \to a+h}^{c\text{-}DTM}(\theta) - \mathcal{L}_{a \to a+h}^{c\text{-}DTM}(\theta^*)\Big), \tag{19}$$

*where $Z = \mathbb{E}_a[e^{hr(x_1)}]$.*

(Back to Proposition 3.4 in the main text.)

*Proof.* As shown in the main text, the $c$-DTM objective employs a tractable random target as an unbiased estimate of the intractable $\pi_{a+h}$. Thus for theoretical analysis on $\rho_{1,a+h}$, it would be helpful to reduce to the ideal loss.

**Step 1: Replace the random target.** First, we reduce the random target $T_c$ to the true posterior $\pi_{a+h}$. By (24), the $c$-DTM objective (16) becomes

$$\mathcal{L}_{a \to a+h}^{c\text{-}DTM}(\theta) = \int_0^1 \frac{\dot{\alpha}(t)}{1 - \alpha(t)} \mathbb{E}_a\Big[\sum_{i: x_t^i = \mathbf{m}} \mathbb{E}_a\Big[e^{hr(x_1)} \mid x_t\Big] \mathrm{CE}\big(\pi_{a+h}(\cdot \mid x_t, i) \mid \pi_\theta(\cdot \mid x_t, i)\big)\Big]dt. \tag{27}$$

**Step 2: Change measure from $\mathbb{P}_a$ to $\mathbb{P}_{a+h}$.** Define the "ideal" (but still trajectory-based) cross-entropy functional

$$\widetilde{L}_{a+h}(\theta) := \int_0^1 \frac{\dot{\alpha}(t)}{1 - \alpha(t)} \mathbb{E}_{a+h}\Big[\sum_{i: x_t^i = \mathbf{m}} \mathrm{CE}\big(\pi_{a+h}(\cdot \mid x_t, i) \,\big\|\, \pi_\theta(\cdot \mid x_t, i)\big)\Big]dt.$$

Applying the change-of-measure (12) with

$$\varphi = \int_0^1 \frac{\dot{\alpha}(t)}{1 - \alpha(t)} \sum_{i: x_t^i = \mathbf{m}} \mathrm{CE}(\pi_{a+h}(\cdot \mid x_t, i)\|\pi_\theta(\cdot \mid x_t, i)) \, dt$$

and comparing with (27), we obtain the relation

$$\mathcal{L}_{a \to a+h}^{c\text{-}DTM}(\theta) = Z \cdot \widetilde{L}_{a+h}(\theta). \tag{28}$$

In particular, if $\theta^\star$ is the optimizer such that $\pi_{\theta^\star} = \pi_{a+h}$, then $\widetilde{L}_{a+h}(\theta^\star) = \min_\theta \widetilde{L}_{a+h}(\theta)$ and

$$\mathcal{L}_{a \to a+h}^{c\text{-}DTM}(\theta) - \mathcal{L}_{a \to a+h}^{c\text{-}DTM}(\theta^\star) = Z\big(\widetilde{L}_{a+h}(\theta) - \widetilde{L}_{a+h}(\theta^\star)\big).$$

**Step 3: Excess cross-entropy equals a path KL, which upper-bounds the terminal KL.** Let $\mathbb{P}_{a+h}$ denote the *true* path measure under tilt $a + h$, and let $\mathbb{P}_\theta$ denote the path measure induced by running the CTMC with the same interpolant but using the learned posterior $\pi_\theta$. It remains to express the difference $\widetilde{L}_{a+h}(\theta) - \widetilde{L}_{a+h}(\theta^\star)$ in terms of the KL divergence between the two path measures.

Recall from Section 2.3 that the discrete stochastic interpolant, or intuitively the unmasking-event mechanism, is itself independent of $\theta$ or the tilt. So the likelihood ratio between $\mathbb{P}_{a+h}$ and $\mathbb{P}_\theta$ depends on $\theta$ *only* through the product of the local choice probabilities $\pi_\theta(\cdot \mid x_t, i)$ at unmasking events. Consequently, by Lemma B.1 with $\pi = \pi_{a+h}$ and $\widehat{\pi} = \pi_\theta$, expanding the KL divergence between path measures gives

$$\text{KL}(\mathbb{P}_{a+h}\|\mathbb{P}_\theta) = \int_0^1 \frac{\dot{\alpha}(t)}{1 - \alpha(t)} \, \mathbb{E}_{a+h}\Big[ \sum_{i:\, X_t^i = \mathbf{m}} \text{KL}\big(\pi_{a+h}(\cdot \mid X_t, i) \,\|\, \pi_\theta(\cdot \mid X_t, i)\big)\Big] dt. \tag{29}$$

On the other hand,

$$\text{CE}(\pi_{a+h}\|\pi_\theta) - \text{CE}(\pi_{a+h}\|\pi_{a+h}) = \text{KL}(\pi_{a+h}\|\pi_\theta),$$

so by definition of $\widetilde{L}_{a+h}$ and (29),

$$\widetilde{L}_{a+h}(\theta) - \widetilde{L}_{a+h}(\theta^\star) = \text{KL}(\mathbb{P}_{a+h}\|\mathbb{P}_\theta). \tag{30}$$

Finally, the terminal marginal $x_1$ is a measurable function of the full trajectory, so by the data processing inequality,

$$\text{KL}(\rho_{1,a+h}\|\rho_\theta) \le \text{KL}(\mathbb{P}_{a+h}\|\mathbb{P}_\theta). \tag{31}$$

Combining (28), (30), and (31) yields

$$\text{KL}(\rho_{1,a+h}\|\rho_\theta) \le \widetilde{L}_{a+h}(\theta) - \widetilde{L}_{a+h}(\theta^\star) = \frac{1}{Z}\Big(L_{a\to a+h}^{\text{c-DTM}}(\theta) - L_{a\to a+h}^{\text{c-DTM}}(\theta^\star)\Big),$$

which is exactly (19). □

# C. DTM with semi-autoregressive (SAR) decoding

We formalize how to align DTM training with semi-autoregressive (SAR) / block-diffusion decoding. The main text notes that DTM is minimized pointwise in $(t, x_t, i)$ and therefore adapts to alternative decoding constraints by (i) changing the distribution of conditioning states $x_t$ to those reachable under SAR and (ii) restricting eligible update indices to the currently active block. Here we specify a SAR-compatible stochastic interpolant, derive the corresponding hazard/time-weighting factor, and state the resulting SAR-aligned $c$-DTM objective.

## C.1. SAR Stochastic Interpolant

Fix a sequence length $L$ and block size $B$, and let $M := L/B$ (assume $B|L$ for simplicity). Define blocks $\mathcal{J}_b := \{bB + 1, \ldots, (b+1)B\}$ for $b \in \{0, \ldots, M-1\}$. We partition global time $t \in [0, 1]$ into $M$ block-intervals by defining the active block index

$$b(t) := \lfloor Mt \rfloor \in \{0, \ldots, M-1\}, \qquad u(t) := Mt - b(t) \in [0, 1),$$

where $u(t)$ is the rescaled (within-block) time.

Let $\alpha : [0, 1] \to [0, 1]$ be a smooth increasing schedule with $\alpha(0) = 0$ and $\alpha(1) = 1$. Given a clean terminal sequence $x_1 \sim \rho_{1,a}$, sample i.i.d. reveal times $\{T^i\}_{i \in [L]}$ with CDF $\mathbb{P}[T^i < t] = \alpha(t)$. Define the SAR-masked state $x_{t,a}^{\text{SAR}}$ coordinate-wise by

$$x_{t,a}^{\text{SAR},i} := \begin{cases} x_1^i, & i \in \bigcup_{b' < b(t)} \mathcal{J}_{b'} \quad \text{(past blocks fully revealed)}, \\ x_1^i, & i \in \mathcal{J}_{b(t)} \text{ and } u(t) \ge T^i \quad \text{(revealed within current block)}, \\ \mathbf{m}, & i \in \mathcal{J}_{b(t)} \text{ and } u(t) < T^i \quad \text{(still masked within current block)}, \\ \mathbf{m}, & i \in \bigcup_{b' > b(t)} \mathcal{J}_{b'} \quad \text{(future blocks fully masked)}. \end{cases} \tag{32}$$

Let $\rho_{t,a}^{\text{SAR}} := \text{Law}(x_{t,a}^{\text{SAR}})$ denote the induced marginal distribution.

**Eligible indices.** Under (32), only masked positions in the *current* block are eligible for an update at time $t$:

$$\mathcal{I}(t, x_t) := \{\, i \in \mathcal{J}_{b(t)} : x_t^i = \mathbf{m} \,\}.$$

## C.2. Hazard Factor under the SAR Interpolant

For a token in the active block, the event "still masked at local time $u$" has probability $1 - \alpha(u)$, and the conditional reveal probability in $[u, u + du]$ is the hazard

$$\lambda(u)\, du \quad \text{with} \quad \lambda(u) = \frac{\alpha'(u)}{1 - \alpha(u)}.$$

Because $u(t) = Mt - b(t)$, we have $du = M\, dt$ on each block interval, hence the global-time hazard is

$$\lambda_{\text{SAR}}(t) = M \cdot \frac{\alpha'(u(t))}{1 - \alpha(u(t))}. \tag{33}$$

This is the analogue of the factor $\alpha'(t)/(1 - \alpha(t))$ that appears in Proposition 3.4 for the any-order interpolant, with an additional time-change due to the blockwise parametrization.

**Simplified choice.** If desired, one may adopt $\alpha(u) = u$ (uniform reveal times within each block), in which case $\lambda_{\text{SAR}}(t) = \frac{M}{1 - u(t)}$ and the weighting is fully explicit.

## C.3. SAR-aligned $c$-DTM Objective

Let $\mathbb{P}_a^{\text{SAR}}$ denote the path measure induced by the SAR interpolant (32) (with terminal $x_1 \sim \rho_{1,a}$), and write $\mathbb{E}_a^{\text{SAR}}[\cdot]$ for expectation under this measure. Define the (tilted) unmasking posterior under SAR as

$$\pi_a^{\text{SAR}}(v \mid x_t, i) := \mathbb{P}_a^{\text{SAR}}[x_1^i = v \mid x_t], \qquad v \in \mathcal{V}.$$

Then the SAR-aligned $c$-DTM objective is

$$\mathcal{L}_{c\text{-DTM}}^{\text{SAR}}(\theta) = \int_0^1 \lambda_{\text{SAR}}(t)\, \mathbb{E}_a^{\text{SAR}}\Big[ \sum_{i \in \mathcal{I}(t, x_t)} e^{hr(x_1)}\, \text{CE}\big(T_c(\cdot \mid x_t, i, x_1) \,\big\|\, \pi_\theta(\cdot \mid x_t, i)\big)\Big] dt, \tag{34}$$

where $\mathcal{I}(t, x_t) = \{i \in \mathcal{J}_{b(t)} : x_t^i = \mathbf{m}\}$ and $T_c$ is defined as in (17).

## C.4. Why SAR Preserves Exactness

The Esscher change-of-measure arguments in Section 3 depend only on (i) the terminal tilt $\rho_{1,a+h}(x) \propto \rho_{1,a}(x) e^{hr(x)}$ and (ii) the fact that the conditional law of $x_t$ given $x_1$ is independent of the tilt parameter $a$. Both properties remain true for the SAR interpolant (32), since the masking schedule and block structure do not depend on $a$. Therefore Proposition 3.1 (conditional tilting) and Proposition 3.2 (pointwise minimizer of $c$-DTM) carry over verbatim with $\mathbb{P}_a^{\text{SAR}}$ in place of $\mathbb{P}_a$. Finally, Proposition 3.4's KL control argument extends by replacing the any-order hazard $\alpha'(t)/(1 - \alpha(t))$ with the SAR hazard (33), yielding the weighting in (34).

# D. Experimental Details

## D.1. Datasets and Reward Functions

We summarize the datasets and reward shaping used for DTM fine-tuning.

**Maze Planning** We fix a randomly generated 41-by-41 maze with door fraction 0.4 (see Fig. 7), and train an 8.9M base model on it using the standard ELBO-based variational loss (5). For two cells $z = (x, y)$ and $z' = (x', y')$ for some integers $x, x', y, y' \in [0, 40]$, we define the Manhattan distance

$$d(z, z') := |x - x'| + |y - y'|,$$

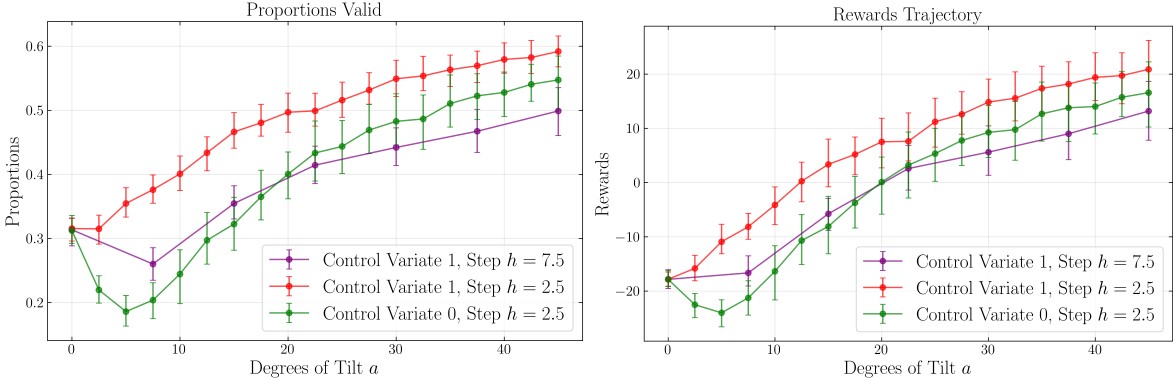

*(a)* Proportions of valid paths against the degrees of tilt       *(b)* Mean reward of paths against degrees of tilt

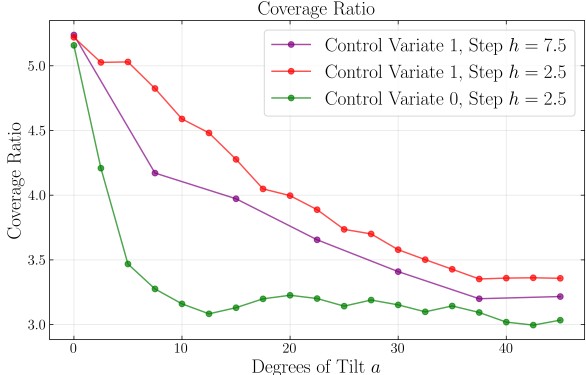

*(c)* Diversity of paths measured by coverage ratio, which is the ratio between the total number of cells that paths cover and number of cells the longest path covers.

*Figure 5.* Proportion of valid paths ($a$) mean rewards ($b$), and diversity of paths ($c$) against degrees of tilt $a$. Our model is trained on three sets of control variate $c$ and annealing steps $h$. Small step size and control variate 1 has higher path diversity, validity and rewards.

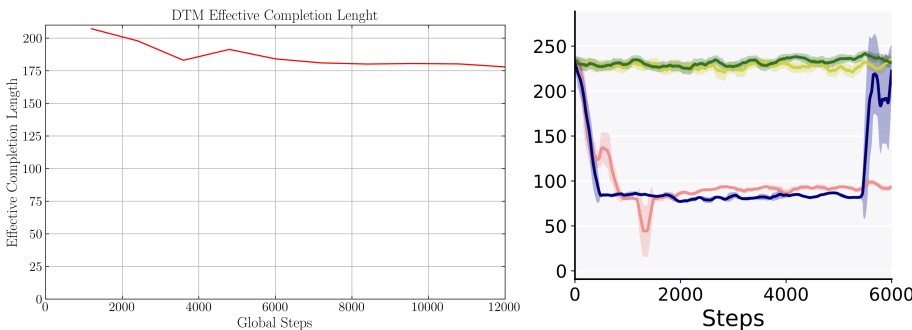

*(a)* Effective generation length of DTM on Countdown       *(b)* d1 (light green), WD1 (red), Uni-GRPO (dark green), SPG (blue) (from Fig. 9 in Wang et al. (2025a))

*Figure 6.* Effective generation length of DTM versus RL baselines. With direct training on state-level posteriors, DTM achieves stable reasoning.

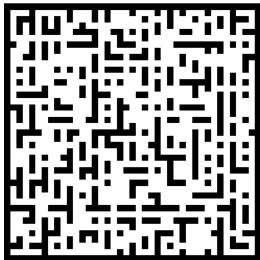

*Figure 7.* The fixed 41-by-41 maze with door fraction 0.4.

i.e. the $L^1$-distance. For each completion of the form $z = (\texttt{s}, \texttt{g}, \texttt{SEP}, z_1, z_2, ..., z_n, \texttt{PAD}, ..., \texttt{PAD})$, we say the path is valid if $z_1 = \texttt{s}$, $z_n = \texttt{g}$, all $z_i$'s are non-wall cells in the maze, and all consecutive cells $(z_i, z_{i+1})$ are direct neighbors in the maze (i.e. with Manhattan distance 1). The base model produces valid paths with probability 0.313 under sampling temperature 1.0.

We define the stay-away-from-center reward by

$$r_{\text{stay-away-from-center}}(z) = \begin{cases} \frac{1}{20} \min_{i \in [n]} d(z_i, \texttt{center}), & \text{if } z \text{ is valid} \\ -0.4, & \text{otherwise} \end{cases}$$

where $\texttt{center} = (20, 20)$ is the center of the maze, and we divide by 20 to signify the distance relative to the maximum attainable distance from the center.

**MATH500** We use the pre-defined train split of the MATH500 dataset[3] for rollouts during training, and evaluate model performance on the test split. Same as in the baselines, the prompts we used are the same during training and evaluation, as shown in Figure 8.

For the reward functions, we follow SPG but slightly adjust the scale, combining a correctness term with a piecewise format score:

- **Correctness reward:** $+1.0$ if the expression inside \boxed{...} matches the correct answer; $-1.0$ otherwise.

- **Format reward:** assign

$$r_{\text{fmt}} = \begin{cases} 0.375, & \text{if <answer>...</answer> is present and contains \boxed,} \\ 0.25, & \text{if answer tags are present but no \boxed,} \\ 0.125, & \text{if no answer tags but \boxed is present,} \\ 0.0, & \text{if neither answer tags nor \boxed is present.} \end{cases}$$

The training reward is just the sum of the correctness and format rewards.

**Countdown** We follow all the baselines exactly: for training rollouts, we use the train split of the Countdown dataset[4] and restrict to instances that use only three numbers; for evaluation, we evaluate on 256 synthetically generated countdown questions with 3 numbers. The same prompts are used for training and evaluation, as shown in Figure 9.

The reward consists of a soft correctness score based on whether the produced expression (i) uses exactly the provided numbers and (ii) reaches the target:

$$r = \begin{cases} 1.0, & \text{if the expression hits the target and uses only the provided numbers,} \\ 0.1, & \text{if it uses the correct numbers but does not reach the target,} \\ 0.0, & \text{otherwise.} \end{cases}$$

---

[3] https://huggingface.co/datasets/ankner/math-500
[4] https://huggingface.co/datasets/Jiayi-Pan/Countdown-Tasks-3to4

```
You are a math expert.  You will be given a question to solve.  Solve it step by step.
Wrap the final answer in a \boxed{}.
Respond in the following format:
<reasoning>
Your reasoning here
</reasoning>
<answer>
\boxed{...}
</answer>

{Actual problem statement inserted here verbatim}
```

*Figure 8.* Prompt used for MATH500 and GSM8K. The problem statement is appended directly after the template.

```
Respond in the following format:
<reasoning>
...
</reasoning>
<answer>
...
</answer>

Using only the numbers [38, 92, 52] create an arithmetic expression that evaluates
to exactly 78.  You must use all numbers from the list, and each number must be used
exactly once.  You may use the operations +, -, *, and / as needed.  After reasoning,
provide only your final expression inside <answer></answer> tags without including
an equals sign or the target number.
For example, if the numbers are [2, 3, 4] and the target is 5, a valid answer is:
<answer>
2*4-3
</answer>
```

*Figure 9.* Prompt used for Countdown. In each instance, the list `[38, 92, 52]` is replaced by the provided numbers in the actual question and `78` is replaced by the actual target value.

**Sudoku** We experiment on the 4×4 Sudoku dataset[5]. We adopted SPG's modification on the original split to avoid train-test leakage: the dataset contains 1M puzzles spanning all 288 possible completed 4×4 solution grids. They randomly selected 200 solutions and included all puzzles whose ground-truth completion is one of these solutions in the training set (694,006 puzzles). For evaluation, they sampled 2-3 puzzles from each of the remaining 88 solutions, forming a 256-instance test set.

We use the same prompt format for training and evaluation as shown in Figure 10; we also follow SPG and use their 3-shot prompts for Sudoku (inserting three fixed puzzle/solution exemplars) because the base model performs poorly in the zero-shot setting. It is ensured that (i) the solutions shown in the 3-shot exemplars do not appear among the test-set solutions, and (ii) there is no puzzle overlap between train and test.

The reward is given by twice the fraction of correctly filled cells in each puzzle.

**GSM8K** We use the train split of the GSM8K dataset (Cobbe et al., 2021) with the same prompt as MATH500 as in Figure 8. Our reward is slightly shifted compared to SPG. We also removed the soft and strict format reward since they are mostly zero in the training of DTM and SPG. Our reward function consists of:

- **Correctness reward:** $+2.0$ if the expression inside `\boxed{...}` matches the correct answer.

- **XML structure reward:** $+0.125$ for correct formatting tag, with small penalties for contents after the closing tag.

- **Integer answer reward:** $-0.5$ penalty if the answer is not an integer.

---

[5]https://github.com/Black-Phoenix/4x4-Sudoku-Dataset

```
Please solve the following 4x4 Sudoku puzzle.  The puzzle is provided as a
16-character string reading left-to-right, top-to-bottom, where '0' represents empty
cells.

Rules:
- Fill empty cells with digits 1-4
- Each row/column must contain digits 1-4 exactly once
- Each 2x2 box must contain digits 1-4 exactly once

Important:  Your solution must be a COMPLETE 16-character string with only digits
1-4.

Respond in this exact format:
<reasoning>
Your step-by-step solving process
</reasoning>
<answer>
[16-character solution string, digits 1--4 only]
</answer>

{3-SHOT EXAMPLES INSERTED HERE}

Question:  Solve the following Sudoku puzzle:  {ACTUAL PUZZLE HERE}
Answer:
```

*Figure 10.* Sudoku prompt template. We use 3-shot prompting: three solved puzzle exemplars are inserted; the evaluation set uses disjoint underlying solutions from the exemplars. To avoid repetition, we refer to Appendix D.3 of Wang et al. (2025a) for the 3 exemplars.

## D.2. Hyperparameters and Implementation Details

Following the baselines, we employ LoRA with a rank of $r = 128$ and scaling factor $\alpha = 64$, 4-bit quantization (Dettmers et al., 2023), and use the AdamW optimizer(Loshchilov & Hutter, 2019), with $\beta_1 = 0.9$, $\beta_2 = 0.99$, and weight decay of 0.1. We conducted training on 8 NVIDIA H100-80G GPU, with the following hyperparameters:

For MATH500, we trained with a per-gpu buffer of 60 distinct prompts with 8 completions per prompt, and refresh a quarter every 50 steps. We use an annealing step size $h = 1$, performing 400 gradient update steps per $h$ phase with per-gpu batch size 72. We use a learning rate of $1 \times 10^{-5}$, with gradient clip at 1.0.

For Sudoku, we trained with a per-gpu buffer of 80 distinct prompts with 1 completion per prompt, and refresh a quarter every 32 steps. We use an annealing step size $h = 5$, performing 256 gradient update steps per $h$ phase with per-gpu batch size 64. We use a learning rate of $3 \times 10^{-6}$, with gradient clip at 2.0.

For Countdown, we trained with a per-gpu buffer of 80 distinct prompts with 6 completions per prompt, and refresh a quarter every 25 steps. We use an annealing step size $h = 6$, performing 800 gradient update steps per $h$ phase with per-gpu batch size 96. We use a learning rate of $4 \times 10^{-6}$, with gradient clip at 2.0.

For GSM8K, we trained with a per-gpu buffer of 80 distinct prompts with 6 completions per prompt, and refresh a quarter every 50 steps. We use an annealing step size $h = 1$, performing 800 gradient update steps per $h$ phase with per-gpu batch size 96. We use a learning rate of $5 \times 10^{-5}$, with gradient clip at 2.0.

## D.3. Ablation on using SAR-aligned objective

We perform an ablation on Countdown comparing training with the default random interpolant as in (9) and (16) against SAR-aligned DTM as in (32) and (34), while fixing all other hyperparameters. As shown in Figure 11, SAR-aligned DTM reaches $81\%$ accuracy versus $67\%$ for the random-interpolant version. This shows that if training optimizes over arbitrary masked states that never arise at inference, compute is wasted on irrelevant trajectories. By aligning DTM with the SAR interpolant used at test time, we concentrate training on the states that actually matter for inference while preserving the same theoretical target. Such flexibility of being aligned with SAR sampling is a practical advantage of the DTM framework.

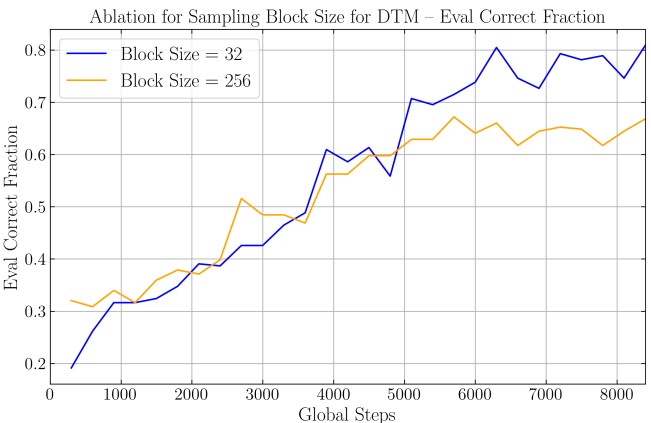

*Figure 11.* Comparison of performance on Countdown for DTM with random interpolant versus SAR-aligned interpolant.

