# Tilt Matching for Scalable Sampling and Fine-Tuning

## Abstract

We propose a simple, scalable algorithm based on stochastic interpolants for sampling from unnormalized densities and for fine-tuning generative models. The approach, Tilt Matching, arises from a dynamical equation relating the flow matching velocity to one targeting the same distribution tilted by a reward, implicitly solving a stochastic optimal control problem. The resulting velocity inherits the regularity of stochastic interpolant transports while minimizing an objective with strictly lower variance than flow matching itself. The update to the velocity field can be interpreted as the sum of all joint cumulants between the interpolant velocity and the reward, and to first order is their covariance. The method requires neither reward gradients nor backpropagation through trajectories of the flow or diffusion. We empirically demonstrate that the approach is efficient and highly scalable, providing state-of-the-art results on sampling under Lennard-Jones systems and competitive performance for fine-tuning Stable Diffusion, without requiring reward multipliers. The framework also applies directly to tilting few-step flow map models.

Base    Tilt Matching    Base    Tilt Matching

*Prompt: Portrait of cute dog in renaissance style clothing.*

*Prompt: The Eiffel Tower in the middle of Times Square.*

*Prompt: Bald eagles playing poker.*

*Prompt: A fantasy landscape with a castle on a hill and a river flowing through a forest with fall leaves.*

*Figure 1.* For each prompt, we display paired samples from the Base model vs our Tilt Matching method.

## 1. Introduction

Generative models built out of dynamical transport like flow and diffusion models are highly scalable tools that serve as building blocks for foundation models across industries (Rombach et al., 2022; Geffner et al., 2025; Watson et al., 2023; Brooks et al., 2024; Zeni et al., 2025). These models work by building a continuous time map connecting a base distribution to a target distribution, realized by solving a differential equation whose coefficients are outputs of neural networks.

[1]Anonymous Institution, Anonymous City, Anonymous Region, Anonymous Country. Correspondence to: Anonymous Author <anon.email@domain.com>.

Preliminary work. Under review by the International Conference on Machine Learning (ICML). Do not distribute.

There is now a vested interest in applying them in settings where there is not *a priori* an abundance of data to learn from to complete a task of interest. These include learning to sample under Boltzmann distributions appearing in molecular dynamics (Noé et al., 2019; Herron et al., 2024; Plainer et al., 2025) and statistical physics (Albergo et al., 2019; Gabrié et al., 2022; Kanwar et al., 2020; Nicoli et al., 2021), as well as fine-tuning an existing generative model so as to produce samples that align with user requests.

Both of these problems can be framed as ***tilting*** some existing distribution toward a new target. For Boltzmann sampling, this means adapting the energy function defining the theory; for fine-tuning, this means adapting the base generative model to score highly against a reward $r(x)$. Our aims in this paper are precisely centered around this picture.

Given access to samples from a distribution with density $\rho_1(x) : \mathbb{R}^d \to \mathbb{R}$, we want to learn to sample the tilted distribution $\rho_{1,a}(x) \propto \rho_1(x)e^{ar(x)}$, where $r(x)$ is a scalar function which defines the **tilt** and $a$ is an annealing parameter that characterizes the extent of the tilt. This initial distribution $\rho_1 = \rho_{1,a=0}$ could be given by an existing generative model, as in the case of fine-tuning, or it may be a reference distribution that is easy to sample with conventional techniques when performing sampling. We will ultimately be interested in $\rho_{1,a=1}$, i.e. the density fully tilted toward the reward.

While diffusions (Song et al., 2020; Ho et al., 2020) and flow-based models (Lipman et al., 2022; Albergo & Vanden-Eijnden, 2022; Liu et al., 2022) work well when data is available, regression objectives for the data-less contexts we focus on here are still not available, or come with caveats. In what follows, we begin by briefly recalling these highly scalable generative models. Then, we will motivate a practical *modification* to these approaches so that they maintain many of their appealing optimization qualities while making them applicable to fine-tuning and sampling. To this end, we specify our **main contributions:**

- **Evolution of flow matching velocity fields under reward tilts.** We derive an exact ordinary differential equation describing how stochastic interpolant velocity fields evolve with the annealing parameter $a$ under exponential reward tilting $\rho_{1,a} \propto \rho_1 e^{ar}$. The infinitesimal change is given by the conditional covariance between the interpolant dynamics and the reward.

- **Tilt Matching for learning reward-tilted transports.** We construct a family of simple iterative regression loss functions for the tilted velocity field that do not rely on backpropagating through generated trajectories, do not require spatial gradients of the reward, can avoid likelihood computations during training, and whose variances are strictly less than those of flow matching.

- **Connection to stochastic optimal control (SOC).** We show that the deterministic ODE drift learned by Tilt Matching is the *probability flow ODE* corresponding to a Doob $h$-transform diffusion obtained by tilting the path measure with the terminal weight $e^{ar(X_1)}$. As a result, Tilt Matching recovers the same controlled transport object as in SOC, but without introducing an SDE-based control formulation, relying instead on simple regression with scalar rewards.

- **Applications to sampling and fine-tuning.** We show how Tilt Matching can be applied to both sampling distributions known up to normalizing constant and to fine-tuning existing generative models, where we achieve state-of-the-art performance on sampling Lennard-Jones potentials with diffusion-based samplers, and

match or surpass the performance of diffusion fine-tuning methods that rely on stochastic simulations, validating the approach on Stable Diffusion 1.5 without reward scaling.

## 2. Preliminaries

### 2.1. Dynamical Transport and Stochastic Interpolants

Many state-of-the-art generative models that aim to model a data distribution $\rho_1(x)$ learned from samples $\{x_1\}_{i=1}^N$ do so by means of dynamically mapping samples from a reference distribution $x_0 \sim \rho_0$. This mapping is defined by a drift coefficient in a flow (Albergo & Vanden-Eijnden, 2022; Lipman et al., 2022) or diffusion process (Song et al., 2020; Ho et al., 2020), e.g. the transport for an ordinary differential equation (ODE) is defined by

$$\dot{x}_t = b_t(x_t), \qquad x_0 \sim \rho_0, \qquad (1)$$

where $b_t : [0,1] \times \mathbb{R}^d \to \mathbb{R}^d$ is a vector field that governs the transport such that the solution to (1) at time 1 produces a sample $x_1 \sim \rho_1$. The density $\rho_t$ of this process satisfies the continuity equation

$$\partial_t \rho_t + \nabla \cdot (b_t \rho_t) = 0, \qquad \rho_{t=0} = \rho_0. \qquad (2)$$

In generative modeling, our aims are given a prescribed path $(\rho_t)_{t \in [0,1]}$, to learn $b_t$ over neural networks such that the marginal law arising from (1) satisfies (2) with the given $\rho_t$. A unifying perspective for dynamical transport models is that of stochastic interpolants (Albergo & Vanden-Eijnden, 2022; Albergo et al., 2023). A stochastic interpolant $I_t(x_0, x_1) : [0,1] \times \mathbb{R}^d \times \mathbb{R}^d \to \mathbb{R}^d$ defined as

$$I_t := \alpha_t x_0 + \beta_t x_1, \qquad (x_0, x_1) \sim \rho_0(x_0)\rho_1(x_1), \quad (3)$$

where $\alpha_t, \beta_t$ are functions of time satisfying $\alpha_0 = \beta_1 = 1$ and $\alpha_1 = \beta_0 = 0$, is a stochastic process which specifies a path $\rho_t := \text{Law}(I_t)$. Importantly, a velocity field associated to this $\rho_t$ which solves (2) has a closed form which is given by $b_t(x) = \mathbb{E}[\dot{I}_t | I_t = x]$, where the expectation is taken over the coupling $(x_0, x_1) \sim \rho_0(x_0)\rho_1(x_1)$ conditional on $I_t = x$. Plugging this expression into a regression loss function to learn $b_t$ over neural networks gives, by the tower property of the conditional expectation,

$$b_t = \arg\min_{\hat{b}_t} \int_0^1 \mathbb{E} \left| \hat{b}_t(I_t) - \dot{I}_t \right|^2 dt. \qquad (4)$$

This procedure is the backbone of various large-scale generative models across various domains such as image and video generation (Esser et al., 2024) and protein design (Geffner et al., 2025). The main question to keep in mind going forward is: *how is the solution $b_t$ of one transport problem related to the solution of another?*

**Flow maps.** Iteratively integrating the ODE (1) is expensive since it requires calling a neural network at each step. Instead, a *flow map* $X_{s,t}(x) : [0,1]^2 \times \mathbb{R}^d \to \mathbb{R}^d$ acts as an arbitrary integrator of (1) and is defined as

$$X_{s,t}(x_s) = x_t \qquad \forall s, t \in [0,1], \tag{5}$$

which directly maps solutions of the ODE $(x_u)_{u \in [0,1]}$ at time $s$ to solutions at any time $t$ (Song et al., 2023; Boffi et al., 2024; Sabour et al., 2025). This reduces the cost to one or a few network evaluations, making generation significantly more efficient. We defer the discussion on how to train flow maps to (Boffi et al., 2024; 2025). We remark that our objectives defined below can also be applied to flow maps. For the sake of clarity, we limit experimental demonstrations just to velocity parameterizations.

### 2.2. Fine-tuning and Sampling as Tilting

One might ask how this $b_t$ could be modified to $b_{t,a}$ so that it solves the transport not for $\rho_1$, but rather the tilted distribution $\rho_{1,a} \propto \rho_1 e^{ar}$ which defines our fine-tuning or sampling problem. That is, how are the velocity fields $b_{t,a=0}$ and $b_{t,a>0}$ related, and is there a learning paradigm that would allow us to estimate $b_{t,a}$ when initially given access only to the ground truth velocity field $b_{t,0}$? This would allow us to ultimately evolve $b_{t,a}$ all the way to $b_{t,1}$, which would be the velocity field that can be used to directly sample the tilted distribution. We emphasize that throughout, $t$ indexes the generative transport time of the flow, while $a$ indexes annealing toward the reward-tilted target and does *not* correspond to physical or diffusion time. We now introduce our method focused on this evolution, which we call **Tilt Matching (TM)**, a scalable procedure for adapting velocity fields under tilting.

> **Core Problem.** Given a velocity field $b_t$ which generates samples from $\rho_1$, how can we modify it to produce a velocity field $b_{t,a}$ which generates samples from the tilted target distribution $\rho_{1,a} \propto \rho_1 e^{ar}$?

## 3. Deriving Tilt Matching

To approach this question, consider modifying (3) so that it instead uses samples from the tilted distribution $x_1^a \sim \rho_{1,a}$, and so we define the tilted interpolant by

$$I_t^a := \alpha_t x_0 + \beta_t x_1^a, \qquad (x_0, x_1^a) \sim \rho_0(x_0)\rho_{1,a}(x_1^a), \tag{6}$$

and define $\rho_{t,a} := \text{Law}(I_t^a)$. Learning the velocity field $b_{t,a}(x) = \mathbb{E}[\dot{I}_t^a | I_t^a = x]$ directly from this interpolant would be convenient, but we do not have samples a priori under $\rho_{1,a>0}$ to construct it, so this object is not immediately useful. However, it is possible to define $b_{t,a}$ in terms of the original interpolant, which we do have access to, combined with weights via the following proposition.

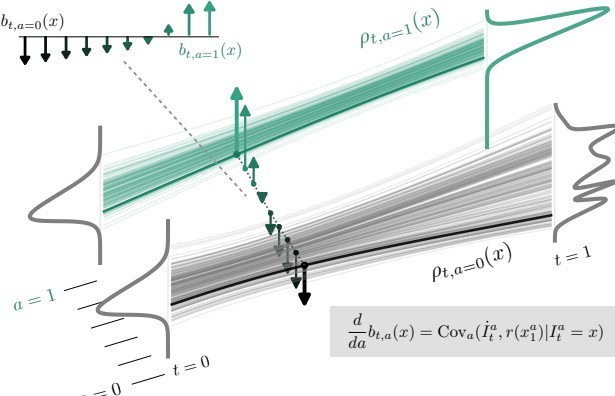

*Figure 2.* Schematic overview of the proposed method. When flow matching with a stochastic interpolant $I_t^{a=0}$ is used to learn a generative model $b_{t,a=0}$ that samples $\rho_{t,a=0}$ and in particular has terminal samples $\rho_{1,a=0}$ (gray curve), then the evolution of that velocity field in $a$ in order to sample $\rho_{1,a>0} = \frac{1}{Z_a}\rho_{1,0}e^{ar(x)}$, where $r$ is a reward function, has closed form given by the conditional covariance of the dynamics of the interpolant at $(t,a)$ and the reward. The velocity field, denoted as up or down arrows showing direction of motion in $x$, changes from negative to positive in the above toy example.

> **Proposition 3.1.** *(Esscher Transform.) Let $I_t^a = \alpha_t x_0 + \beta_t x_1^a$ be the interpolant constructed from samples $x_1^a \sim \rho_{1,a}$. Then the augmented velocity field $b_{t,a}(x)$ is given by:*
>
> $$b_{t,a}(x) = \frac{\mathbb{E}[\dot{I}_t^0 e^{ar(x_1)} | I_t^0 = x]}{\mathbb{E}[e^{ar(x_1)} | I_t^0 = x]}. \tag{7}$$
>
> *Furthermore, for any shift $h$ from $a$ to $a+h$, the updated velocity $b_{t,a+h}(x)$ satisfies:*
>
> $$b_{t,a+h}(x) = \frac{\mathbb{E}[\dot{I}_t^a e^{hr(x_1^a)} | I_t^a = x]}{\mathbb{E}[e^{hr(x_1^a)} | I_t^a = x]}. \tag{8}$$

Proposition 3.1 is proven in Appendix A and is related to the Esscher transform (Esscher, 1932), which characterizes how densities evolve under exponential tilts. In principle, (8) can be used to compute the final target velocity field $b_{t,1}(x)$ which is used to sample the fully tilted distribution. However, if $h$ is large, then the variance of this expression may make any computational realizations of it impractical. Instead, by taking the derivative of (8) with respect to $a$, we can ask how $b_{t,a}$ should evolve to anneal it toward our target velocity field. The following proposition shows that the *evolution* of the velocity field $b_{t,a}(x)$ associated to (6) with respect to $a$ has a closed form defined solely in terms of known or learnable quantities.

**Proposition 3.2.** *(Covariance ODE.) The augmented drift $b_{t,a}(x)$ satisfies $b_{t,a=0}(x) = b_t(x)$ and*

$$\frac{\partial b_{t,a}(x)}{\partial a} = \mathbb{E}[(\dot{I}_t^a - b_{t,a}(x))r(x_1^a) | I_t^a = x], \quad (9)$$

*where the expectation is taken over the law of $I_t^a$ conditional on $I_t^a = x$. The right-hand side of this equation is the conditional covariance $\mathrm{Cov}_a(\dot{I}_t^a, r(x_1^a) | I_t^a = x)$.*

Proposition 3.2 is proven in Appendix A. The above relation can be interpreted as a dynamical formulation of the Esscher transform arising from (8). Applied to stochastic interpolant velocity fields, tilting by $e^{hr(x)}$ induces a flow on $b_{t,a}$ whose infinitesimal generator is the conditional covariance between the interpolant and the reward. Importantly, this evolution of $b_{t,a}$ in $a$ depends only on $b_{t,a}$, the modified interpolant (6), and the reward.

### 3.1. Explicit Tilt Matching

We begin by deriving our first concrete algorithm using the Covariance ODE (Proposition 3.2). Notice that if we have access to $b_{t,a}$, then we can also produce samples $x_1^a \sim \rho_{1,a}$, which together are sufficient to compute the derivative in (9). This suggests that the corrections to $b_{t,a=0}$ that need to be learned to sample the true tilted density be learned in an iterative fashion by discretizing (9). That is, for $0 < h \ll 1$, we can write an **explicit Euler discretization** as

$$b_{t,a+h}(x) \qquad\qquad (10)$$
$$= b_{t,a}(x) + h\frac{\partial b_{t,a}(x)}{\partial a} + \mathcal{O}(h^2)$$
$$= b_{t,a}(x) + h\mathbb{E}\left[(\dot{I}_t^a - b_{t,a}(x))r(x_1^a)\Big| I_t^a = x\right] + \mathcal{O}(h^2).$$

This perspective suggests an iterative, covariance-guided procedure. We start with $b_{t,0}$. Next, we approximate $\hat{b}_{t,h}$ over all $t \in [0,1]$, $x \in \mathbb{R}^d$ using (10). We then continue to iterate and generate successive updates $\hat{b}_{t,2h}, \hat{b}_{t,3h}, \dots, \hat{b}_{t,1}$ that gradually transform the velocity field toward the fully tilted distribution. As $h \to 0$, this discretization recovers the continuous evolution in (9), ensuring convergence to the desired $b_{t,1}$. To formalize this, we introduce the **residual operator**:

$$T_{t,a,h}^{\mathrm{ETM}} := b_{t,a}(I_t^a) + \underbrace{h\left(\dot{I}_t^a - b_{t,a}(I_t^a)\right)r(x_1^a)}_{\text{residual}}. \quad (11)$$

The following proposition shows that $b_{t,a+h}$ can be efficiently regressed and is first-order accurate using what we call **Explicit Tilt Matching (ETM)**:

**Proposition 3.3.** *(Explicit Tilt Matching.) Let $h > 0$. Then, the unique minimizer of the regression objective*

$$\mathcal{L}_{a \to a+h}^{\mathrm{ETM}}(\hat{b}) := \int_0^1 \mathbb{E}\left\|\hat{b}_t(I_t^a) - T_{t,a,h}^{\mathrm{ETM}}\right\|^2 dt, \quad (12)$$

*is given by*

$$\hat{b}_{t,a+h}(x) = \mathbb{E}\left[T_{t,a,h}^{\mathrm{ETM}} \,\big|\, I_t^a = x\right]. \quad (13)$$

*So, training $\hat{b}_{t,a+h}$ to optimality on (12) produces a first-order accurate Euler update of the tilted velocity.*

A proof of Proposition 3.3 is provided in Appendix A. This results shows that iterating for $a_k = kh$ with samples $x_1^{a_k}$ drawn using the current model defines a consistent scheme that converges to $b_{t,1}$ as $h \to 0$. See Algorithm 1.

**Regularity of transport.** Annealing across $a$ for $b_{t,a}$ via a procedure like this gives a velocity field $b_{t,a=1}$ with favorable regularity conditions in time $t$, since the ultimate transport from $\rho_{t=0,a=1}$ to $\rho_{t=1,a=1}$ follows the interpolant path. This makes $b_{t,a=1}$ well-posed to be estimated with neural networks, as the transport for such paths starting from the Gaussian is geometrically smooth and does not exhibit any teleportation. At the same time, there are two main approximation errors to account for, which we outline next.

**Approximation error due to incomplete minimization.** A first source of error arises if the regression problem in (12) is not minimized exactly at each iteration. This means the learned drift $\hat{b}_{t,a}$ may deviate from the ideal $b_{t,a}$, with errors compounding over successive updates. To mitigate this, we can introduce importance weights during training, which correct for residual mismatch between the model distribution and the tilted target. Alternatively, in settings where the potentials defining $\rho_{1,a}$ are known, this error can be mitigated by applying MCMC correction steps (e.g., MALA). Both strategies help debias the procedure and prevent incomplete optimization from undermining convergence.

**Discretization error in $a$.** A second source of error comes from discretizing the Covariance ODE in the annealing parameter $a$. Since (9) defines a continuous evolution, replacing it with discrete steps introduces a bias. In practice, this issue is negligible: one can choose $h$ to be very small, so the resulting discretization bias can be almost completely eliminated. Moreover, the step size can be adapted dynamically to ensure that updates remain close to the continuous trajectory. However, both remedies increase the computational cost of the method. This motivates alternative approaches that eliminate discretization error by construction, rather than managing it with such adaptive schemes.

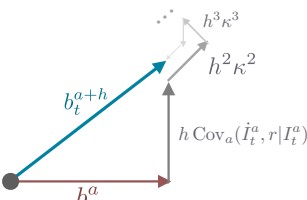

*Figure 3.* Pictorial additivity of higher-order corrections to $b_{t,a+h}$. The first-order term is the covariance, while higher-order terms are cumulants $\kappa^n$.

**PINN objective.** One approach to remove the discretization error would be to parametrize a network $\hat{b}_{t,a}$ to learn over all $a$ continuously, i.e. to minimize the residual

$$\mathcal{L}_{\text{PINN}}[\hat{b}] := \int_{[0,1]^2} \mathbb{E} \left\| \frac{\partial \hat{b}_{t,a}}{\partial a}(I_t^a) - (\dot{I}_t^a - \hat{b}_{t,a}(I_t^a))r(x_1^a) \right\|^2 da\, dt,$$

where the expectation is taken over $I_t^a$. However, this loss function requires computing derivatives in $a$ and would still need to be learned by annealing out in $a$ so as to properly construct the Monte Carlo estimator. For purposes that make bookkeeping easier computationally, we propose an alternative approach below.

### 3.2. Implicit Tilt Matching

**Higher-order expansions.** The explicit scheme defined by (12) arises by discretizing the evolution of $b_{t,a}$ given in (9) with a forward Euler step. While convenient, such updates inherit a discretization bias, which, even if small, might compound over successive steps in $a$. A natural extension is to consider taking higher-order Taylor expansions. Extending (10) to all orders gives

$$b_{t,a+h}(x) = b_{t,a}(x) + \sum_{n>0} \frac{h^n}{n!} \frac{\partial^n}{\partial a^n} [b_{t,a}(x)]. \tag{14}$$

This statement on its own is contentless, but the following proposition shows that each term in the expansion has rich meaning:

> **Proposition 3.4.** *(Tilt Expansion.)* The $n^{th}$ term $\frac{\partial^n}{\partial a^n}[b_{t,a}(x)]$ in the expansion in (14) is the $(n+1)^{th}$ order joint cumulant of the interpolant's velocity and $n$ instances of the reward, $\kappa^n(\dot{I}_t^a, r(x_1^a), \ldots, r(x_1^a))$.

This result is proven in Appendix A and relies on a relation between the Esscher representation of $b_{t,a+h}$ and the joint moment generating function of the interpolant's velocity and the reward. Since the cumulants involve higher-order moments of the reward, a Monte Carlo training objective that attempts to match these term by term would be computationally infeasible.

**Expanding to all orders.** In order to fully eliminate the discretization error, we should consider all higher-order cumulants. Surprisingly, this expression is tractable as we show in the following proposition. If we define the **implicit residual operator** as

$$T_{t,a,h}^{\text{ITM}}(\hat{b}) := b_{t,a}(I_t^a) + \underbrace{(e^{hr(x_1^a)} - 1)\left(\dot{I}_t^a - \text{sg}(\hat{b}_t)(I_t^a)\right)}_{\text{residual}},$$
$$\tag{15}$$

where $\text{sg}(\cdot)$ denotes a stop gradient, then we can learn the infinite cumulant expansion by directly matching against it.

> **Proposition 3.5.** *(Implicit Tilt Matching.)* Let $b_{t,a+h}$ be defined to all orders as in (14). Then
>
> $$\sum_{n>0} \frac{h^n}{n!} \frac{\partial^n}{\partial a^n} [b_{t,a}(x)] \tag{16}$$
> $$= \mathbb{E}[(e^{hr(x_1^a)} - 1)\left(\dot{I}_t^a - b_{t,a+h}(x)\right)|I_t^a = x].$$
>
> *Furthermore, for any $h$, $b_{t,a+h}$ is the unique fixed point of the following objective:*
>
> $$\mathcal{L}_{a \to a+h}^{\text{ITM}}(\hat{b}) := \int_0^1 \mathbb{E} \left\| \hat{b}_t(I_t^a) - T_{t,a,h}^{\text{ITM}}(\hat{b}) \right\|^2 dt, \tag{17}$$
>
> *where the expectation is taken over $(x_0, x_1^a) \sim \rho_0(x_0)\rho_{1,a}(x_1^a)$ conditional on $I_t^a = x$.*

For a proof of Proposition 3.5, see Appendix A. This result *removes the discretization error* inherent to ETM and shows that all orders of the correction to the interpolant velocity field are directly learnable. Enforcing that the residual update to $b_{t,a}$ is given by (16) is equivalent to enforcing that the residual update is exact *to all orders*. We call this condition **Implicit Tilt Matching (ITM)** because the residual term that we add to $b_{t,a}$ in (15) depends on our current estimate $\hat{b}_t$ of $b_{t,a+h}$, leading to this fixed-point method. See Algorithm 2.

The expression on the right-hand side of (16) may seem opaque, but it can be motivated with a simple derivation. Starting from the expression for the velocity $b_{t,a+h}$ in (8), we can multiply both sides by $\mathbb{E}[e^{hr(x)}|I_t^a = x]$ and rearrange terms to obtain an equivalent condition that $\hat{b}_t(x) = b_{t,a+h}(x)$ if and only if:

$$\mathbb{E}\left[e^{hr(x_1^a)}\left(\hat{b}_t(x) - \dot{I}_t^a\right) \Big| I_t^a = x\right] = 0. \tag{18}$$

Observe that (18) is precisely the first-order optimality condition of the ITM objective in (17), which can be seen by taking the derivative of the objective conditional on $I_t^a = x$.

**Variance reduction via control variates.** We can further introduce a scalar (or more generally a matrix-valued) control variate $c_t(x) : \mathbb{R}^d \to \mathbb{R}$ into (18) to obtain a generalized

optimality condition that $\hat{b}_t(x) = b_{t,a+h}(x)$ if and only if:

$$\mathbb{E}\Big[c_t(x)\big(\hat{b}_t(x) - b_{t,a}(x)\big) \tag{19}$$

$$+ \big(e^{hr(x_1^a)} - c_t(x)\big)\big(\hat{b}_t(x) - \dot{I}_t^a\big)\,\Big|\, I_t^a = x\Big] = 0,$$

where we used the fact that $\mathbb{E}[c_t(x)\dot{I}_t^a | I_t^a = x] = \mathbb{E}[c_t(x)b_{t,a}(x)|I_t^a = x]$. The optimality condition (19) is valid for any choice of $c_t(x)$ and therefore suggests a family of valid implicit objectives we could use to find $b_{t,a+h}$. We can enforce (19) with the following stop-gradient objective (valid for $c_t \neq 0$):

$$\mathcal{L}_{a \to a+h}^{\text{c-ITM}}(\hat{b}) := \int_0^1 \mathbb{E}\big\|c_t(I_t^a)\big(\hat{b}_t(I_t^a) - b_{t,a}(I_t^a)\big) \tag{20}$$

$$+ \big(e^{hr(x_1^a)} - c_t(I_t^a)\big)\big(\text{sg}(\hat{b}_t(I_t^a)) - \dot{I}_t^a\big)\big\|^2 dt.$$

The role of $c_t(x)$ is to control the variance of the Monte Carlo estimator of the loss function. Notice that the choice $c_t(x) = 1$ recovers (17) exactly. Moreover, this choice has the convenient property that for $h \ll 1$, it is close to the optimal control variate since $c_t(x) = 1$ clearly minimizes the variance conditional on $I_t^a = x$ when $h = 0$. More generally, one can optimize $c_t(x)$ to minimize the variance, yielding adaptive control variates that further stabilize training. See Appendix B for details on the control variate. We also remark that as an alternative to (20), we can enforce (19) via a weighted regression loss, with this c-ITM objective taking the general form (valid for any $c_t$):

$$\mathcal{L}_{a \to a+h}^{\text{c-ITM}}(\hat{b}) := \int_0^1 \mathbb{E}\Big[e^{-hr(x_1^a)}\big\|c_t(I_t^a)\big(\hat{b}_t(I_t^a) - b_{t,a}(I_t^a)\big)$$

$$+ \big(e^{hr(x_1^a)} - c_t(I_t^a)\big)\big(\hat{b}_t(I_t^a) - \dot{I}_t^a\big)\big\|^2\Big] dt. \tag{21}$$

The two optimization procedures behave similarly since the gradient of (20) is a $c_t(I_t^a)$ scaling of the gradient of (21). In fact, when $c_t(x) = 1$, both gradients are identical.

*Remark* 3.6. If learning a flow map $X_{s,t}(x) = x + (t - s)v_{s,t}(x)$ which takes solutions of (1) at time $s$ to its solution at time $t$ and $v_{s,t}(x)$ is learned as a neural network, then enforcing (18) over $v_{t,t}$ using ITM and minimizing any consistency loss for flow maps (Boffi et al., 2024; 2025) would allow us to straightforwardly apply Tilt Matching to the output of any-step maps.

## 3.3. Weighted Flow Matching

We now explore a particular instantiation of c-ITM. In principle, the tilted drift $b_{t,a+h}$ could be obtained by applying flow matching directly to the interpolant $I_t^{a+h}$ with samples $x_1^{a+h} \sim \rho_{1,a+h}$:

$$b_{t,a+h} = \arg\min_{\hat{b}_t} \int_0^1 \mathbb{E}\Big[\big\|\hat{b}_t(I_t^{a+h}) - \dot{I}_t^{a+h}\big\|^2\Big] dt. \tag{22}$$

Since we do not have samples from $\rho_{t,a+h}$, the expectation in (22) must be expressed in terms of $\rho_{t,a}$, from which we do have samples. Introducing importance weights leads to what we call the *weighted flow matching* (WFM) objective:

$$\mathcal{L}_{a \to a+h}^{\text{WFM}}(\hat{b}) := \int_0^1 \mathbb{E}\Big[e^{hr(x_1^a)}\big\|\hat{b}_t(I_t^a) - \dot{I}_t^a\big\|^2\Big] dt. \tag{23}$$

This is precisely the c-ITM loss (21) with $c_t(x) = 0$. Therefore WFM is an **instantiation** of c-ITM. Notice that WFM regresses directly on $\dot{I}_t^a$, whereas ITM substitutes $\dot{I}_t^a$ with its conditional expectation $b_{t,a}(I_t^a)$. Consequently, ITM enjoys strictly lower variance than the WFM objective for sufficiently small $h$. This behavior is reflected in Figure 4b, where WFM shows a faster decrease in effective sample size (ESS) with increasing $a$ compared to ITM, and is formalized in the following proposition.

**Proposition 3.7.** *(Variance Control.) Let $\mathcal{L}_{a \to a+h}^{\text{ITM}}$ and $\mathcal{L}_{a \to a+h}^{\text{WFM}}$ be the regression losses in (17) and (23). For sufficiently small $h$, the gradient estimator of WFM has variance at least as large as that of ITM:*

$$\text{Var}\big[\nabla \mathcal{L}_{a \to a+h}^{\text{WFM}}\big] \geq \text{Var}\big[\nabla \mathcal{L}_{a \to a+h}^{\text{ITM}}\big]. \tag{24}$$

We provide a proof of Proposition 3.7 in Appendix A. This result formalizes that ITM enjoys a variance advantage over WFM because it centers updates on the conditional mean $b_{t,a}(I_t^a)$ rather than the noisy sample $\dot{I}_t^a$. Furthermore, optimizing for the optimal control variate $c_t(x)$ guarantees a lower variance for any choice of step size $h$.

## 3.4. Connection to Doob's $h$-Transform and Stochastic Optimal Control

Given that we have so far presented several objectives for learning the tilted velocity field $b_{t,a}$, it is useful to relate $b_{t,a}$ to the drift used for tilting in Doob's $h$-transform (Doob, 1984; Dai Pra, 1991) and stochastic optimal control (SOC) (Fleming & Rishel, 2012; Domingo-Enrich et al., 2025). We begin by defining the interpolant's value function

$$v_{t,a}(x) := \log \mathbb{E}\Big[e^{ar(x_1)} \mid I_t^0 = x\Big]. \tag{25}$$

This value function characterizes the tilted marginals along the interpolant path. By definition of the tilt and the tower property of the conditional expectation, we can write $\rho_{t,a}$ as

$$\rho_{t,a}(x) \propto \mathbb{E}\Big[\delta(x - I_t^0)e^{ar(x_1)}\Big] \tag{26}$$

$$= \mathbb{E}\Big[\delta(x - I_t^0)\mathbb{E}\big[e^{ar(x_1)} \mid I_t^0\big]\Big] = \rho_{t,0}(x)\,e^{v_{t,a}(x)},$$

where $\delta$ is the Dirac delta. The value function also determines the drift correction required to realize $\rho_{t,a}$ from a reference stochastic dynamics. To make this connection

explicit, we convert the ODE (1) into an SDE as in (Song et al., 2020; Albergo et al., 2023), and consider the reference dynamics with marginals $\rho_{t,0}$ starting at $X_0 \sim \rho_0$:

$$dX_t = \left[b_{t,0}(X_t) + \tfrac{\sigma_t^2}{2}\nabla \log \rho_{t,0}(X_t)\right] dt + \sigma_t \, dB_t. \quad (27)$$

Doob's $h$-transform (equivalently, an SOC construction with quadratic control cost) tilts the path measure by the terminal weight $e^{ar(X_1)}$. The resulting controlled dynamics has a drift correction determined by the gradient of the uncontrolled process's value function, $V_{t,a}(x) := \log \mathbb{E}[e^{ar(X_1)} \mid X_t = x]$, yielding the controlled diffusion:

$$dX_t^a = \left[b_{t,0}(X_t^a) + \tfrac{\sigma_t^2}{2}\nabla \log \rho_{t,0}(X_t^a)\right. \quad (28)$$
$$\left. + \sigma_t^2 \nabla V_{t,a}(X_t^a)\right] dt + \sigma_t \, dB_t, \qquad X_0^a \sim \rho_0.$$

Interestingly, as we show in the following proposition, the Tilt Matching drift $b_{t,a}$ coincides with the drift arising from this Doob/SOC formulation for a specific choice of $\sigma_t$.

> **Proposition 3.8.** *(Doob's Probability Flow ODE.)*
> *Define the diffusion coefficient $\frac{\sigma_t^2}{2} = \frac{\dot\beta_t}{\beta_t}\alpha_t^2 - \dot\alpha_t\alpha_t$. Then $b_{t,a}$ which is the solution to (9) and the minimizer of $\mathcal{L}_{0\to a}^{\text{ITM}}$ is given by*
>
> $$b_{t,a}(x) = b_{t,0}(x) + \frac{\sigma_t^2}{2}\nabla v_{t,a}(x), \qquad (29)$$
>
> *and is precisely the drift corresponding to the probability flow ODE of (28).*

See Appendix C for a proof. Crucially, the choice of diffusion coefficient in Proposition 3.8 ensures that the conditional endpoint laws of the uncontrolled SDE (27) are equal to that of the interpolant, i.e., $\text{Law}(X_1 \mid X_t = x) = \text{Law}(x_1 \mid I_t^0 = x)$, which results in the equality of the two value functions, $v_{t,a}(x) = V_{t,a}(x)$. This provides a formal link between Tilt Matching and optimal-control-based viewpoints. At the same time, Tilt Matching avoids standard SOC routes for obtaining $\nabla v_{t,a}$ which typically require simulating trajectories, computing reward gradients, or tackling boundary value problems. Instead, Tilt Matching circumvents these challenges by recovering the optimal control through a simple iterative regression on scalar rewards.

## 4. Numerical Experiments

In what follows, we test ITM on both sampling Lennard-Jones (LJ) potentials (of 13 and 55 particles) and fine-tuning Stable Diffusion 1.5. All experiments use ITM, as it matches the simplicity of ETM while avoiding discretization error.

### 4.1. Fine-tuning Stable Diffusion 1.5

To validate our proposed method, we fine-tune Stable Diffusion 1.5 (Rombach et al., 2022) using the ImageReward score (Xu et al., 2023) as the objective. Our implementation builds upon the codebase and parameters established in (Domingo-Enrich et al., 2025; Blessing et al., 2025).

To ensure a comprehensive evaluation and mitigate concerns of overfitting to a single reward metric, we additionally assess performance across three distinct axes: (1) text-to-image consistency, measured by CLIPScore (Hessel et al., 2021); (2) human aesthetic preference, quantified by HPSv2 (Wu et al., 2023); and (3) sample diversity, evaluated with DreamSim (Fu et al., 2023). We primarily benchmark

*Table 1.* Fine-tuning results on Stable Diffusion 1.5. We compare our method against Adjoint Matching (AM) (Domingo-Enrich et al., 2025). We report on ImageReward (IR) (Xu et al., 2023), CLIPScore (CLIP) (Hessel et al., 2021), HPSv2 (Wu et al., 2023), and DreamSim (DS) (Fu et al., 2023). For all metrics, higher values are better ($\uparrow$). See Table 3 for standard deviations.

| Method | IR ($\uparrow$) | CLIP ($\uparrow$) | HPSv2 ($\uparrow$) | DS ($\uparrow$) |
|---|---|---|---|---|
| SD 1.5 (Base) | 0.1873 | 0.2746 | 0.2566 | **0.3849** |
| AM ($\lambda = 1$) | 0.2170 | 0.2754 | 0.2576 | 0.3826 |
| **ITM** ($\lambda = 1$) | **0.4465** | **0.2794** | **0.2659** | 0.3383 |
| AM ($\lambda = 10^2$) | 0.7873 | 0.2792 | 0.2791 | 0.3363 |

against adjoint matching (Domingo-Enrich et al., 2025), the current state-of-the-art for reward fine-tuning, which has demonstrated superior performance over prominent methods like DRaFT (Clark et al., 2024), DPO (Wallace et al., 2024), and ReFL (Xu et al., 2023). We emphasize that we fine-tune *only* on the ImageReward score, but measure performance on other scores as well.

Our results are summarized in Table 1, and sample uncurated images can be found in Appendix E. It is standard practice for adjoint matching to employ a reward multiplier, $\lambda$, which amplifies the reward signal to steer the learned distribution towards $\rho_1(x)e^{\lambda r(x)}$. A key finding in our experiments is that Tilt Matching achieves competitive performance without requiring a reward multiplier ($\lambda = 1$) or extensive hyperparameter tuning. This suggests that our approach provides a direct and stable mechanism for incorporating reward signals into the generation process, likely due to the fact that it does not rely on spatial gradients of the reward for training or generation. Further gains could likely be made by hyperparameter sweeps.

### 4.2. Sampling Lennard-Jones Potentials

In the context of sampling, the goal is to draw samples from a target density $\rho_{1,a=1}$, which is typically the Boltzmann distribution for a given potential energy function $E_1(x)$, such that $\rho_{1,a=1}(x) \propto \exp(-E_1(x))$. For TM, we begin with a simple prior density, $\rho_{1,a=0}$, which corresponds to an initial potential $E_0(x)$, and define an annealing path via

*Table 2.* Performance comparison on LJ-13 and LJ-55 using the effective sample size ESS, 1D energy histogram $E(\cdot)\ \mathcal{W}_2$ metric, and a geometric Geo $\mathcal{W}_2$ metric from (Klein et al., 2023). Missing $--$ entries indicate the metric is not available. We omit the ESS comparison for LJ-55 because it is too computationally intensive for us to compute, and other works do not provide a number to juxtapose with.

| | LJ-13 | | | LJ-55 | |
| Method | ESS ($\uparrow$) | $E(\cdot)\ \mathcal{W}_2$ ($\downarrow$) | Geo $\mathcal{W}_2$ ($\downarrow$) | $E(\cdot)\ \mathcal{W}_2$ ($\downarrow$) | Geo $\mathcal{W}_2$ ($\downarrow$) |
|---|---|---|---|---|---|
| DDS (Vargas et al., 2023) | 0.101 | 24.61 | 1.99 | 173.09 | 4.60 |
| iDEM (Akhound-Sadegh et al., 2024) | 0.231 | 30.78 | 1.61 | 93.53 | 4.69 |
| Adjoint Sampling (Havens et al., 2025) | $--$ | 2.40 | 1.67 | 30.83 | 4.04 |
| ASBS (Liu et al., 2025a) | $--$ | 1.99 | 1.59 | 28.10 | 4.00 |
| PITA (Akhound-Sadegh et al., 2025) | $--$ | 2.26 | 1.65 | $--$ | $--$ |
| **ITM (Ours)** | **0.507** | **0.879** | **1.54** | **26.37** | **3.98** |

linear interpolation:

$$E_a(x) = (1-a)E_0(x) + aE_1(x). \tag{30}$$

This defines a family of densities $\rho_{1,a}(x) \propto \exp(-E_a(x))$ for $a \in [0,1]$. This path is equivalent to the geometric annealing path described by the reward-tilted formulation, where the reward is given by:

$$r(x) = E_0(x) - E_1(x) \tag{31}$$

A common choice for the prior $\rho_{1,a=0}$ is a Gaussian distribution. For molecular systems, a more effective strategy is to define the prior as a high-temperature analogue of the target by setting the initial potential as $E_0(x) = E_1(x)/T_0$, where $T_0 \gg 1$ is a high temperature. The resulting prior, $\rho_{1,a=0} \propto \exp(-E_1(x)/T_0)$, has a smoothed energy landscape that facilitates more efficient MCMC sampling. We adopt this temperature annealing approach for our numerical experiments as it is a standard practice in the Boltzmann sampling literature and recent neural sampler approaches (Akhound-Sadegh et al., 2025; Rissanen et al., 2025). Our results are reported in Table 2 and Figure 4.

## 5. Related Work

**Neural samplers** Transport-based Monte Carlo sampling (Marzouk et al., 2016), popularized via neural normalizing flows (Rezende & Mohamed, 2015; Dinh et al., 2017; Noé et al., 2019; Albergo et al., 2019), has recently shifted toward continuous-time flow and diffusion models. While some approaches utilize optimal control (Zhang & Chen, 2022; Tzen & Raginsky, 2019; Vargas et al., 2023; Havens et al., 2025) these typically require simulating or backpropagating through stochastic trajectories, which can be unstable and expensive. In contrast, our method relies on couplings rather than trajectories, allowing us to leverage the efficiency of any-step flow maps (Boffi et al., 2024; 2025; Sabour et al., 2025). Furthermore, unlike PTSD (Rissanen et al., 2025) which relies on finite-difference score approximations, ITM provides an exact objective that remains compatible with complementary techniques like local parallel tempering refinement.

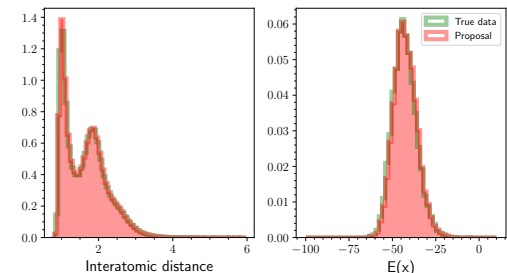

*(a)* Comparison of ITM samples vs ground truth molecular data.

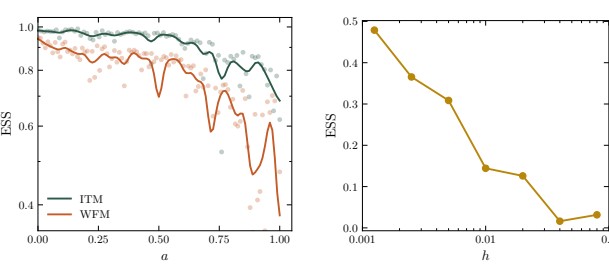

*(b)* ESS evolution with $a$.     *(c)* ITM ESS with step size $h$.

*Figure 4.* Sampling performance and ablation results for LJ13.

**Fine-tuning flows and diffusions** Existing work follows two high-level approaches: 1) reward-maximizing methods that directly optimize the quality of generated samples, such as D-Flow (Ben-Hamu et al., 2024) and DRaFT (Clark et al., 2024), and 2) distribution matching techniques that align the model with a reward-tilted distribution, seen in DEFT (Denker et al., 2024), adjoint matching (Domingo-Enrich et al., 2025), GFlowNet approaches (Zhang et al., 2024; Liu et al., 2025b), and approaches adapted from DPO (Wallace et al., 2024). Nevertheless, these algorithms frequently suffer from major disadvantages, including the need to differentiate through trajectories (Denker et al., 2024; Ben-Hamu et al., 2024; Clark et al., 2024) or the requirement of a differentiable reward function (Ben-Hamu et al., 2024; Clark et al., 2024; Zhang et al., 2024; Liu et al., 2025b; Domingo-Enrich et al., 2025), while some are only approximate (Wallace et al., 2024). The proposed Tilt Matching method is free from these limitations.

## Impact Statement

This paper presents work whose goal is to advance the field of Machine Learning. There are many potential societal consequences of our work, none which we feel must be specifically highlighted here.

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

## A. Proofs

**Proposition 3.1.** *(Esscher Transform.) Let $I_t^a = \alpha_t x_0 + \beta_t x_1^a$ be the interpolant constructed from samples $x_1^a \sim \rho_{1,a}$. Then the augmented velocity field $b_{t,a}(x)$ is given by:*

$$b_{t,a}(x) = \frac{\mathbb{E}[\dot{I}_t^0 e^{ar(x_1)} | I_t^0 = x]}{\mathbb{E}[e^{ar(x_1)} | I_t^0 = x]}. \tag{7}$$

*Furthermore, for any shift $h$ from $a$ to $a + h$, the updated velocity $b_{t,a+h}(x)$ satisfies:*

$$b_{t,a+h}(x) = \frac{\mathbb{E}[\dot{I}_t^a e^{hr(x_1^a)} | I_t^a = x]}{\mathbb{E}[e^{hr(x_1^a)} | I_t^a = x]}. \tag{8}$$

*(Back to Proposition 3.1 in the main text.)*

*Proof.* Let $Z_a := \mathbb{E}[e^{ar(x_1^0)}]$ be the normalizing constant for the tilted distribution $\rho_{1,a}(x) = \rho_{1,0}(x)e^{ar(x)}/Z_a$. The probability density function of the tilted interpolant $I_t^a$ is given by:

$$\rho_{t,a}(x) = \frac{1}{Z_a}\mathbb{E}[\delta(x - I_t^0)e^{ar(x_1^0)}], \tag{32}$$

where the expectation is over the base coupling $(x_0, x_1^0) \sim \rho_0(x_0)\rho_{1,0}(x_1^0)$. Taking the time derivative of $\rho_{t,a}$ and applying the chain rule to the delta function, we have:

$$\partial_t \rho_{t,a}(x) = \frac{1}{Z_a}\partial_t \mathbb{E}[\delta(x - I_t^0)e^{ar(x_1^0)}] \tag{33}$$

$$= \frac{1}{Z_a}\mathbb{E}[-\nabla\delta(x - I_t^0) \cdot \dot{I}_t^0 e^{ar(x_1^0)}] \tag{34}$$

$$= -\nabla \cdot \left(\frac{1}{Z_a}\mathbb{E}[\delta(x - I_t^0)\dot{I}_t^0 e^{ar(x_1^0)}]\right). \tag{35}$$

The continuity equation for $\rho_{t,a}$ is $\partial_t \rho_{t,a} + \nabla \cdot (b_{t,a}\rho_{t,a}) = 0$. Comparing terms inside the divergence, we identify:

$$b_{t,a}(x)\rho_{t,a}(x) = \frac{1}{Z_a}\mathbb{E}[\delta(x - I_t^0)\dot{I}_t^0 e^{ar(x_1^0)}]. \tag{36}$$

Assuming $\rho_{t,a}(x) > 0$, we solve for $b_{t,a}(x)$ by dividing by $\rho_{t,a}(x)$:

$$b_{t,a}(x) = \frac{\frac{1}{Z_a}\mathbb{E}[\delta(x - I_t^0)\dot{I}_t^0 e^{ar(x_1^0)}]}{\frac{1}{Z_a}\mathbb{E}[\delta(x - I_t^0)e^{ar(x_1^0)}]}. \tag{37}$$

The normalizing constant $1/Z_a$ cancels out. We then divide the numerator and denominator by the marginal density of the base interpolant, $\rho_{t,0}(x) = \mathbb{E}[\delta(x - I_t^0)]$, to express the result in terms of conditional expectations:

$$b_{t,a}(x) = \frac{\mathbb{E}[\dot{I}_t^0 e^{ar(x_1^0)} | I_t^0 = x]}{\mathbb{E}[e^{ar(x_1^0)} | I_t^0 = x]}. \tag{38}$$

This establishes (7). The derivation for (8) follows identically by replacing the base distribution $\rho_{1,0}$ with $\rho_{1,a}$ and the tilt $e^{ar}$ with $e^{hr}$. $\qquad\square$

**Proposition 3.2.** *(Covariance ODE.) The augmented drift $b_{t,a}(x)$ satisfies $b_{t,a=0}(x) = b_t(x)$ and*

$$\frac{\partial b_{t,a}(x)}{\partial a} = \mathbb{E}[(\dot{I}_t^a - b_{t,a}(x))r(x_1^a)|I_t^a = x], \tag{9}$$

*where the expectation is taken over the law of $I_t^a$ conditional on $I_t^a = x$. The right-hand side of this equation is the conditional covariance $\mathrm{Cov}_a(\dot{I}_t^a, r(x_1^a) \,|\, I_t^a = x)$.*

*(Back to Proposition 3.2 in the main text.)*

*Proof.* To show (9), we begin by recalling (37) which states

$$b_{t,a}(x) = \frac{\mathbb{E}[\delta(x - I_t^0)\dot{I}_t^0 e^{ar(x_1^0)}]}{\mathbb{E}[\delta(x - I_t^0)e^{ar(x_1^0)}]}. \tag{39}$$

Taking its derivative with respect to a and using the quotient rule, we have

$$\frac{\partial}{\partial a}b_{t,a}(x) = \frac{\mathbb{E}[\delta(x - I_t^0)\dot{I}_t^0 e^{ar(x_1^0)}r(x_1^0)]}{\mathbb{E}[\delta(x - I_t^0)e^{ar(x_1^0)}]} - \frac{\mathbb{E}[\delta(x - I_t^0)\dot{I}_t^0 e^{ar(x_1^0)}]}{\mathbb{E}[\delta(x - I_t^0)e^{ar(x_1^0)}]}\frac{\mathbb{E}[\delta(x - I_t^0)e^{ar(x_1^0)}r(x_1^0)]}{\mathbb{E}[\delta(x - I_t^0)e^{ar(x_1^0)}]} \tag{40}$$

$$= \mathbb{E}[\dot{I}_t^a r(x_1^a)|I_t^a = x] - b_{t,a}(x)\,\mathbb{E}[r(x_1^a)|I_t^a = x], \tag{41}$$

which completes the proof. □

**Proposition 3.3.** *(Explicit Tilt Matching.) Let $h > 0$. Then, the unique minimizer of the regression objective*

$$\mathcal{L}_{a\to a+h}^{\mathrm{ETM}}(\hat{b}) := \int_0^1 \mathbb{E}\left\|\hat{b}_t(I_t^a) - T_{t,a,h}^{\mathrm{ETM}}\right\|^2 dt, \tag{12}$$

*is given by*

$$\hat{b}_{t,a+h}(x) = \mathbb{E}\left[T_{t,a,h}^{\mathrm{ETM}} \,\big|\, I_t^a = x\right]. \tag{13}$$

*So, training $\hat{b}_{t,a+h}$ to optimality on (12) produces a first-order accurate Euler update of the tilted velocity.*

*(Back to Proposition 3.3 in the main text.)*

*Proof.* By the Hilbert $L^2$ projection theorem, among all functions of $I_t^a$, the optimizer is the conditional expectation $\mathbb{E}[T_{t,a,h} \mid I_t^a = x]$. Expanding $b_{t,a+h} = b_{t,a} + h\,\partial_a b_{t,a} + \mathcal{O}(h^2)$ and using (9) yields the result. □

**Proposition 3.4.** *(Tilt Expansion.) The $n^{th}$ term $\frac{\partial^n}{\partial a^n}\big[b_{t,a}(x)\big]$ in the expansion in (14) is the $(n+1)^{th}$ order joint cumulant of the interpolant's velocity and $n$ instances of the reward, $\kappa^n(\dot{I}_t^a, r(x_1^a), \ldots, r(x_1^a))$.*

*(Back to Proposition 3.4 in the main text.)*

*Proof.* For a fixed $x$, define the joint conditional cumulant generating function of $r(x_1^a)$ and $\dot{I}_t^a$ as

$$M(\mu,\nu) = \log\mathbb{E}\left[e^{\mu r(x_1^a)+\langle\nu,\dot{I}_t^a\rangle}|I_t^a = x\right], \tag{42}$$

for $\mu \in \mathbb{R}$ and $\nu \in \mathbb{R}^d$. Its partial derivative with respect to $\nu$ evaluated at 0 is

$$\frac{\partial}{\partial\nu}M(\mu,0) = \frac{\mathbb{E}[\dot{I}_t^a e^{\mu r(x_1^a)}|I_t^a = x]}{\mathbb{E}[e^{\mu r(x_1^a)}|I_t^a = x]} = b_{t,a+\mu}(x), \tag{43}$$

where the second equality is (8). Taking $n$ derivatives with respect to $\mu$ and evaluating at 0, we obtain

$$\frac{\partial^{n+1}}{\partial\mu^n\partial\nu}M(0,0) = \frac{\partial^n}{\partial\mu^n}b_{t,a+\mu}(x)\Big|_{\mu=0} = \frac{\partial^n}{\partial a^n}b_{t,a}(x). \tag{44}$$

The leftmost term is precisely the $(n+1)^{\mathrm{th}}$ order joint cumulant. □

**Proposition 3.5.** *(Implicit Tilt Matching.)* *Let $b_{t,a+h}$ be defined to all orders as in* (14). *Then*

$$\sum_{n>0} \frac{h^n}{n!} \frac{\partial^n}{\partial a^n} \left[ b_{t,a}(x) \right] \tag{16}$$

$$= \mathbb{E}[(e^{hr(x_1^a)} - 1) (\dot{I}_t^a - b_{t,a+h}(x)) | I_t^a = x].$$

*Furthermore, for any h, $b_{t,a+h}$ is the unique fixed point of the following objective:*

$$\mathcal{L}_{a \to a+h}^{\mathrm{ITM}}(\hat{b}) := \int_0^1 \mathbb{E} \left\| \hat{b}_t(I_t^a) - T_{t,a,h}^{\mathrm{ITM}}(\hat{b}) \right\|^2 dt, \tag{17}$$

*where the expectation is taken over $(x_0, x_1^a) \sim \rho_0(x_0)\rho_{1,a}(x_1^a)$ conditional on $I_t^a = x$.*

*(Back to Proposition 3.5 in the main text.)*

*Proof.* By taking the functional gradient of the stop-gradient objective (17), we see that $\hat{b}_t$ is a fixed point of the optimization procedure precisely when

$$\mathbb{E}\left[ \left( \hat{b}_t(x) - b_{t,a}(x) \right) + \left( e^{hr(x_1^a)} - 1 \right) \left( \hat{b}_t(x) - \dot{I}_t^a \right) \,\Big|\, I_t^a = x \right] = 0. \tag{45}$$

Using the identity $b_{t,a}(x) = \mathbb{E}[\dot{I}_t^a \mid I_t^a = x]$, the above equality occurs iff

$$\mathbb{E}\left[ e^{hr(x_1^a)} \left( \hat{b}_t(x) - \dot{I}_t^a \right) \,\Big|\, I_t^a = x \right] = 0. \tag{46}$$

Rearranging this equality, we obtain the equivalent condition

$$\hat{b}_t(x) = \frac{\mathbb{E}\left[ \dot{I}_t^a e^{hr(x_1^a)} \mid I_t^a = x \right]}{\mathbb{E}\left[ e^{hr(x_1^a)} \mid I_t^a = x \right]} = b_{t,a+h}(x). \tag{47}$$

Therefore the unique fixed point of the optimization procedure induced by (17) is $b_{t,a+h}$. Now we turn to showing (16). Notice that the left-hand side of (16) is equal to $b_{t,a+h}(x) - b_{t,a}(x)$ since the series contains all terms but the $0^{\mathrm{th}}$ order term in the Taylor series expansion in $h$ for $b_{t,a+h}(x)$. Next, we rewrite (45) with $\hat{b}_t = b_{t,a+h}$

$$\mathbb{E}\left[ \left( b_{t,a+h}(x) - b_{t,a}(x) \right) + \left( e^{hr(x_1^a)} - 1 \right) \left( b_{t,a+h}(x) - \dot{I}_t^a \right) \,\Big|\, I_t^a = x \right] = 0. \tag{48}$$

Rearranging, we obtain the following

$$b_{t,a+h}(x) - b_{t,a}(x) = \mathbb{E}\left[ \left( e^{hr(x_1^a)} - 1 \right) \left( \dot{I}_t^a - b_{t,a+h}(x) \right) \,\Big|\, I_t^a = x \right], \tag{49}$$

which is the right-hand side of (16). $\square$

**Proposition 3.7.** *(Variance Control.)* *Let $\mathcal{L}_{a \to a+h}^{\mathrm{ITM}}$ and $\mathcal{L}_{a \to a+h}^{\mathrm{WFM}}$ be the regression losses in* (17) *and* (23). *For sufficiently small h, the gradient estimator of WFM has variance at least as large as that of ITM:*

$$\mathrm{Var}\left[ \nabla \mathcal{L}_{a \to a+h}^{\mathrm{WFM}} \right] \geq \mathrm{Var}\left[ \nabla \mathcal{L}_{a \to a+h}^{\mathrm{ITM}} \right]. \tag{24}$$

*(Back to Proposition 3.7 in the main text.)*

*Proof.* The first variation (Gateaux derivative) of the loss $\mathcal{L}_{a \to a+h}^{\mathrm{c-ITM}}$ is

$$\delta \mathcal{L}_{a \to a+h}^{\mathrm{c-ITM}}(\hat{b}) = 2 \int_0^1 \mathbb{E} \left[ c_t(I_t^a) \left( \hat{b}_t(I_t^a) - b_{t,a}(I_t^a) \right) + \left( e^{hr(x_1^a)} - c_t(I_t^a) \right) \left( \hat{b}_t(I_t^a) - \dot{I}_t^a \right) \right] dt. \tag{50}$$

The following Monte Carlo estimator of the gradient is used in $\mathcal{L}_{a \to a+h}^{\mathrm{c-ITM}}$:

$$\xi_c := 2 \left( c_t(I_t^a) \left( \hat{b}_t(I_t^a) - b_{t,a}(I_t^a) \right) + \left( e^{hr(x_1^a)} - c_t(I_t^a) \right) \left( \hat{b}_t(I_t^a) - \dot{I}_t^a \right) \right), \tag{51}$$

where $t \sim \text{Unif}[0,1]$. We will use the law of total variance, which states the following

$$\text{Var}(\xi_c) = \mathbb{E}[\text{Var}(\xi_c|I_t^a)] + \text{Var}(\mathbb{E}[\xi_c|I_t^a]). \tag{52}$$

Notice that the conditional expectation of our estimator,

$$\mathbb{E}[\xi_c|I_t^a] = \mathbb{E}\left[e^{hr(x_1^a)}\left(\hat{b}_t(I_t^a) - \dot{I}_t^a\right)|I_t^a\right], \tag{53}$$

is independent of $c$ and therefore the same for any c-ITM variant. On the other hand, we have that

$$\text{Var}(\xi_c|I_t^a) = \text{Var}\left(e^{hr(x_1^a)}\left(\hat{b}_t(I_t^a) - \dot{I}_t^a\right) + c_t(I_t^a)\dot{I}_t^a|I_t^a\right). \tag{54}$$

Writing $e^{hr(x_1^a)} = 1 + \mathcal{O}(h)$, we see that

$$\mathbb{E}\left[\text{Var}(\xi_c|I_t^a)\right] = \mathbb{E}\left[\text{Var}\left((1 - c_t(I_t^a))\dot{I}_t^a|I_t^a\right)\right] + \mathcal{O}(h). \tag{55}$$

Recall that $\mathcal{L}_{a \to a+h}^{\text{ITM}}$ corresponds to taking $c_t(x) = 1$ and $\mathcal{L}_{a \to a+h}^{\text{WFM}}$ to taking $c_t(x) = 0$. When $c_t(x) = 1$, we have $\mathbb{E}\left[\text{Var}(\xi_c|I_t^a)\right] = \mathcal{O}(h)$. When $c_t(x) = 0$, we have $\mathbb{E}\left[\text{Var}(\xi_c|I_t^a)\right] = \mathbb{E}[\text{Var}(\dot{I}_t^a|I_t^a)] + \mathcal{O}(h)$. Since $\mathbb{E}[\text{Var}(\dot{I}_t^a|I_t^a)] > 0$, this completes the proof. Note that when $\hat{b}$ takes a parametric form, a similar proof holds. $\qquad\square$

# B. Control Variates and Joint Optimization

In this Appendix, we provide additional details on the control variates introduced in Section 3.2. In order to learn the optimal control variate $c_t^*(x)$, one may parameterize $\hat{c}_t(x)$ as a small additional head or a standalone network and train it jointly with the velocity field to minimize the Monte Carlo variance of the ITM estimator. Here we prove that such a procedure is valid and yields the correct velocity field. We augment the c-ITM loss in (21) to include $\hat{c}$ as an input. We first write the loss pointwise. Using the tower property, we have

$$\mathcal{L}_{a \to a+h}^{\text{c-ITM}}(\hat{b}, \hat{c}) := \int_0^1 \mathbb{E}\left[e^{-hr(x_1^a)}\left\|\hat{c}_t(I_t^a)\left(\hat{b}_t(I_t^a) - b_{t,a}(I_t^a)\right) + \left(e^{hr(x_1^a)} - \hat{c}_t(I_t^a)\right)\left(\hat{b}_t(I_t^a) - \dot{I}_t^a\right)\right\|^2\right]dt$$

$$= \int_0^1 \mathbb{E}\left[\mathbb{E}\left[e^{-hr(x_1^a)}\left\|\hat{c}_t(x)\left(\hat{b}_t(x) - b_{t,a}(x)\right) + \left(e^{hr(x_1^a)} - \hat{c}_t(x)\right)\left(\hat{b}_t(x) - \dot{I}_t^a\right)\right\|^2 \mid I_t^a = x\right]\right]dt$$

$$= \int_0^1 \mathbb{E}\left[J_t(\hat{b}_t(I_t^a), \hat{c}_t(I_t^a), I_t^a)\right]dt,$$

where we define the pointwise loss at $I_t^a = x$ as

$$J_t(\bar{b}, \bar{c}, x) := \mathbb{E}\left[e^{-hr(x_1^a)}\left\|\bar{c}\left(\bar{b} - b_{t,a}(x)\right) + \left(e^{hr(x_1^a)} - \bar{c}\right)\left(\bar{b} - \dot{I}_t^a\right)\right\|^2 \mid I_t^a = x\right]. \tag{56}$$

We use $\bar{b} \in \mathbb{R}^d, \bar{c} \in \mathbb{R}$ to denote that these are vectors rather than the time-dependent functions $\hat{b}_t : \mathbb{R}^d \to \mathbb{R}^d, \hat{c}_t : \mathbb{R}^d \to \mathbb{R}$. Since the distribution of $I_t^a$ is fixed at $\rho_{t,a}$, independent of $\hat{b}, \hat{c}$, it is clear that to minimize the c-ITM objective over $\hat{b}, \hat{c}$, we want to minimize it pointwise:

$$\hat{b}_t(x), \hat{c}_t(x) = \underset{\bar{b}, \bar{c} \in \mathbb{R}^d}{\arg\min}\, J_t(\bar{b}, \bar{c}, x). \tag{57}$$

Therefore we investigate the pointwise minimization problem and show that we can optimize over $\bar{b}$ and $\bar{c}$ separately.

**Proposition B.1.** *(Weighted Bias-Variance Decomposition). Let $w = e^{hr(x_1^a)}$. The pointwise loss $J_t(\bar{b}, \bar{c}, x)$ decomposes into two independent terms:*

$$J_t(\bar{b}, \bar{c}, x) = \|\bar{b} - b_{t,a+h}(x)\|^2 \mathbb{E}[w \mid I_t^a = x] + \mathbb{E}\left[w\left\|\frac{\bar{c}b_{t,a}(x) + (w - \bar{c})\dot{I}_t^a}{w} - b_{t,a+h}(x)\right\|^2 \mid I_t^a = x\right]. \tag{58}$$

*Minimizing with respect to $\bar{b}$ minimizes the bias (first term), while minimizing with respect to $\bar{c}$ minimizes the weighted variance of the regression target (second term). Therefore the minimizer $\bar{b}^*$ is the target drift $b_{t,a+h}(x)$. Furthermore, the optimal control variate $\bar{c}^*$ is given by*

$$\bar{c}^* = \frac{\mathbb{E}\left[\langle \dot{I}_t^a - b_{t,a}(x), \dot{I}_t^a - b_{t,a+h}(x)\rangle \mid I_t^a = x\right]}{\mathbb{E}\left[w^{-1}\|\dot{I}_t^a - b_{t,a}(x)\|^2 \mid I_t^a = x\right]}. \tag{59}$$

*Proof.* We start with the expression inside the squared norm of the objective in (56). We group the terms multiplying $\bar{b}$:

$$\bar{c}(\bar{b} - b_{t,a}(x)) + (w - \bar{c})(\bar{b} - \dot{I}_t^a) = (\bar{c} + w - \bar{c})\bar{b} - (\bar{c}b_{t,a}(x) + (w - \bar{c})\dot{I}_t^a) \tag{60}$$

$$= w\bar{b} - (\bar{c}b_{t,a}(x) + w\dot{I}_t^a - \bar{c}\dot{I}_t^a). \tag{61}$$

To simplify the notation, we define the stochastic target $T$:

$$T := \frac{\bar{c}b_{t,a}(x) + w\dot{I}_t^a - \bar{c}\dot{I}_t^a}{w}. \tag{62}$$

The pointwise loss can now be rewritten as a weighted mean squared error:

$$J_t(\bar{b}, \bar{c}, x) = \mathbb{E}\left[w^{-1}\|w(\bar{b} - T)\|^2 \mid I_t^a = x\right] = \mathbb{E}\left[w\|\bar{b} - T\|^2 \mid I_t^a = x\right]. \tag{63}$$

We add and subtract the true tilted velocity $b_{t,a+h}(x)$ inside the norm:

$$J_t(\bar{b}, \bar{c}, x) = \mathbb{E}\left[w\|(\bar{b} - b_{t,a+h}(x)) + (b_{t,a+h}(x) - T)\|^2 \mid I_t^a = x\right] \tag{64}$$

$$= \|\bar{b} - b_{t,a+h}(x)\|^2 \mathbb{E}[w \mid I_t^a = x] + \mathbb{E}\left[w\|T - b_{t,a+h}(x)\|^2 \mid I_t^a = x\right] \tag{65}$$

$$+ 2\langle \bar{b} - b_{t,a+h}(x), \mathbb{E}[w(b_{t,a+h}(x) - T) \mid I_t^a = x]\rangle. \tag{66}$$

We now show that the cross-term in the last line vanishes. We compute $\mathbb{E}[wT \mid I_t^a = x]$ explicitly:

$$\mathbb{E}[wT \mid I_t^a = x] = \mathbb{E}\left[\bar{c}b_{t,a}(x) + w\dot{I}_t^a - \bar{c}\dot{I}_t^a \mid I_t^a = x\right] \tag{67}$$

$$= \bar{c}b_{t,a}(x) + \mathbb{E}[w\dot{I}_t^a \mid I_t^a = x] - \bar{c}\underbrace{\mathbb{E}[\dot{I}_t^a \mid I_t^a = x]}_{b_{t,a}(x)} \tag{68}$$

$$= \mathbb{E}[w\dot{I}_t^a \mid I_t^a = x]. \tag{69}$$

From (8), the true tilted drift satisfies $b_{t,a+h}(x)\mathbb{E}[w \mid I_t^a = x] = \mathbb{E}[w\dot{I}_t^a \mid I_t^a = x]$. Therefore:

$$\mathbb{E}[w(b_{t,a+h}(x) - T) \mid I_t^a = x] = b_{t,a+h}(x)\mathbb{E}[w \mid I_t^a = x] - \mathbb{E}[wT \mid I_t^a = x] = 0. \tag{70}$$

Since the cross-term is zero, the loss separates into the two terms stated in the proposition. To find the optimal $\bar{c}^*$, we minimize the second term with respect to $\bar{c}$. The variance term is:

$$V(\bar{c}) := \mathbb{E}\left[w\left\|-\frac{\bar{c}}{w}(\dot{I}_t^a - b_{t,a}(x)) + \dot{I}_t^a - b_{t,a+h}(x)\right\|^2 \mid I_t^a = x\right] \tag{71}$$

$$= \mathbb{E}\left[w\left(\frac{\bar{c}^2}{w^2}\|\dot{I}_t^a - b_{t,a}(x)\|^2 - 2\frac{\bar{c}}{w}\langle \dot{I}_t^a - b_{t,a}(x), \dot{I}_t^a - b_{t,a+h}(x)\rangle + \|\dot{I}_t^a - b_{t,a+h}(x)\|^2\right) \mid I_t^a = x\right] \tag{72}$$

$$= \bar{c}^2 \mathbb{E}\left[w^{-1}\|\dot{I}_t^a - b_{t,a}(x)\|^2 \mid I_t^a = x\right] - 2\bar{c}\mathbb{E}\left[\langle \dot{I}_t^a - b_{t,a}(x), \dot{I}_t^a - b_{t,a+h}(x)\rangle \mid I_t^a = x\right] + \text{const.} \tag{73}$$

Minimizing the quadratic by solving $\frac{d}{d\bar{c}}V(\bar{c}) = 0$ yields the result. $\qquad\square$

*Remark* B.2. *(Interpretation of Variance Minimization).* The functional gradient of the c-ITM objective (21) with respect to $\hat{b}$ is given by the estimator $\hat{g} = 2w(\hat{b} - T)$, using the notation in our proof. The variance of this estimator depends on the probability measure against which it is evaluated. Our objective minimizes the variance of the regression target $T$ under the *target measure $\rho_{t,a+h}$* since

$$\mathbb{E}_{\rho_{t,a}}\left[w\|\hat{b} - T\|^2\right] \propto \mathbb{E}_{\rho_{t,a+h}}\left[\|\hat{b} - T\|^2\right]. \tag{74}$$

*Remark* B.3. For the stop-gradient objective $\mathcal{L}^{\text{c}-\text{sg}-\text{ITM}}$, joint optimization is also possible, but the optimal control variate is different and is given by:

$$\bar{c}_{\text{sg}}^* = \frac{\mathbb{E}\left[w\langle \dot{I}_t^a - b_{t,a+h}(x), \dot{I}_t^a - b_{t,a}(x)\rangle \mid I_t^a = x\right]}{\mathbb{E}\left[\|\dot{I}_t^a - b_{t,a}(x)\|^2 \mid I_t^a = x\right]}. \tag{75}$$

In this case, the optimal control variate explicitly minimizes the variance of the gradient estimator $\hat{g}$ itself, rather than the variance of the target $T$, under the *sampling measure* $\rho_{t,a}$. This follows from the law of total variance decomposition, $\text{Var}(\hat{g}) = \mathbb{E}[\text{Var}(\hat{g} \mid I_t^a)] + \text{Var}(\mathbb{E}[\hat{g} \mid I_t^a])$. Since the expected gradient $\mathbb{E}[\hat{g} \mid I_t^a]$ is the same for any control variate $c$, minimizing the conditional variance term pointwise minimizes the total variance.

## C. Tilt Matching, Doob's $h$-Transform and Stochastic Optimal Control

In this Appendix, we provide additional details on the discussion in Section 3.4, demonstrating that the drift learned by Tilt Matching $b_{t,a}$ is mathematically equivalent to the probability flow ODE of a Doob $h$-transformed SDE, provided the diffusion coefficient is chosen to match the conditional laws of the interpolant.

**Setup.** Consider the base probability flow ODE (1) defined by the velocity field $b_t(x)$:

$$\dot{x}_t = b_t(x_t), \quad x_0 \sim \rho_0, \tag{76}$$

with $\text{Law}(x_t) = \rho_t$. For any diffusion schedule $\sigma_t$, we define the following SDE as in (Song et al., 2020; Albergo et al., 2023):

$$dX_t = \left[b_t(X_t) + \frac{\sigma_t^2}{2}\nabla \log p_t(X_t)\right]dt + \sigma_t dB_t. \tag{77}$$

Then $\text{Law}(X_t) = \text{Law}(x_t) = \rho_t$. We specifically choose $\frac{\sigma_t^2}{2} = \frac{\dot{\beta}_t}{\beta_t}\alpha_t^2 - \dot{\alpha}_t\alpha_t$ so that the conditional endpoint laws of the SDE match those of the interpolant $I_t = \alpha_t x_0 + \beta_t x_1$:

$$\text{Law}(X_1 \mid X_t) = \text{Law}(I_1 \mid I_t). \tag{78}$$

**The tilted SDE via Doob's $h$-transform.** We wish to sample from the tilted distribution $\rho_{1,a}(x) \propto \rho_1(x)e^{ar(x)}$. Let the value function of the SDE be defined as:

$$V_{t,a}(x) := \log \mathbb{E}[e^{ar(X_1)} \mid X_t = x]. \tag{79}$$

Using Doob's $h$-transform, the SDE that targets the tilted distribution and has marginals $\rho_{t,a}(x) \propto \rho_t(x)e^{V_{t,a}(x)}$ is obtained by adding the drift correction $\sigma_t^2 \nabla V_{t,a}(x)$:

$$d\tilde{X}_t = \left[b_t(\tilde{X}_t) + \frac{\sigma_t^2}{2}\nabla \log \rho_t(\tilde{X}_t) + \sigma_t^2\nabla V_{t,a}(\tilde{X}_t)\right]dt + \sigma_t dB_t. \tag{80}$$

**The tilted probability flow ODE.** We convert the tilted SDE (80) back into its corresponding probability flow ODE. Recall that the probability flow ODE for a general SDE is given by subtracting half the diffusion-scaled score from the SDE's drift. Using the drift from (80) and the score $\nabla \log \rho_{t,a}(x) = \nabla \log \rho_t(x) + \nabla V_{t,a}(x)$:

$$\dot{\tilde{x}}_t = \left[b_t(\tilde{x}_t) + \frac{\sigma_t^2}{2}\nabla \log \rho_t(\tilde{x}_t) + \sigma_t^2\nabla V_{t,a}(\tilde{x}_t)\right] - \frac{\sigma_t^2}{2}\left[\nabla \log \rho_t(\tilde{x}_t) + \nabla V_{t,a}(\tilde{x}_t)\right] \tag{81}$$

$$= \underbrace{b_t(\tilde{x}_t) + \frac{\sigma_t^2}{2}\nabla V_{t,a}(\tilde{x}_t)}_{=:\tilde{b}_{t,a}(\tilde{x}_t)}, \tag{82}$$

which is the original ODE drift plus a correction coming from the gradient of the value function.

**Connection to tilt matching.** We claim that $\tilde{b}_{t,a}(x) = b_{t,a}(x)$, where $b_{t,a}$ is the drift obtained via Tilt Matching. At $a = 0$, both recover the base velocity: $b_t(x) = \tilde{b}_{t,a=0}(x) = b_{t,a=0}(x)$. To show they are identical for all $a \in [0,1]$, it suffices to show that their derivatives with respect to $a$ are equal. First, we differentiate the Doob-derived drift $\tilde{b}_{t,a}$ defined in equation 82:

$$\frac{\partial}{\partial a}\tilde{b}_{t,a}(x) = \frac{\sigma_t^2}{2}\nabla\left(\frac{\partial}{\partial a}V_{t,a}(x)\right) \tag{83}$$

$$= \frac{\sigma_t^2}{2}\nabla\left(\frac{\partial}{\partial a}\log\mathbb{E}[\exp(ar(X_1)) \mid X_t = x]\right) \tag{84}$$

$$= \frac{\sigma_t^2}{2}\nabla\left(\frac{\mathbb{E}[r(X_1)\exp(ar(X_1)) \mid X_t = x]}{\mathbb{E}[\exp(ar(X_1)) \mid X_t = x]}\right) \tag{85}$$

$$= \frac{\sigma_t^2}{2}\nabla\left(\frac{\mathbb{E}[r(I_1)\exp(ar(I_1)) \mid I_t = x]}{\mathbb{E}[\exp(ar(I_1)) \mid I_t = x]}\right) \tag{86}$$

$$= \frac{\sigma_t^2}{2}\nabla\mathbb{E}[r(x_1^a) \mid I_t^a = x]. \tag{87}$$

Recall that $\rho_{1|t,a}(x_1|x_t) = \mathcal{N}(x_t; \beta_t x_1, \alpha_t^2 I)\rho_{1,a}(x_1)/\rho_{t,a}(x_t)$. Thus the score function is

$$\nabla_{x_t}\log\rho_{1|t,a}(x_1|x_t) = \frac{\beta_t x_1 - x_t}{\alpha_t^2} - \nabla_{x_t}\log\rho_{t,a}(x_t). \tag{88}$$

Therefore the gradient of the conditional expectation from (87) is:

$$\nabla\mathbb{E}[r(x_1^a) \mid I_t^a = x] = \int r(x_1)\nabla_x\rho_{1|t,a}(x_1|x)dx_1 \tag{89}$$

$$= \int r(x_1)\rho_{1|t,a}(x_1|x)\nabla_x\log\rho_{1|t,a}(x_1|x)dx_1 \tag{90}$$

$$= \mathbb{E}\left[r(x_1^a)\left(\frac{\beta_t x_1^a - x}{\alpha_t^2} - \nabla_x\log\rho_{t,a}(x)\right) \mid I_t^a = x\right]. \tag{91}$$

By Tweedie's formula

$$\nabla_x\log\rho_{t,a}(x) = \mathbb{E}\left[\frac{\beta_t I_1^a - x}{\alpha_t^2} \mid I_t^a = x\right]. \tag{92}$$

Substituting this into (91), we obtain the covariance relation:

$$\nabla\mathbb{E}[r(x_1^a) \mid I_t^a = x] = \mathrm{Cov}\left(r(x_1^a), \frac{\beta_t x_1^a - x}{\alpha_t^2} \mid I_t^a = x\right) \tag{93}$$

$$= \frac{\beta_t}{\alpha_t^2}\mathrm{Cov}(r(x_1^a), x_1^a \mid I_t^a = x). \tag{94}$$

Substituting this back into (87):

$$\frac{\partial}{\partial a}\tilde{b}_{t,a}(x) = \frac{\sigma_t^2}{2}\frac{\beta_t}{\alpha_t^2}\mathrm{Cov}(r(x_1^a), x_1^a \mid I_t^a = x). \tag{95}$$

Now we examine the time derivative of the interpolant. We can rewrite $\dot{I}_t^a = \dot{\alpha}_t x_0 + \dot{\beta}_t I_1^a$ by substituting $x_0 = (I_t^a - \beta_t x_1^a)/\alpha_t$:

$$\dot{I}_t^a = \frac{\dot{\alpha}_t}{\alpha_t}I_t^a + \left(\dot{\beta}_t - \frac{\dot{\alpha}_t\beta_t}{\alpha_t}\right)x_1^a. \tag{96}$$

The Tilt Matching covariance update is given by Proposition 3.2:

$$\frac{\partial}{\partial a}b_{t,a}(x) = \mathrm{Cov}(r(x_1^a), \dot{I}_t^a \mid I_t^a = x) \tag{97}$$

$$= \left(\dot{\beta}_t - \frac{\dot{\alpha}_t\beta_t}{\alpha_t}\right)\mathrm{Cov}(r(x_1^a), x_1^a \mid I_t^a = x). \tag{98}$$

Comparing the coefficients, equality holds if and only if:

$$\frac{\sigma_t^2}{2}\frac{\beta_t}{\alpha_t^2} = \dot{\beta}_t - \frac{\dot{\alpha}_t\beta_t}{\alpha_t}. \tag{99}$$

Multiplying through by $\alpha_t^2/\beta_t$, we require $\frac{\sigma_t^2}{2} = \alpha_t^2\frac{\dot{\beta}_t}{\beta_t} - \alpha_t\dot{\alpha}_t$, which matches exactly the definition of $\sigma_t$ chosen in the setup. Thus $\tilde{b}_{t,a}(x) = b_{t,a}(x)$.

## D. Experiments

### D.1. Sampling Lennard-Jones Potentials

The Lennard-Jones (LJ) potential is a widely used mathematical model that describes the potential energy between two neutral, non-bonding particles. This energy is calculated as a function of the distances between particles, capturing the balance between long-range attractive forces and short-range repulsive forces. It has the form

$$E^{\text{LJ}}(x) = \frac{\epsilon}{2\tau}\sum_{ij}\left(\left(\frac{r_m}{d_{ij}}\right)^6 - \left(\frac{r_m}{d_{ij}}\right)^{12}\right), \tag{100}$$

where $d_{ij} = \|x_i - x_j\|$ is the distance between particles $i$ and $j$, $\epsilon$ is the potential well depth, $r_m$ is the equilibrium distance at which the potential is minimized, and $\tau$ is the system temperature. We follow (Köhler et al., 2020; Akhound-Sadegh et al., 2025) in adding a harmonic potential to the energy:

$$E^{\text{Total}}(x) = E^{\text{LJ}}(x) + \frac{1}{2}\sum_i\|x_i - \bar{x}\|^2, \tag{101}$$

where $\bar{x}$ is the center of mass of the system. We use the same parameters $\epsilon = 2.0, r_m = 1$ and $\tau = 1$ as (Akhound-Sadegh et al., 2025) for our experiments. For the LJ-13 and LJ-55 datasets we use samples provided by the codebase in (Akhound-Sadegh et al., 2025) which uses the No-U-Turn-Sampler (NUTS) (Hoffman & Gelman, 2011).

In our experiments we use an EGNN (Satorras et al., 2022). For LJ-13 we use three layers and 32 hidden dimensions which is approximately 45,000 parameters. For LJ-55 we use five layers and 128 hidden dimensions for a parameter count of approximately 580,000.

To compute the Effective Sample Size (ESS) we evaluate likelihoods $p_1(x_1)$ under our model $\hat{b}_t$ by

$$\log p_1(x_1) = \log p_0(x_0) - \int_0^1 \nabla \cdot \hat{b}_t(x_t)dt \tag{102}$$

to compute importance weights $w(x_1) = \frac{\rho_{1,a=0}(x_1)e^{r(x_1)}}{p_1(x_1)}$ and then compute the ESS as

$$\text{ESS} = \frac{(\sum_{i=1}^N w_i)^2}{N\sum_{i=1}^N w_i^2}. \tag{103}$$

For ITM we use a fixed step size of $h = 0.001$. We use 800 gradient steps per anneal update. We use a simple Euler integrator with 100 steps in each case. We use the linear interpolant $I_t = (1-t)x_0 + tx_1$ for our experiments.

### D.2. Algorithm Pseudocode

---

**Algorithm 1** Explicit Tilt Matching (ETM)

---

**Require:** Pretrained drift $b_{t,0}(x)$; reward $r(x)$; annealing schedule $\{a_k\}_{k=0}^K$; interpolant coefficients $\alpha_t, \beta_t$; epochs $E$; batch size $B$

**Ensure:** Tilted drift $b_{t,1}(x)$

1: **for** $k = 0, \ldots, K-1$ **do**
2:    # Current model is $b_{t,a_k}$; goal is to compute $b_{t,a_{k+1}}$
3:    Initialize $\hat{b}_t \leftarrow b_{t,a_k}$
4:    **for** epoch $= 1, \ldots, E$ **do**
5:       # Use $b_{t,a_k}$ to draw samples from $\rho_{1,a_k}$ (can be stored in a buffer)
6:       Draw $B$ samples $(x_0, x_1^{a_k}, t)$ with $x_0 \sim \rho_0, x_1^{a_k} \sim \rho_{1,a_k}, t \sim \text{Unif}[0,1]$
7:       $I_t^{a_k} \leftarrow \alpha_t x_0 + \beta_t x_1^{a_k}$;   $\dot{I}_t^{a_k} \leftarrow \dot{\alpha}_t x_0 + \dot{\beta}_t x_1^{a_k}$;   $h_k = a_{k+1} - a_k$
8:       $T_{t,a_k,h_k} \leftarrow b_{t,a_k}(I_t^{a_k}) + h_k r(x_1^{a_k})\big(\dot{I}_t^{a_k} - b_{t,a_k}(I_t^{a_k})\big)$
9:       $\mathcal{L} \leftarrow \frac{1}{B}\sum \|\hat{b}_t(I_t^{a_k}) - T_{t,a_k,h_k}\|^2$
10:     Update the parameters of $\hat{b}_t$ by gradient descent on $\mathcal{L}$
11:    **end for**
12:    Set $b_{t,a_{k+1}} \leftarrow \hat{b}_t$
13: **end for**
14: **return** $b_{t,1}$

---

---

**Algorithm 2** Implicit Tilt Matching (ITM)

---

**Require:** Pretrained drift $b_{t,0}(x)$; reward $r(x)$; annealing schedule $\{a_k\}_{k=0}^K$; interpolant coefficients $\alpha_t, \beta_t$; epochs $E$; batch size $B$

**Ensure:** Tilted drift $b_{t,1}(x)$

  **for** $k = 0, \ldots, K-1$ **do**
    # Current model is $b_{t,a_k}$; goal is to compute $b_{t,a_{k+1}}$
    Initialize $\hat{b}_t \leftarrow b_{t,a_k}$
    **for** epoch $= 1, \ldots, E$ **do**
       # Use $b_{t,a_k}$ to draw samples from $\rho_{1,a_k}$ (can be stored in a buffer)
       Draw $B$ samples $(x_0, x_1^{a_k}, t)$ with $x_0 \sim \rho_0, x_1^{a_k} \sim \rho_{1,a_k}, t \sim \text{Unif}[0,1]$
       $I_t^{a_k} \leftarrow \alpha_t x_0 + \beta_t x_1^{a_k}$;   $\dot{I}_t^{a_k} \leftarrow \dot{\alpha}_t x_0 + \dot{\beta}_t x_1^{a_k}$;   $h_k = a_{k+1} - a_k$
       $T_{t,a_k,h_k} \leftarrow b_{t,a_k}(I_t^{a_k}) + \big(e^{h_k r(x_1^{a_k})} - 1\big)\big(\dot{I}_t^{a_k} - \text{stopgrad}(\hat{b}_t)(I_t^{a_k})\big)$
       $\mathcal{L} \leftarrow \frac{1}{B}\sum \|\hat{b}_t(I_t^{a_k}) - T_{t,a_k,h_k}\|^2$
       Update the parameters of $\hat{b}_t$ by gradient descent on $\mathcal{L}$
    **end for**
    Set $b_{t,a_{k+1}} \leftarrow \hat{b}_t$
  **end for**
  **return** $b_{t,1}$

---

## D.3. Additional Tables

*Table 3.* Fine-tuning results on Stable Diffusion 1.5. We compare our method against Adjoint Matching (AM) (Domingo-Enrich et al., 2025). We report on ImageReward (Xu et al., 2023), CLIPScore (Hessel et al., 2021), HPSv2 (Wu et al., 2023), and DreamSim (Fu et al., 2023). For all metrics, higher values are better, as indicated by the up-arrow ($\uparrow$).

| Method | ImageReward ($\uparrow$) | CLIPScore ($\uparrow$) | HPSv2 ($\uparrow$) | DreamSim ($\uparrow$) |
|---|---|---|---|---|
| SD 1.5 (Base) | 0.1873 ± 0.0762 | 0.2746 ± 0.0032 | 0.2566 ± 0.0030 | **0.3849 ± 0.0105** |
| AM ($\lambda = 1$) | 0.2170 ± 0.0755 | 0.2754 ± 0.0032 | 0.2576 ± 0.0030 | 0.3826 ± 0.0104 |
| **ITM** ($\lambda = 1$) | **0.4465 ± 0.0709** | **0.2794 ± 0.0036** | **0.2659 ± 0.0027** | 0.3383 ± 0.0116 |
| AM ($\lambda = 10^2$) | 0.7873 ± 0.0689 | 0.2792 ± 0.0033 | 0.2791 ± 0.0028 | 0.3363 ± 0.0101 |

## E. Example Images from Fine-Tuning Experiments

In this Appendix, we display **random uncurated** images from the base model and the model fine-tuned under the Tilt Matching objective. It can be seen that images generated from the fine-tuned model better adhere to the given text prompt, which aligns with the numerical results in the main text.

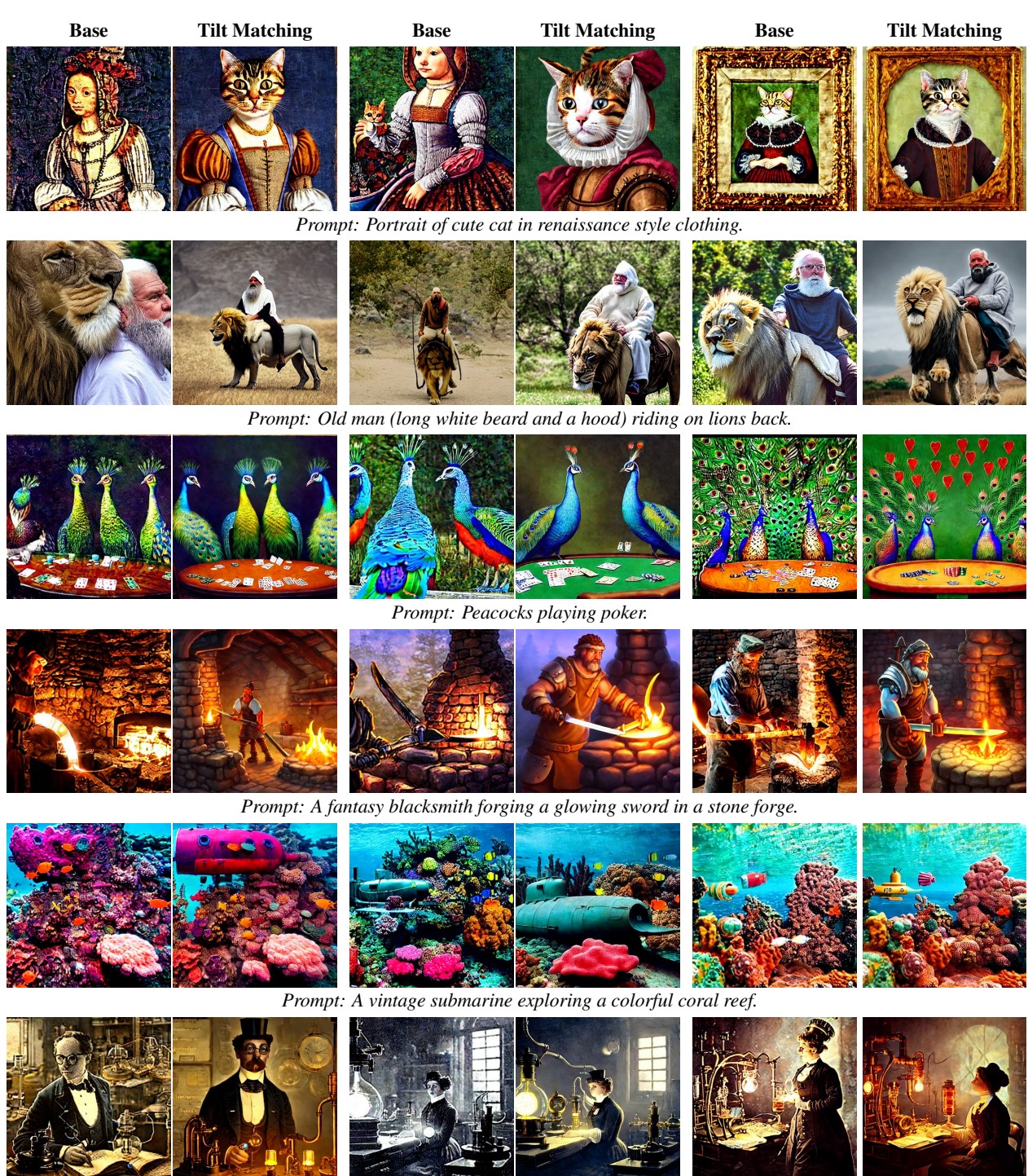

| Base | Tilt Matching | Base | Tilt Matching | Base | Tilt Matching |

*Prompt: Portrait of cute cat in renaissance style clothing.*

*Prompt: Old man (long white beard and a hood) riding on lions back.*

*Prompt: Peacocks playing poker.*

*Prompt: A fantasy blacksmith forging a glowing sword in a stone forge.*

*Prompt: A vintage submarine exploring a colorful coral reef.*

*Prompt: A Victorian-era scientist working in a glowing steampunk laboratory.*

*Figure 5.* Uncurated paired samples coming from the Base model vs our Tilt Matching method.

# LLM Usage

In preparing this paper, we used large language models (LLMs) as assistive tools. Specifically, LLMs were used for (i) editing and polishing the text for clarity and readability, and (ii) formatting matplotlib code. The authors take full responsibility for the content of this paper.