# OpenReview forum: "Discrete Tilt Matching"
_ICML.cc/2026/Conference — ICML 2026 regular_

### Official Review · Reviewer_fTJu · 2026-02-17

**Soundness:** 4
**Presentation:** 3
**Significance:** 3
**Originality:** 3
**Overall Recommendation:** 5
**Confidence:** 1

**Summary:**

This paper introduces Discrete Tilt Matching, an algorithm to, given a reward function, perform fine-tuning on masked diffusion models. DTM avoids the need to compute the likelihood of token sequences, mostly intractable in diffusion models, by sequentially tilting the unmasking posterior of the model by small increments h, with the ultimate goal of producing the terminal marginal that matches the fully reward-tilted distribution. The authors present extensive theory regarding DTM, derive the objective and the steps that make its optimization tractable, and conduct experiments both with masked diffusion models and fine-tuning a LLaDA-8B-Instruct model on a variety of tasks.

**Compliance With Llm Reviewing Policy:**

Affirmed.

**Final Justification:**

The authors' rebuttal was informative, and I believe the paper is of high quality.

**Key Questions For Authors:**

- The derivation of DTM in Section 3, and especially in Section 3.3, appears to hold for any annealing step size h. While I understand that the motivation is to use small h, which seems beneficial in terms of variance (as shown in Proposition 3.3), is it possible, at least in principle, to take $h = A$ and perform a single step instead? In that case, would the variance of the estimator in Eq. (22) be the main issue?

- As the authors note in Section 4.3, when fine-tuning LLaDA-8B-Instruct, DTM appears to outperform the baseline on Countdown and Sudoku (Table 1). However, these are arguably quite restricted tasks, making it difficult to assess how DTM would compare to other baselines in more realistic settings. Given that SPG appears to outperform DTM on MATH500, do the authors have any insights into how DTM might perform on more realistic tasks?

**Limitations:**

Yes

**Strengths And Weaknesses:**

**Strengths**

- Despite the technical depth of the paper, the authors succeed in providing sufficient detail and context for the reader to follow the material. In particular, the introduction and background on CTMC and MDM are quite satisfactory. Overall, the paper is very well written and well contextualized.

- The technical contribution of the paper appears significant and, to the best of my understanding, sound. The authors also provide rigorous proofs in the appendix.

**Weaknesses**
- The empirical evaluation of DTM is carried out in relatively simplified scenarios and tasks. While I do not have a deep understanding of whether this is common practice in the dLLM literature, it may be difficult for a general reader to assess whether DTM can be effectively applied at scale in real-world applications. Additionally, evaluations on a unique and relatively small model, such as LLaDa-8B-Instruct, do not provide a fully comprehensive picture. That said, I understand that the primary contribution of the paper is theoretical in nature.

- I believe there are some minor aspects to polish in the presentation. In my view, the role of Proposition 3.1 and how it connects with the following section and with the objective in eq 16 could be clarified further. In line 147, the bold text is probably meant to start a new paragraph. In line 163, has the notation $x[x^i \leftarrow v]$ been defined before? Is it standard? In line 203, "with" seems to be a typo for "will".



*Note: diffusion models are not my main area of expertise. My review is an educated guess, and my confidence score is chosen accordingly.*

---

> ### Author Rebuttal · Authors · 2026-03-31
>
> We thank the reviewer for the thoughtful and positive assessment of the paper, and especially for recognizing both the technical depth and the clarity of the presentation. Below we clarify on the reviewer’s concerns:
> 1. **Taking \$h=A\$ in one step**: Theoretically, our derivation and Proposition 3.2 do not place restrictions on \$h\$, so in principle one could set \$h=A\$ and attempt a one-shot update. But our choice of incremental annealing is instead for statistical and optimization reasons: when \$h\$ is large, the importance weights \$e^{hr(x_1)}\$ in Eq. (22) become much more variable, and the target tilted posterior can move much farther from the current model, which makes optimization substantially less stable. This is also consistent with the motivation cited in Section 2.3 from [1], which explicitly advocates proximal updates to avoid the mode-collapse behavior that can arise when directly targeting a difficult multimodal distribution in one shot. In addition, our own ablation in Section 4.2 shows the same phenomenon empirically: on the maze task, taking too big an annealing step leads to worse coverage and mode collapse. Thus, while a one-step tilt is valid in principle, our current evidence suggests that annealing with reasonable step sizes is important in practice as it mitigates variance and mode-collapse issues.
> 2. **On realism / scale of the experiments**: We agree that our empirical study is not intended to be a comprehensive real-world deployment evaluation. At the same time, we would like to emphasize that the benchmarks we use are standard benchmarks in the recent dLLM fine-tuning literature, and are also employed by the baselines we compare against, which allows for a controlled and apples-to-apples evaluation. Moreover, our large-scale experiment is conducted on LLaDA-8B-Instruct, and to the best of our knowledge this is already the largest model scale used in the current dLLM fine-tuning literature. So the evaluation is not limited to small synthetic models. Our intent was therefore to combine: (i) a theoretically focused contribution, with (ii) experiments on standard and accepted post-training benchmarks, and (iii) validation on a model scale that is already at the frontier of what prior masked diffusion fine-tuning baselines have demonstrated. We will revise the paper to make this positioning more explicit.
> 3. **Performance on MATH500**: We refer to paragraph 2 in our response to Reviewer vqS9.
> 4. **Proposition 3.1 and its role relative to Eq. (16)**: We thank the reviewer for pointing this out. Our intent is that Eq. (16) represents the ideal but intractable objective, while Proposition 3.1 is the key bridge that replaces the inaccessible target \$\pi_{a+h}\$​ with a tractable characterization (accessible via samples) in terms of weighted conditional moments under \$P_a\$​. Section 3.3 then uses this characterization to derive the practical \$c\$-DTM objective. We will revise the text to make this logical flow more explicit.
> 5. **Minor presentation issues**: We really appreciate the reviewer's careful reading in catching these typos and notational/formatting issues. In the final version, we will correct the formatting around line 147, explicitly define the notation \$x[x^i\leftarrow v]\$ as the partially denoised sequence \$x\$ with \$i\$-th coordinate replaced by the token \$v\$, and fix the typos.
> Overall, we appreciate the reviewer’s comments and believe the suggested clarifications will improve the paper.
>
> [1] Guo, W., Choi, J., Zhu, Y., Tao, M., and Chen, Y. Proximal diffusion neural sampler, 2025. arXiv:2510.03824v1

---

> > ### Author Rebuttal · Reviewer_fTJu · 2026-03-31
> >
> > I would like to thank the authors for their response. They addressed my questions, and I would like to keep my (already positive) opinion about the paper.

---

> > > ### Author Response · Authors · 2026-04-08
> > >
> > > Thank you for the follow-up. We are very glad that the rebuttal addressed your concerns about the annealing, the scope of our experiments and performance, and theoretical presentation. We also appreciate your positive view of the paper’s technical contribution and presentation.

---

### Official Review · Reviewer_mwrD · 2026-03-12

**Soundness:** 4
**Presentation:** 3
**Significance:** 3
**Originality:** 4
**Overall Recommendation:** 5
**Confidence:** 3

**Summary:**

This paper proposes a framework called Discrete Tilt Matching (DTM). This allows to finetune diffusion language models (dLLMs) without reliance on their exact likelihood estimation as it is required by policy gradient methods in RL. Here, DTM uses a cross-entropy loss with controllable variance.

On the experimental side, the framework is tested and compared on masked diffusion models (MDMs), over fine-tuning tasked widely used in the literature.

**Compliance With Llm Reviewing Policy:**

Affirmed.

**Final Justification:**

Overall, my questions are resolved to a satisfactory extent, and I would like to maintain my already positive score. I continue to view this as a technically solid and valuable paper addressing an important problem.

**Key Questions For Authors:**

I have three main questions.

1. Regarding the weakness, can the authors better disentangle the source of the empirical gains, and clarify how much comes from the DTM objective itself versus auxiliary choices such as SAR-aligned training, confidence-based decoding, replay buffering, and task-specific hyperparameter tuning?

2. The abstract claims lower compute cost than prior RL-based methods, but I could not find a detailed wall-clock or training-compute comparison. Could the authors report runtime / GPU-hours / update counts for DTM versus the main baselines?

3. A last question is about the generality of the method. The paper uses non trivial CTMC theory, and I wonder if DTM is also valid for other type of discrete diffusion models than masked diffusion (e.g. uniform diffusion). Can the authors provide a discussion about this, or explain if the applicability of the framework is limited to masked diffusion?

Despite these questions, I think that the paper is really solid. I am clearly in favor for acceptance.

**Limitations:**

A discussion about limitation is not provided. I suggest that the paper should discuss more wall-clock time or potential limitation to masked diffusion only.

**Strengths And Weaknesses:**

- Strenghts ;

  - The paper treats an understudied problem in the area of post training for diffusion models. It tackles a failure mode of the direct adaptation of policy gradient methods for dLLMs : the intractable exact likelihood computation that is totally overcomed.
  - The paper is well written, and it's quite clear to understand the background, the existing methods, and the problem it aims to tackle.
  - The theoretical part of the paper is technically solid. I have spend some time on the math and to me, the paper is dense but correct, with a non trivial theoretical work that should be highlighted.
  - Empirical results are interesting ; in particular in the Sudoku and Countdown benchmarks, the method seems to outperform current policy gradient methods used for finetuning dLLMs.

- Weaknesses ;

  - Maybe, one weakness is that, the paper does not really isolate what part of the gain comes from the new objective itself, versus the surrounding decoding / training choices.

---

> ### Author Rebuttal · Authors · 2026-03-31
>
> We thank the reviewer for checking the theory in detail and recognizing both the technical solidity and the empirical strength of the paper. We are encouraged by the reviewer’s observation that the theoretical contribution is nontrivial and addresses an important bottleneck of RL post-training for dLLMs. Below, we clarify the reviewer’s concerns.
> 1. **Source of empirical gains and role of auxiliary choices**: We agree it is important to disentangle the gains due to the DTM objective from those due to choices such as SAR alignment, confidence-based decoding and replay buffering.
> Our view is that **the core source of improvement is the DTM objective itself**: it replaces sequence-likelihood-based RL objectives, which are unnatural and intractable for MDMs, with a principled cross-entropy objective over local unmasking posteriors. This is the central algorithmic change; the other components are practical choices about how training states are sampled.
>
> (i) **Confidence-based decoding** is the standard inference strategy in dLLMs and in our baselines, so we adopt it by default to keep comparisons aligned with prior work. In Countdown, the originally reported mixed schedule (confidence-based decoding in the first three phases, then uniformly random within-block updates) was not essential: as discussed in paragraph 4 of our response to Reviewer zE25, DTM is indeed *robust* to both choices, showing the gains are not from that particular schedule.
>
> (ii) We view the **SAR-aligned interpolant/objective** as part of the DTM framework itself. Since SAR decoding is widely used in practice (including recent LLaDA-2.0 [1]), it is an advantage of DTM’s flexibility to focus training on partially denoised sequences that will be encountered at inference. Importantly, as shown in App. C.4, SAR alignment keeps the *same* minimizer of the objective. We also ran ablations showing that SAR-DTM outperforms the default random interpolant, which is consistent with improved train-test alignment. We refer to paragraph 3 in our response to Reviewer vqS9 for details on the ablation.
>
> (iii) **Replay buffering** is mainly a sample-efficiency device, not a conceptual component of DTM. It plays the same role as rollout reuse in RL, amortizing the cost of sampling by extracting multiple training states from one rollout. Thus replay buffering is not a source of the qualitative gains over RL baselines, but part of making DTM even more compute-efficient.
> Overall, we appreciate this question and will revise the paper to better disentangle the inherent benefit of DTM from the supporting but standard practical choices.
>
> 2. **Runtime comparison**: We refer to paragraph 3 in our response to Reviewer zE25.
> 3. **Generalization to other types of discrete diffusion**:
> We agree this is important, and our Appendix A helps clarify that DTM is not limited to MDMs at the level of its core principle.
> App. A shows our derivation is an instance of a general *Esscher-tilted regression / Bregman projection principle*: if a tilted measure \$P’\$ satisfies \$\frac{dP’}{dP}\propto w\$, then minimizing a weighted Bregman objective \$\mathbb E[w D_\Phi(T \\ || f(X))]\$ recovers the tilted conditional expectation \\[f^*(X)=\mathbb E’[T \mid X]=\frac{\mathbb E[wT \mid X]}{\mathbb E[w \mid X]}.\\]
> For other types of discrete diffusion models, the same Esscher / tilted-regression backbone still applies. What changes is just the choice of the random local target \$T\$ and thus the induced divergence/projection problem. (In masked diffusion, \$T\$ is a masked-token target and the projection is cross-entropy.)
> Concretely, consider a uniform-state CTMC where one coordinate at a time jumps to a uniformly chosen token. In that case, the natural reverse-time object is not an unmasking posterior, but a local reverse jump rate (or equivalently, a local density ratio). If \$x^{(\ell\to v)}\$ denotes the sequence obtained by replacing \$\ell\$-th coordinate of \$x\$ by token \$v\$, then the reverse dynamics are governed by \\[s^\star(x,t,\\ell,v)=\frac{\rho_t(x^{(\ell\to v)})}{\rho_t(x)}.\\]
>
> Thus, in the uniform-state case, the natural analogue of DTM would project onto this reverse-rate / local-ratio object. Applying the same argument gives the tilted target as \\[s^\star_{a+h}(x,t,\ell,v)=\frac{\mathbb{E}_a[e^{h r(x_1)}T(x_t,x_1,\ell,v)\mid x_t=x]}{\mathbb{E}_a[e^{h r(x_1)}\mid x_t=x]}.\\]
>
> This object is a positive density-ratio quantity rather than a probability vector, so the appropriate projection would be a Bregman divergence on \$\mathbb R_{>0}\$​, e.g. MSE or the scalar relative-entropy divergence \$y \log⁡(y/z)−y+z\$.
> Therefore, our framework can motivate works beyond masked diffusion. We will revise the discussion to make this distinction explicit and point readers to App. A as the abstract principle underlying the masked-diffusion construction.
>
> [1] Bie, et. al. LLaDA2.0: Scaling Up Diffusion Language Models to 100B. arXiv:2512.15745

---

> > ### Author Rebuttal · Reviewer_mwrD · 2026-04-02
> >
> > Thank you for the detailed clarifications.
> >
> > Your rebuttal addressed my main questions well. I particularly appreciated the discussion on the generality of the framework beyond masked diffusion, which helped clarify that the core Esscher / tilted-regression principle is broader, even if the masked-diffusion instantiation leads to the specific cross-entropy form studied here.
> >
> > I also appreciate the additional explanation regarding the empirical gains. While I still think the paper could benefit from stronger large-scale ablations to more cleanly isolate the contribution of the core DTM objective from the surrounding training and decoding choices, your response clarified well how these different components should be interpreted.
> >
> > Finally, the added wall-clock comparison against SPG is useful and addresses my compute question in a concrete way. I strongly encourage the authors to include this comparison in the final version, since it strengthens the paper significantly.
> >
> > Overall, my questions are resolved to a satisfactory extent, and I would like to maintain my already positive score. I continue to view this as a technically solid and valuable paper addressing an important problem.

---

> > > ### Author Response · Authors · 2026-04-08
> > >
> > > Thank you again for the thoughtful follow-up. We especially appreciate that you took the time to engage with the broader Esscher / tilted-regression view. We are also happy that our clarification helped convey the picture we intended: the gains are driven primarily by the DTM objective itself, with SAR alignment being a natural part of the framework that reflects its flexibility, while ingredients such as confidence-based decoding and replay buffering are standard practical ingredients. Finally, thank you for noting the value of the wall-clock comparison, and we will follow your suggestion to retain it (as well as other ablation results) in the final version.

---

### Official Review · Reviewer_zE25 · 2026-03-12

**Soundness:** 3
**Presentation:** 3
**Significance:** 3
**Originality:** 2
**Overall Recommendation:** 3
**Confidence:** 4

**Summary:**

This paper introduces Discrete Tile Matching (DTM), an iterative update scheme designed using the closed form of the KL regularized reward maximization problem. The authors start with a cross entropy loss iterative update, and after resolving two issues with this formulation, arrive at the c-DTM objective. The paper then tests this algorithm in three tasks (Math500, Countdown, and Sudoku) showing good performance in countdown and sudoku tasks.

**Compliance With Llm Reviewing Policy:**

Affirmed.

**Final Justification:**

(conditional on no reply to my new questions). Overall, I will maintain my score as I find it a little strange that they follow SGP directly but omit a task. Likewise, I think the novelty of the method compared to the continuous version is minor. However, there are many merits to this work, which other reviewers have discussed, hence my final score.

Update based on reply:

I am still inclined to retain my score, but would not be strongly opposed to acceptance. I think GSM8K should have been presented or this should have been mentioned, given that they replicate the environment of SPG (which also isn't directly mentioned). I believe that novelty of this work remains slightly limited compared to the other paper, but I am most encouraged by the mode preservation (which I hope that authors add into the manuscript and discuss thoroughly).

**Key Questions For Authors:**

1. Can the authors ablate the annealing schedule?
2. How stable is this method (page 7 discusses this briefly, but not very fully)? What is the justification for the changes between each setting?
(see weakness for other questions)
3. Because there is a direct correlation between performance and step size it seems, how does runtime compare to the other methods (mainly just SPG)
4. I notice that the SPG numbers are exactly the same as the SPG paper. Do we know the the training dataset, and environment is reproduced exactly?

**Limitations:**

yes

**Strengths And Weaknesses:**

**Strengths**
1. This paper does a good job introducing the problem set up and derivation of the algorithm.
2. This work seems to fairly compare against the other baselines, on tasks of importance.
3. The introduction of $T_c$, and Esscher transform is nice.

**Weaknesses**
1. The novelty of this work is slightly limited, drawing significant inspiration from the other work proposed by the authors, and applying it to the discrete-time scenario.
2. While the annealing process of $a$ is central to the work, it is not ablated or discussed in detail. Thus, it is still unclear the sensitivity of this method in this respect.
3. Experimentally, one would desire to see the value of the importance weight (and whether or not instability can arise from it).
4. Many design choices are left unjustified and change significantly between the problem settings. In particular, the discussion of Countdown "we use the same confidence-based decoding in the first three annealing steps to improve initial correctness, then switch to uniformly random within-block updates." (page 7) but this is not discussed.

---

> ### Author Rebuttal · Authors · 2026-03-31
>
> We thank the reviewer for the positive comments on our DTM algorithm derivation and empirical performance, as well as for raising a few thoughtful concerns. We now make further clarifications on these concerns and illustrate the effectiveness and efficiency of our method with more experiments.
> 1. **Novelty**: We appreciate this concern and agree it is important to distinguish DTM from a direct adaptation of continuous-space tilting work. Our novelty is threefold: (i) One key insight of DTM is to reformulate MDM post-training around local state-level unmasking posteriors, rather than intractable sequence-level likelihoods; (ii) we developed a discrete-specific cross-entropy objective with an explicit minimizer and terminal-KL control; and (iii) App. A develops a broader conditional-Esscher / tilted-regression framework that abstracts the derivation beyond the concrete DTM construction. We also view the SAR-aligned extension as a practically important contribution, since it resolves a train-test mismatch without changing the target minimizer. We provide more detail on these points in our response to Reviewer vqS9, paragraphs 1 and 3.
> 2. **Annealing step size, importance weights, and robustness**: We agree that the annealing step size \$h\$ is an important hyperparameter and deserves a fuller ablation. Following the reviewer’s suggestion, we performed ablations on Countdown with \$h\in\\{1.5, 3, 6, 12\\}\$, while matching the overall compute budget and keeping the remaining hyperparameters fixed. As shown in the plot (https://ibb.co/5WG4WV37), the best performance is attained with \$h=6\$, while DTM remains competitive across other choices of \$h\$, which indicates that the method is reasonably **robust** to the choice of $h$. We did *not* observe instability caused by the importance weighting in these runs. Empirically, the main effect of \$h\$ appears to be the expected tradeoff: when \$h\$ is too small, each phase makes only a limited update to the policy, and errors due to insufficient minimization in each phase accumulate; when $h$ is too large, each phase tilts toward a target that is farther away and optimization becomes less effective, with a greater risk of mode collapse (also consistent with [2] and our Fig. 3 ablation). A moderate value (\$h=6\$) provides the best balance between these two effects, achieving high convergence without mode collapse - Countdown questions naturally have two modes: ones solvable with addition/subtraction and ones requiring multiplication/division; the second mode is tiny with base LLaDA achieving only 1.95% accuracy, yet our \$h=6\$ fine-tuned model is able to **preserve the mode** and improve it to 2.78% accuracy, whereas this mode collapsed (0%) in the SPG fine-tuned model. We appreciate the reviewer’s suggestion on these ablations and will include the results and discussions in the final version.
> 3. **Compute comparison**: We performed a wall-clock comparison against SPG on Sudoku. As shown in the plot (https://ibb.co/FLWRdg81), **DTM converges to a higher final reward with substantially less compute**: DTM reaches reward >0.95 within about 40 total GPU hours, whereas SPG reaches around 0.8 after about 175 GPU hours. This is consistent with DTM benefiting from avoiding sequence-likelihood-based RL estimation, which reduces optimization bias. We will include this comparison and discussion in the final version.
> 4. **Design choices across settings, especially Countdown decoding**. Our original Countdown setup (using confidence-based decoding for the first three annealing steps and then switching to uniformly random within-block updates) was an accidental choice made during final experiments. To test whether this choice is essential, we ran an additional variant that keeps confidence-based decoding throughout training, and it performs comparably, achieving **81.6%** accuracy. This suggests that the reported performance is *not* dependent on that specific mixed decoding schedule. We will revise the paper to better separate what is essential to DTM from what was an accidental task-specific configuration.
> 5. **Evaluation protocols**: We appreciate this concern as it is an important question that we should have clarified. We developed our implementation directly on top of the SPG codebase and reused their evaluation pipeline as closely as possible, including the evaluation dataset, model loading, decoding strategy, temperature, and correctness criterion. Therefore, the SPG numbers in our table are exactly those from their setup by design. We did this precisely to ensure that comparisons are fair, so that the gains come from the training algorithm itself rather than confounding differences in evaluation. We will clarify this explicitly in the final version.
>
> [1] Bie, et. al. LLaDA2.0: Scaling Up Diffusion Language Models to 100B. arXiv:2512.15745
> [2] Guo, W., Choi, J., Zhu, Y., Tao, M., and Chen, Y. Proximal diffusion neural sampler, 2025. arXiv:2510.03824

---

> > ### Author Rebuttal · Reviewer_zE25 · 2026-04-03
> >
> > I thank the authors for their comments, additional experiments, and clarifications.
> >
> > I have a few follow up questions remaining:
> >
> > 1. The authors mention that in countdown mode preservation which is not the case in SPG. This is very compelling for DTM, but I did not find it in the manuscript.
> >
> > 2. The authors mention close-following to SPG, but omit GSM8K task as well. Is there a reason for this?

---

> > > ### Author Response · Authors · 2026-04-08
> > >
> > > We appreciate the reviewer’s thoughtful follow-up and the concrete questions and suggestions throughout the discussion. We are glad that the rebuttal helped clarify several of the reviewer’s concerns, including the annealing-step ablation and robustness, the runtime comparison, the role of the Countdown decoding choices, and our effort to match the baseline evaluation pipeline. We now clarify the reviewer’s additional questions.
> > > 1. Mode preservation: We agree that DTM’s ability to preserve the modes in Countdown is a compelling practical advantage, which was not explicitly mentioned in our original draft. We appreciate the reviewer’s suggestion and will include this result and discussion in the final version.
> > > 2. On GSM8K: We did not include the evaluation on GSM8K because, in the SPG codebase, the GSM8K setup differs qualitatively from the other benchmarks and was therefore harder for us to interpret as a clean apples-to-apples comparison. In particular, SPG uses different system prompts and correctness criteria at training time and evaluation time. As a result, the model is effectively trained and evaluated under different conditions: the prompt seen during training is not the same as the one used at test time, and the same response may be judged differently depending on whether it is scored during training or evaluation. This mismatch is also reflected in the reward trajectory reported in the SPG paper: although the base LLaDA model can achieve about 77% accuracy on GSM8K, the training reward starts from what corresponds to a much lower effective accuracy (roughly 30%, under their reward scaling, where a correct answer contributes 2.5 reward in addition to formatting rewards). Because of this inconsistency in the baseline implementation, we prioritized the other three benchmarks, where the setup was clearer and more directly reusable for fair comparison. That said, to address the reviewer’s concern as fully as possible, we corrected the training configuration and launched a GSM8K run. With this setup, we obtained 81.6% accuracy after 40 hours of training on 8 H100 GPUs. We expect this result can likely be improved further with longer training and additional hyperparameter sweeps.

---

### Official Review · Reviewer_vqS9 · 2026-03-16

**Soundness:** 3
**Presentation:** 3
**Significance:** 2
**Originality:** 2
**Overall Recommendation:** 5
**Confidence:** 2

**Summary:**

This paper addresses the problem that existing RL fine-tuning frameworks for dLLMs rely on computing the marginal likelihood, which is intractable. The proposed method, DTM, frames post-training as sampling from a reward-tilted distribution, thereby avoiding sequence-level likelihood estimation. DTM outperforms other RL-based baselines in Sudoku and Countdown, but not MATH500.

**Compliance With Llm Reviewing Policy:**

Affirmed.

**Final Justification:**

Authors' rebuttal addressed my concerns relating to originality and SAR sampling. The justification of DTM's performance for MATH500 is also reasonable and well supported with additional experiments. Therefore, I raised my score to 5. However, diffusion language model is not my primary area of expertise, so I would lower my confidence and encourage the AC to weigh my assessment with caution.

**Key Questions For Authors:**

1. Is there a reason why the proposed method did not outperform RL style baselines for MATH500?



2. The semi-autoregressive sampling is practically useful, but it is unclear how it affect model performance. Could the authors include a more direct comparison between standard DTM and DTM with SAR sampling?

**Limitations:**

The authors did not include limitations but included broader impacts.

**Strengths And Weaknesses:**

**soundness:** This paper is mathematically and empirically solid overall. How to generate a cross-entropy training objective for the tilted marginal distribution is well supported and clean. Experiments include both maze planning and fine-tuning LLaDA-8B-Instruct for MATH500, COUNTDOWN, and SUDOKU, demonstrating both empirically and quantitatively the effectiveness of the proposed model. That said, the proposed method seems to be a repackaging of existing ideas of exponential reward tilting, posterior matching, and incremental annealing for discrete masked diffusion models. The resulting method is technically sound and reasonable, but the paper in this current format does not show a deeper conceptual insight beyond being a well-designed objective for the discrete setting.


**presentation:** The paper is well organized and easy to follow. The motivation for avoiding intractable marginal likelihood and how to build a local optimization objective is clearly stated.


**significance:** Post-training diffusion language models is a relevant and timely problem, especially because likelihood-based RL fine-tuning is less natural in this setting than for autoregressive models. However, the performance of the proposed method falls short on a more unstructured and diverse task MATH500, which questions the significance of the method for its ability to generalize to unstructured tasks. Illustrating why the method is less performative on unstructured tasks than traditional RL-style finetuning will help clarify the type of tasks the proposed method will be applicable to.


**originality:** The paper presents a reasonable and well-executed adaptation of reward tilting to discrete masked diffusion models, but the overall contribution feels incremental. The method does not appear to introduce a substantially new pipeline for diffusion model post-training. Instead, it translates an existing tilt-matching perspective into the masked discrete setting and augments it with practical components such as semi-autoregressive sampling.

---

> ### Author Rebuttal · Authors · 2026-03-31
>
> We thank the reviewer for the careful reading and for recognizing the paper’s technical solidity and empirical strength. We address the concerns below.
> 1. **Originality and insights**: Our work is indeed inspired by prior reward tilting methods, but we believe it makes three distinct contributions:
> (i) **A paradigm shift from sequence-level likelihoods to state-level posteriors**. A main conceptual contribution is to reformulate MDM post-training as learning the *local state-level unmasking posterior* of the tilted target distribution, rather than optimizing *sequence-level likelihood-based* RL objectives. Prior work often adapts likelihood-based methods from AR models, which requires biased or high-variance estimators for MDMs. Our formulation instead targets the native object of MDMs and opens a more suitable design space for post-training.
> (ii) DTM does not simply import incremental reward tilting, but derives a **discrete-specific, cross-entropy** objective over local posteriors. Cross-entropy (equivalently, KL) is the canonical objective that respects the *geometry in discrete domains* and naturally *matches standard MDM pretraining*. It also gives both an explicit global minimizer and the terminal KL control in Prop. 3.4. Thus we view the paper as contributing a discrete masked-diffusion formulation with its own theoretical and practical consequences.
> (iii) **Generalized reward-tilting framework**: Beyond the DTM objective, Appendix A develops a general framework based on conditional Esscher transforms and tilted regression. This suggests extensions beyond masked diffusion to e.g. uniform-state diffusion (details in response to Reviewer mwrD, paragraph 3).
> 2. **Performance on less structured tasks such as MATH500**:
> We agree that on MATH500 our results are competitive rather than dominant. However, we do not view this as evidence against DTM’s core formulation. DTM trains directly on *state-level local unmasking posteriors* instead of estimating sequence-level likelihoods. We hypothesize such *direct supervision on intermediate denoising states improves local reasoning consistency*.
> At the same time, the harder problems in MATH500 place stronger demands on maintaining *globally coherent reasoning over longer and more diverse decoding trajectories*. In such settings, improvements in state-level posterior prediction are still valuable, but may translate less directly into final-answer accuracy. This is consistent with the fact that DTM improves from **36.0** at generation length 256 to **40.2** at length 512 on MATH500, where it is second-best and close to SPG’s **41.8**. This suggests that **additional decoding budget helps convert stronger local predictions into better long-horizon reasoning performance**.
> To further support the claim that DTM’s direct state-level training improves reasoning consistency, we will provide additional qualitative evidence from Countdown, where reasoning is more structured and easier to inspect. There, **DTM produces more stable reasoning traces and a more stable effective generation length** (https://ibb.co/mFbNhYp7), whereas sequence-level RL (e.g. SPG) can show a sharp drop in effective generation length, suggesting reward-hacking behavior in which the model effectively bypasses the intended reasoning process. In contrast, DTM maintains stable generation behavior and, upon manual inspection, produces substantially more coherent and correct reasoning traces.
> We therefore view the current MATH500 result not as contradicting DTM, but as indicating that for more open-ended math reasoning, the DTM-fine-tuned model may be more *sensitive to reasoning horizon or inference budget*. We will revise the paper to clarify this point and appropriately qualify our empirical claims.
> 3. **Role of SAR-aligned DTM and additional ablation**. SAR decoding is the default inference strategy in frontier dLLMs such as LLaDA and LLaDA 2.0 [1], and in all our baselines. A key advantage of DTM is that it can be adapted to SAR *without changing the objective’s minimizer* (App. C.4). This **addresses a train-test mismatch**: if training optimizes over arbitrary masked states that never arise at inference, compute is spent on irrelevant trajectories. By aligning DTM with the SAR interpolant used at test time, we concentrate training on the states that actually matter for inference while preserving the same theoretical target. Following the reviewer’s suggestion, we performed an ablation on Countdown comparing training with the default random interpolant against SAR-aligned DTM, while fixing all other hyperparameters. As shown in the plot (https://ibb.co/JRKyQvxM), SAR-aligned DTM reaches **>80%** accuracy versus **～67%** for the random-interpolant version. This shows that SAR alignment is a practical advantage of DTM framework as it improves train-test alignment while retaining the same underlying optimum.
>
> [1] Bie, et. al. LLaDA2.0: Scaling Up Diffusion Language Models to 100B. arXiv:2512.15745

---

> > ### Author Rebuttal · Reviewer_vqS9 · 2026-04-03
> >
> > I would like to thank the authors for addressing the concerns relating to originality and SAR sampling. The justification of DTM's performance for MATH500 is also reasonable and well supported with additional experiments. Therefore, I will raise my score to 5. However, diffusion language model is not my primary area of expertise, so I would lower my confidence and encourage the AC to weigh my assessment with caution.

---

> > > ### Author Response · Authors · 2026-04-08
> > >
> > > Thank you again for your careful reading of our paper. We sincerely appreciate your thoughtful follow-up after the rebuttal, and we are glad that the additional clarification and experiments helped address the concerns regarding originality, SAR-aligned DTM, and the MATH500 results. We are grateful for your positive assessment of the paper’s technical soundness and empirical strengths, and for your increased score.

---

### Decision · Program_Chairs · 2026-04-30

**Decision:**

Accept (regular)

**Comment:**

The paper introduces Discrete Tilt Matching (DTM), a novel masked diffusion LLM (dLLM) fine-tuning algorithm.  Given an RL reward function, direct computation of a dLLM's marginal likelihood is practically intractable.  To render RL feasible, DTM reformulates MDM in terms of local unmasking posteriors, thus allowing incremental updates (with theoretical guarantees) which allow tractable fine-tuning.  DTM was applied to Llada-8B-Instruct, compared to several recent dLLM preference-tuning algorithms--i.e., Llada-1.5, D1, WD1, SPG, and UniGRPO--and evaluated on several standard dLLM benchmarks.  On the majority of tasks, DTM outperforms the other SOTA algorithms by a wide margin.  Given the recent push to scale dLLMs to higher parameter counds (e.g., Llada-2.0), this work is timely and has the potential for significant impact.

All reviewers agree on the merits of the work--the paper is well written, the performance of DTM is impressive, and the theoretical framework is sound--with the majority of reviewers in favor of acceptance.  Throughout the rebuttal, the reviewers asked a large number of questions and raised concerns regarding originality/novelty and the lack of GSM8K evals.  However, the authors did well to clarify questions and address primary concerns.  I ask that authors include the final GSM8K results described during the rebuttal in the manuscript.  Finally, I encourage the authors to release their adjusted SPG codebase containing DTM, which will greatly help foster further growth in this line of work.